# GraphIP–Bench: A Benchmark for Extraction Attacks and Defenses in Graph Learning

## Abstract

Graph machine learning models support applications in recommendation, finance and biomedicine, yet their parameters and training data are proprietary assets that face threats such as model extraction, inversion and membership inference. Prior work proposes many attacks and defenses, but comparisons remain unreliable because experiments use inconsistent datasets, threat models and evaluation metrics. We introduce *GraphIP-Bench*, a unified benchmark and library that provides standardized datasets, reference implementations of diverse attack and defense families and a common evaluation protocol. The benchmark specifies clear metrics for extraction fidelity, task utility and computational cost, which together measure the protection–utility trade-off under a prescribed query-access threat model. Empirical analysis across citation, coauthor and commercial graphs shows that several watermarking methods preserve utility and enable ownership verification with different cost profiles, while data-free extraction lags behind data-driven extraction even at large budgets. To the best of our knowledge, this is the first benchmark that standardizes the rigorous evaluation of model-extraction attacks and defenses for graph neural networks. Our implementation is publicly available at: https://anonymous.4open.science/r/GraphIPBench-7F7F.

## 1 Introduction

Graph neural networks (GNNs) are key components of modern data driven services. Commercial platforms use them for product recommendation (Yang et al., 2023), autonomous vehicle perception (Monti et al., 2021), and molecular property prediction (Li et al., 2024). Their main advantage is that they aggregate information over arbitrary relational structures, which arise in social, financial, and product data. Cloud providers now expose pretrained GNNs through public inference endpoints, which allow customers to deploy state of the art analytics without local training or data collection.

However, public access to these endpoints creates new risks and vulnerabilities. Adversaries can submit carefully designed queries and use the returned labels or confidence scores to train a surrogate that reproduces the target model's behavior, which is known as model extraction (Wu et al., 2021; Zhuang et al., 2024; Zhang et al., 2022b). A successful surrogate leaks the owner's intellectual property, reduces pay per query revenue, and enables competitors to recreate proprietary functionality at low cost. For example, a copied GNN can expose decision boundaries that criminals exploit to bypass screening, and in pharmaceutical research it can reveal assay knowledge that is encoded in model parameters. These risks threaten both revenue and privacy, which motivates the need for reliable evaluation of extraction threats and defenses.

To mitigate these risks, recent work proposes several defenses. Output perturbation and query filtering aim to limit the information that each response reveals (Liang et al., 2024; Kesarwani et al., 2018; Hu et al., 2024). Watermarking and fingerprinting embed verifiable patterns in trained graph models so that unauthorized copies can be identified (Wang et al., 2023; Xu et al., 2023; Dai et al., 2024a; You et al., 2024; Wu et al., 2024a; Waheed et al., 2024). Surveys summarize these techniques and the broader security landscape for deep learning (Peng et al., 2023; Xue et al., 2021; Sun et al., 2023b; Zhao et al., 2025a;b) and for graph learning models in particular (Dai et al., 2024b; Wu et al., 2022c; Zhang et al., 2022a; Wu et al., 2022b; Sun et al., 2024; Zheng et al., 2021). However, experimental practice remains fragmented. Studies rely on private data splits, incompatible query budgets, and inconsistent metrics, and the few existing benchmarks focus on robustness or privacy while ex-

cluding model extraction and omitting watermarking and fingerprinting baselines (Sun et al., 2023a; Zheng et al., 2021). As a result, the community lacks a consistent empirical basis to assess progress.

Several challenges must be addressed to enable fair and informative comparison. First, the community needs a single experimental protocol that fixes public data splits, shared query sets, query budgets, and explicit threat models, including whether the endpoint returns only labels or also confidence scores, and that treats data driven and data free attacks on equal terms (Wu et al., 2021; Zhang et al., 2022b). Second, evaluation must align success criteria with method goals: residual extraction should be measured for attacks, while ownership verification should be measured for watermarking and fingerprinting, since these mechanisms aim to provide verifiable evidence of model ownership rather than to reduce agreement with the target (Wang et al., 2023; Xu et al., 2023). Third, studies should report the balance between protection and utility under matched conditions, since practical adoption depends on how defenses affect task accuracy and inference latency. Fourth, benchmarks should include computational complexity and practical efficiency, which cover time and memory use and make transparent the real cost of deployment; prior work often omits these costs, which obscures feasibility (Liang et al., 2024; Hu et al., 2024). Finally, consistent method naming, representative citations, standardized hardware and software settings, and public reporting of random seeds and tuning procedures are necessary to ensure reproducibility and to prevent protocol induced bias. Existing studies only partially satisfy these requirements, which limits both scientific understanding and industrial uptake.

We address these challenges with *GraphIP-Bench*, a reproducible benchmark and library that we build for standardized evaluation. The suite integrates nine representative extraction attacks and six defense families and evaluates them on five public graph datasets under a single, clearly specified black box threat model. A unified hyperparameter search and a consistent metric suite record security, utility, and efficiency, which provide a complete and comparable view of attack and defense performance. To the best of our knowledge, *GraphIP-Bench* is the first benchmark that offers a standardized and rigorous evaluation of model extraction attacks and defenses for graph neural networks. The main contributions are:

- **Unified protocol.** We fix public splits, shared query sets, standardized budgets, and explicit endpoint assumptions, and we implement all nine attacks and six defenses under identical settings, which eliminates incompatible setups and enables fair comparison.
- **Goal aligned evaluation.** We separate an extraction track that measures residual agreement with the target and downstream utility from an ownership track that evaluates watermarking and fingerprinting on a standardized verification set, so that each method is judged by outcomes that match its objective.
- **Reproducible infrastructure.** We release reference implementations, configuration files, and a unified search procedure with fixed seeds on five public datasets and a shared hardware and software stack, which supports reliable replication and extension.
- **Protection–utility characterization.** We sweep defense configurations and attacker budgets and summarize operating points with attack agnostic frontiers, which make the protection–utility balance transparent for deployment decisions.
- **Complexity and efficiency reporting.** We provide asymptotic formulas and automated profilers that record training time, memory use, inference latency, verification time, and estimated monetary cost, which reveal the practical cost of each method.

## 2 PRELIMINARIES

**Notation.** We denote an attributed graph by $\mathcal{G} = (\mathcal{V}, \mathcal{E}, \mathbf{X}, \mathbf{A})$, where $\mathcal{V}$ is the node set, $\mathcal{E}$ is the edge set, and $\mathbf{X}$ is the node–feature matrix; $\mathbf{A}$ is the adjacency matrix (binary for unweighted graphs and real-valued for weighted graphs). We use four data–availability regimes for extraction: `both` (features and structure), `features only`, `structure only`, and `data free`. Query budgets are reported as multiples of the test-set size. Fidelity denotes the agreement rate between a surrogate and the target on the test split; accuracy and macro F1 denote task utility.

**Model Extraction Attacks.** We consider a standard black box threat model in which an adversary has query access to a deployed graph neural network and has no knowledge of its weights, architecture, or training data. The adversary submits inputs, which are either genuine graph instances or

synthetic samples, and records the returned labels or probability vectors. The adversary then trains a local model to minimize the discrepancy between these outputs and its predictions. The resulting model is a surrogate, which replicates the behavior of the protected network and enables extraction of the owner's intellectual property (Wu et al., 2021; Zhuang et al., 2024; Zhang et al., 2022b). Prior work identifies three main query strategies. Random querying submits subgraphs from public data and can succeed when the decision boundary is smooth (Wu et al., 2021). Adaptive querying selects inputs that maximize information gain, which can use disagreement between the current surrogate and the target model (Zhuang et al., 2024). Data free generation removes the need for public data by training a graph generator, which produces queries during extraction (Zhang et al., 2022b). These studies show that modest query budgets, which are often no larger than the number of nodes in a benchmark dataset, can recover a model that matches the original on downstream tasks.

**Defense against Model Extraction Attacks.** Defenses against model extraction can be divided into two categories. *Information limiting* methods modify the target's outputs so that an adversary receives less useful signal. Output perturbation adds calibrated noise or rounds confidence scores (Liang et al., 2024; Kesarwani et al., 2018), and query filtering detects and blocks suspicious request patterns (Hu et al., 2024). These methods can reduce residual agreement with a surrogate, although they may also degrade accuracy for legitimate users. *Ownership tracing* methods embed artifacts that allow the owner to verify infringement. Graph watermarking modifies weights or decision regions so that the model reveals a secret on inputs that carry a trigger (Wang et al., 2023; Xu et al., 2023; Dai et al., 2024a). Fingerprinting instead derives stable signatures from the model's output distribution, which leaves the original parameters unchanged (You et al., 2024; Wu et al., 2024a; Waheed et al., 2024). Surveys of these approaches identify open questions about robustness, utility loss, and verification cost (Peng et al., 2023; Xue et al., 2021; Sun et al., 2023b; Zhao et al., 2025a;b). Our benchmark places these methods under a single protocol with matched datasets, budgets, and threat assumptions, which enables objective comparison of their practical trade offs.

## 3 BENCHMARK DESIGN

In this section we describe the experimental protocol of *GraphIP-Bench*. We first state the protocol design, datasets, attacks, defenses, and implementation details, then articulate the four research questions that guide our empirical study.

### 3.1 EXPERIMENTAL SETTINGS AND IMPLEMENTATIONS

**Protocol Design.** *GraphIP-Bench* defines a single black box protocol that fixes four disjoint splits for each dataset (train, validation, test, query), shares the same query sets across methods, and uses *standardized* query budgets at 0.05, 0.10, 0.25, 0.50, and 1.00 times the test size, where "standardized" means that every method receives the *same* number of queries at each ratio and that these ratios are fixed across datasets. The set spans the commonly studied ranges in prior work (Orekondy et al., 2019; Tramèr et al., 2016), which include very small budgets that test sample efficiency (0.05 to 0.10), medium budgets where most gains occur (0.25 to 0.50), and a large budget that approximates saturation (1.00). This design makes results comparable and representative across methods and datasets. The protocol states explicit endpoint assumptions, which include whether the service returns labels only or also confidence scores and whether rate limits apply (Wu et al., 2021; Zhang et al., 2022b). We separate evaluation into an extraction track and an ownership track. The extraction track measures how well black box attacks learn a surrogate of an undefended target, and it reports test accuracy with respect to ground truth and fidelity with respect to the target. The ownership track evaluates watermarking or fingerprinting on a defended target without pairing each defense to a particular attack, and it reports defended accuracy, fidelity to the original target, and verification on a standardized verification set (Wang et al., 2023; Xu et al., 2023; Dai et al., 2024a; Zhang et al., 2024; Wu et al., 2024b). To place different settings on equal footing, we control data availability with four regimes: features only, structure only, features and structure, and data free. We report security, utility, and efficiency include total attack time, total defense time, and peak GPU memory for both attacks and defenses to make deployment cost clear.

**Datasets.** We use seven attributed graphs from three domains: citation (Cora, CiteSeer, PubMed), coauthor networks (CoauthorCS, CoauthorPhysics), and product co-purchase (Computers, Photo). The graphs differ in size, class balance, and feature dimension, which enables stress testing across

structural and statistical regimes; detailed statistics are in Appendix B. For extraction we split each dataset into four disjoint subsets—train, validation, test, and query—and we ensure no overlap. For watermarking we reserve a fixed subset of the training data as the watermark set.

**Metrics.** We report two groups of metrics under a single protocol. *Performance* for attacks includes test accuracy, macro F1, and fidelity which is the agreement between the surrogate and the target on the test set. *Performance* for defenses includes the defended model's test accuracy and macro F1, its fidelity to the original target, and ownership verification on a standardized verification set. *Efficiency* for attacks includes total attack time and peak GPU memory; *efficiency* for defenses includes total defense time and peak GPU memory.

**Model Extraction Attacks.** We evaluate a suite of black box attacks that covers both data driven and data free settings under the unified protocol in Section 3.2. The suite includes six attack methods from a canonical study on query synthesis and surrogate training, which we denote as `MEA0` through `MEA5` (Wu et al., 2022a), an adaptive adversarial querying method `AdvMEA` (DeFazio & Ramesh, 2019), a progressive centrality and entropy strategy `CEGA` (Wang et al., 2025), and a structure aware pipeline `Realistic` (Guan et al., 2024). To incorporate fully data free settings in a fair black box manner, we include three variants following Zhuang et al. (2024) and remove any use of target gradients: `DFEA_I` learns a surrogate by minimizing the KL divergence between surrogate logits and target logits (soft-label distillation), `DFEA_II` trains on hard labels returned by the endpoint (label-only supervision), and `DFEA_III` augments label-only training with a consistency loss between two surrogates. We treat distinct hyperparameter settings of the same algorithm as separate methods, which enables fine-grained comparison under shared query sets and budgets. To place heterogeneous input conditions on equal footing, we control feature and adjacency availability with `attack_x_ratio` and `attack_a_ratio` and evaluate four regimes: `features only` (denoted as X-only), `structure only` (denoted as A-only), `features and structure` (denoted as both), and `data free` (denoted as data-free). The extraction track reports test accuracy with respect to ground truth and *fidelity*, which is the agreement between the surrogate and the target on the test split; all metrics are averaged over three seeds.

**Model Extraction Defenses.** We focus on ownership and integrity mechanisms that permit post hoc verification. The benchmark includes four watermarking methods `RandomWM`, `BackdoorWM`, `SurviveWM`, and `ImperceptibleWM` (Zhao et al., 2021; Xu et al., 2023; Wang et al., 2023; Zhang et al., 2024), and a query based integrity verification method `Integrity` (Wu et al., 2024b). Each defense protects the same target architecture. We report the defended model's test accuracy and its fidelity to the original target. For watermarking we evaluate ownership verification on a standardized verification set using accuracy, depending on the method. For integrity verification we report detection and verification results on the same standardized setting. Models are labeled as *target* for the original model, *defense* for the protected model, *surrogate* for the extracted model, and *independent* for a separately trained baseline.

## 3.2 RESEARCH QUESTIONS

**RQ1. How does the effectiveness of extraction attacks change as the query budget increases?** We evaluate black box extraction on undefended targets under shared query sets with budgets equal to 0.05, 0.10, 0.25, 0.50, and 1.00 times the size of the test split. We treat distinct hyperparameter settings as separate variants and average results over three random seeds. We also control feature and adjacency availability with four regimes: features only, structure only, features and structure, and data free. We report test accuracy, F1 score, and fidelity as evaluation.

**RQ2. How effective are ownership and integrity defenses when they protect the same target model?** We evaluate watermarking methods RandomWM, BackdoorWM, SurviveWM, and ImperceptibleWM, together with the query based integrity method Integrity. Each defense protects an identical target architecture. We report the defended model's test accuracy and its fidelity, and we report ownership verification on a standardized verification set using accuracy as appropriate.

**RQ3. How well do defenses balance protection and utility at their selected configurations?** We characterize the protection–utility balance for each defense by reporting utility loss, which is the drop in test accuracy relative to the undefended target, and fidelity shift. We present these values at the selected configuration for each defense.

**RQ4. What are the computational complexity and practical efficiency costs of attacks and defenses?** We report asymptotic time and memory complexity as functions of graph size, feature dimension, training epochs, and query budget, with derivations in the Appendix. On NVIDIA A100 hardware we measure total attack time and peak GPU memory for the extraction track. For the ownership track we measure total defense time and peak GPU memory.

## 4 EMPIRICAL INVESTIGATION

We now present an empirical study that follows the unified protocol in Section 3, uses shared query sets and standardized budgets, and reports security, utility, and efficiency under identical hardware and software settings.

### 4.1 BUDGET SENSITIVITY OF MODEL EXTRACTION ATTACKS (RQ1)

To answer RQ1, we evaluate twelve black box attacks on undefended targets under the unified protocol in Section 3.2. We average results over three seeds and report fidelity to the target and accuracy with respect to ground truth. Table 1 shows a representative case on *Computers* when both features and adjacency are available. Figure 1 summarizes sample efficiency across all seven datasets and four regimes; the $y$-axis is the *median budget to* 90% *of best fidelity*, which for each attack is computed by finding, on every dataset and regime, the smallest budget that reaches at least 90% of that attack's best fidelity and then taking the median of these budgets across datasets and regimes (lower is better). Figure 2 reports regime sensitivity across budgets; each cell is a *fidelity ratio vs.* `both` at a fixed budget, defined as fidelity in a constrained regime divided by fidelity in the `both` regime for the same attack. Figure 3 plots accuracy, fidelity, and F1 across datasets to reveal

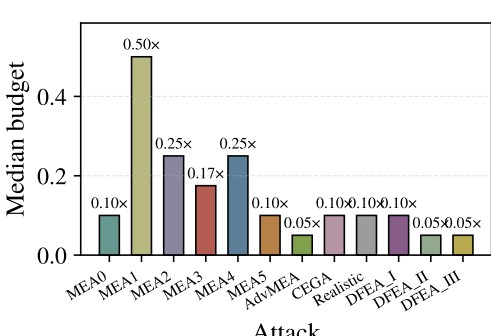

Figure 1: Sample efficiency across seven datasets and four regimes. Bars show, for each attack, the median budget level at which the attack reaches ninety percent of its own best fidelity under the same dataset and regime. Lower bars indicate better sample efficiency.

full learning trajectories. For more detailed results, please check Appendix D.1.

Table 1: RQ1 on *Computers* (both features and adjacency available). Fidelity (%) across query budgets with mean $\pm$ standard deviation over three seeds. Higher is better.

| Attack method | 0.05× | 0.10× | 0.25× | 0.50× | 1.00× |
|---|---|---|---|---|---|
| MEA0 | 51.0 ± 7.01 | 67.3 ± 3.12 | 67.0 ± 1.53 | 71.4 ± 2.71 | 80.5 ± 4.63 |
| MEA1 | 41.5 ± 15.5 | 27.7 ± 14.0 | 46.2 ± 5.54 | 49.4 ± 5.64 | 56.3 ± 12.6 |
| MEA2 | 17.0 ± 7.12 | 31.5 ± 8.22 | 32.2 ± 10.4 | 26.3 ± 12.5 | 43.1 ± 6.32 |
| MEA3 | 53.8 ± 3.01 | 64.8 ± 5.54 | 67.6 ± 6.21 | **74.9 ± 4.33** | 79.5 ± 5.2 |
| MEA4 | 38.4 ± 13.4 | 37.9 ± 12.6 | 63.6 ± 4.03 | 74.3 ± 5.02 | **83.7 ± 4.28** |
| MEA5 | 61.3 ± 4.32 | 65.6 ± 6.41 | 66.5 ± 2.11 | 73.9 ± 3.51 | 76.3 ± 4.16 |
| AdvMEA | 35.5 ± 17.8 | 25.7 ± 24.7 | 26.9 ± 22.4 | 46.1 ± 12.5 | 22.8 ± 11.8 |
| CEGA | 36.0 ± 25.0 | 36.7 ± 23.0 | 43.4 ± 17.5 | 53.9 ± 29.2 | 54.1 ± 33.4 |
| Realistic | **64.5 ± 33.7** | **67.9 ± 45.4** | **67.7 ± 45.3** | 67.4 ± 44.0 | 67.0 ± 44.8 |
| DFEA_I | 51.7 ± 7.23 | 49.2 ± 4.63 | 43.6 ± 4.91 | 16.6 ± 8.31 | 21.4 ± 14.8 |
| DFEA_II | 21.4 ± 12.4 | 25.0 ± 6.92 | 14.4 ± 4.23 | 22.5 ± 15.5 | 21.5 ± 14.7 |
| DFEA_III | 8.43 ± 6.85 | 18.1 ± 15.8 | 21.5 ± 14.7 | 21.5 ± 14.7 | 21.5 ± 14.7 |

*First, CEGA is the most sample-efficient attack.* It reaches high fidelity at very small budgets on *Computers* and shows the same early gains on other datasets (Table 1, Figure 3). This follows from selecting high-centrality and high-entropy nodes, which increases the information per query and covers decision boundaries quickly; the sample-efficiency bar confirms the smallest median budget (Figure 1). *Second, the MEA family saturates once the budget reaches a medium level.* Fidelity improves rapidly up to about 0.50× and then changes little, which indicates diminishing returns when additional queries become near duplicates of explored regions. *Third, data-free variants re-*

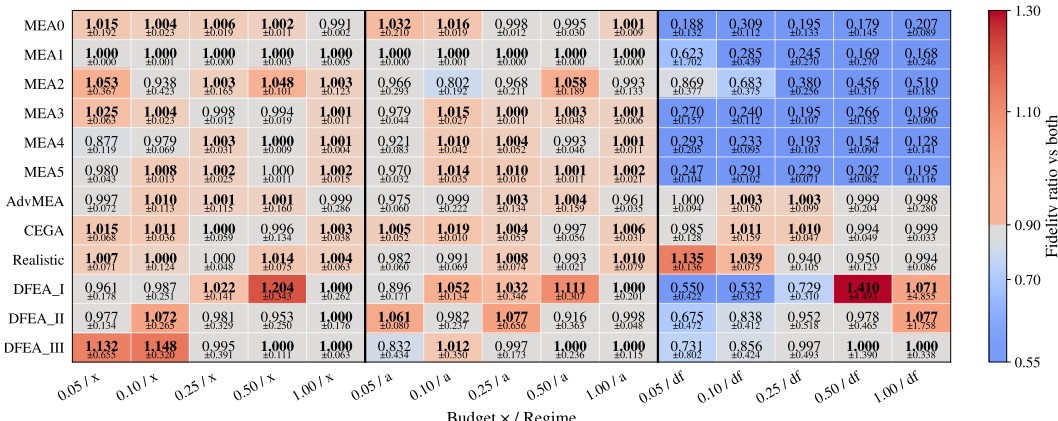

Figure 2: Regime sensitivity across budgets. Cells show the ratio between the average fidelity in a constrained regime and the average fidelity in the features and structure regime for the same attack at the same budget; darker color indicates larger drops. The map aggregates over seven datasets and separates dependence on features and adjacency from raw budget effects.

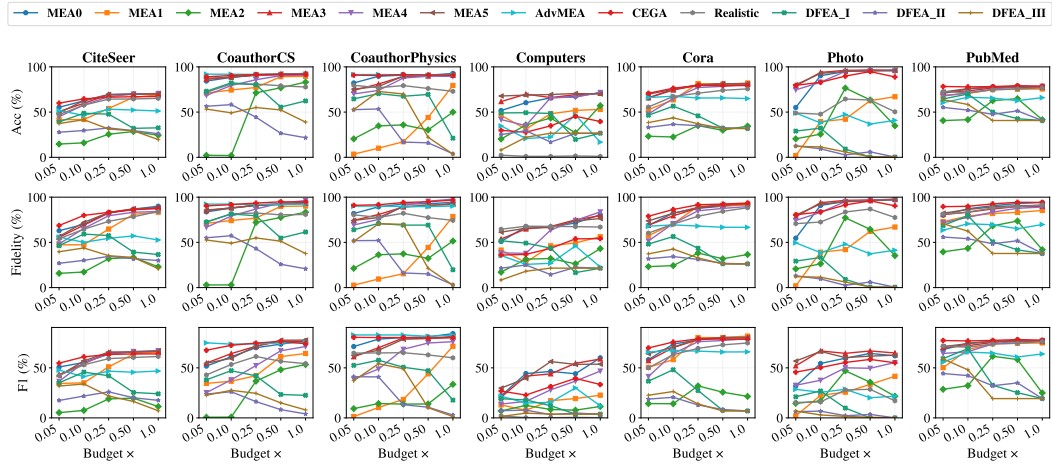

Figure 3: Budget–metric curves across seven datasets. Columns correspond to datasets and rows show accuracy, fidelity and macro F1. Lines are the twelve attacks with mean over three seeds; shaded bands indicate standard deviation.

*main well below data-driven methods at every budget and can even degrade at large budgets.* The decline arises from distribution mismatch: synthetic graphs and resampled features do not match the joint structure–feature statistics of the target, so the surrogate overfits synthetic correlations as synthetic queries dominate. *Fourth, strong data-driven attacks are insensitive to removing either features or structure but drop sharply in the data-free regime, whereas data-free variants are most sensitive to the absence of real inputs.* Ratios for CEGA and strong MEA variants stay near one in the features-only and structure-only blocks, while many methods fall well below one in the data-free block (Figure 2), which shows that real features or real edges alone already provide sufficient signal for effective querying and imitation. Overall, *effective extraction in graphs is primarily a sample-efficiency problem rather than a brute-force budget problem*, because methods that prioritize informative nodes and diverse neighborhoods reach peak fidelity quickly and then stabilize; *fully data-free extraction is unlikely to close the gap without better generative priors that match the joint statistics of structure and features*, since larger synthetic budgets may harm fidelity.

Table 2: RQ2 summary across seven datasets. Median (IQR) over datasets (higher is better). Utility drop is the absolute drop in test accuracy (pp) versus the undefended target (lower is better).

| Defense | F1 (%) | Fidelity (%) | Owner. verif. (%) | Utility drop (pp) ↓ | Time (s) | Peak mem. (GB) |
|---|---|---|---|---|---|---|
| RandomWM | 64.99 (12.02) | 74.13 (10.70) | 72.00 (24.7) | 3.93 (6.18) | 34.8 (14.6) | **0.09** (0.26) |
| BackdoorWM | 69.13 (15.51) | **80.07** (15.95) | **100.0** (0.00) | 3.27 (2.87) | 1.98 (0.45) | 0.16 (0.68) |
| SurviveWM | 67.47 (27.86) | 79.93 (32.92) | 21.76 (32.4) | **0.13** (18.2) | 2.27 (0.92) | 0.32 (0.96) |
| ImperceptibleWM | 69.49 (9.19) | 77.63 (13.88) | **100.0** (0.00) | 1.65 (6.28) | 676(697) | 2.30 (2.58) |
| Integrity | **73.43** (35.00) | 76.03 (22.52) | 66.67 (50.0) | 4.03 (21.8) | **1.38** (0.45) | 0.20 (0.90) |

## 4.2 Effectiveness of Ownership and Integrity Defenses (RQ2)

To answer RQ2, we evaluate watermarking methods in Section 3. All defenses protect the same target under identical data splits and a shared software stack. We report task utility (F1 and accuracy), alignment with the original target (fidelity), and ownership verification on a fixed verification set; all results are averaged over three seeds. In Figures 4 and 5, *utility drop (pp)* is defended accuracy minus undefended accuracy on the same dataset, and *ownership verification (%)* is accuracy on the verification set. Across datasets we summarize with the *median* and show variability with the *interquartile range (IQR)*. Figure 4 summarizes utility drop and verification across seven datasets, and Table 2 reports median performance with interquartile ranges together with time and memory. For more detailed results, please check Appendix D.2.

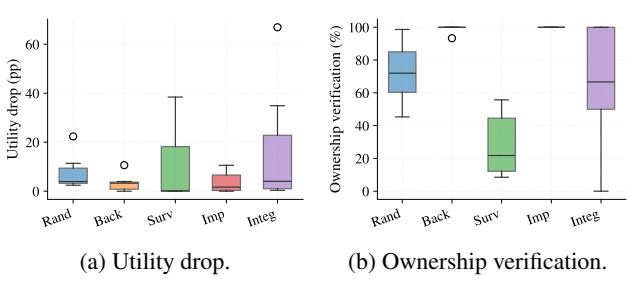

(a) Utility drop.  (b) Ownership verification.

Figure 4: Defense effectiveness across seven datasets: (a) utility drop and (b) ownership verification.

*First, BackdoorWM offers the best protection–utility balance.* It reaches the highest median fidelity (80.07%) and perfect verification (100%) with a small utility drop (median 3.27 pp) and low cost, which indicates that a compact trigger aligned with the training objective yields a stable, easily verifiable pattern without harming clean predictions. *Second, ImperceptibleWM also achieves perfect verification but at a much higher computational cost.* The boxplots show a small median loss, while Table 2 reports long training time and higher memory because representation-level objectives add extra losses and buffers. *Third, RandomWM and SurviveWM reveal a stability verification trade-off.* RandomWM has moderate utility and verification with a wide spread across datasets, which suggests sensitivity to how random triggers align with the data distribution; SurviveWM has the smallest median loss (0.13 pp) but the weakest verification, which supports the view that increasing resilience to extraction can reduce detection strength when triggers are rare or weak. *Fourth, Integrity preserves utility and provides a lightweight ownership signal, but verification varies because the current metric is a single binary event.* It attains the highest median F1 (73.43%) with low time and memory, while the variation in verification follows from evaluating a one-bit fingerprint-flip indicator that depends on the dataset and the seed. Together these results show that backdoor triggers provide the strongest and most efficient protection in our setting, representation-level watermarks guarantee verification at a notable computational cost, methods based on random graphs or survival under extraction are sensitive to data properties and thus more variable, and integrity verification is practical when a binary ownership indicator suffices and strict label stability is required.

## 4.3 Protection–Utility Balance of Defenses (RQ3)

To answer RQ3, we evaluate the defended model's task utility and its alignment with the original target together with ownership verification on a fixed verification set under the unified protocol in Section 3. All defenses protect the same architecture and use identical splits; we compute metrics for every dataset and every seed. Figure 5 aggregates *all seven datasets and all three seeds* in a single view: each point is one dataset–seed run, the horizontal axis shows utility loss (pp) relative to the undefended target, and the vertical axis shows F1 (%). The cloud of points concentrates in the

Table 3: RQ4: efficiency of attacks at the $1.00\times$ budget on undefended targets (features and structure regime). Each cell reports *total attack time (min)* as mean $\pm$ standard deviation over three seeds.

| Attack | Cora | CiteSeer | CoauthorCS | CoauthorPhys | Computers | Photo | PubMed |
|---|---|---|---|---|---|---|---|
| MEA0 | $0.69 \pm 0.09$ | $\mathbf{0.72 \pm 0.08}$ | $0.86 \pm 0.10$ | $1.36 \pm 0.03$ | $2.18 \pm 1.84$ | $0.79 \pm 0.04$ | $1.21 \pm 0.09$ |
| MEA1 | $0.69 \pm 0.09$ | $0.73 \pm 0.08$ | $0.86 \pm 0.10$ | $1.34 \pm 0.07$ | $2.18 \pm 1.88$ | $\mathbf{0.76 \pm 0.06}$ | $1.17 \pm 0.01$ |
| MEA2 | $1.51 \pm 0.11$ | $1.73 \pm 0.22$ | $2.01 \pm 0.08$ | $2.25 \pm 0.11$ | $2.21 \pm 0.79$ | $1.59 \pm 0.18$ | $2.02 \pm 0.09$ |
| MEA3 | $\mathbf{0.66 \pm 0.08}$ | $0.74 \pm 0.08$ | $\mathbf{0.70 \pm 0.04}$ | $\mathbf{0.86 \pm 0.10}$ | $3.27 \pm 1.93$ | $1.14 \pm 0.09$ | $1.20 \pm 0.03$ |
| MEA4 | $0.76 \pm 0.09$ | $0.94 \pm 0.03$ | $2.58 \pm 0.09$ | $6.46 \pm 0.29$ | $2.50 \pm 1.82$ | $0.84 \pm 0.04$ | $1.73 \pm 0.07$ |
| MEA5 | $0.69 \pm 0.10$ | $0.75 \pm 0.07$ | $0.77 \pm 0.08$ | $0.92 \pm 0.18$ | $2.66 \pm 0.99$ | $1.25 \pm 0.20$ | $1.23 \pm 0.01$ |
| AdvMEA | $2.35 \pm 1.37$ | $4.39 \pm 2.52$ | $8.88 \pm 0.55$ | $4.93 \pm 0.05$ | $13.0 \pm 8.45$ | $10.3 \pm 11.4$ | $4.51 \pm 4.76$ |
| CEGA | $1.07 \pm 0.11$ | $1.03 \pm 0.07$ | $1.48 \pm 0.14$ | $2.15 \pm 0.05$ | $3.51 \pm 3.41$ | $1.07 \pm 0.10$ | $1.75 \pm 0.11$ |
| Realistic | $90.3 \pm 2.12$ | $111 \pm 3.72$ | $472 \pm 4.89$ | $976 \pm 7.34$ | $840 \pm 17.7$ | $248 \pm 0.14$ | $529 \pm 18.1$ |
| DFEA_I | $0.96 \pm 0.06$ | $1.11 \pm 0.04$ | $1.02 \pm 0.04$ | $0.97 \pm 0.08$ | $2.82 \pm 2.51$ | $1.00 \pm 0.09$ | $1.32 \pm 0.08$ |
| DFEA_II | $0.82 \pm 0.06$ | $0.88 \pm 0.09$ | $0.80 \pm 0.03$ | $\mathbf{0.86 \pm 0.09}$ | $\mathbf{1.28 \pm 0.57}$ | $0.87 \pm 0.09$ | $\mathbf{1.07 \pm 0.11}$ |
| DFEA_III | $1.48 \pm 0.14$ | $1.55 \pm 0.12$ | $1.46 \pm 0.16$ | $1.52 \pm 0.15$ | $3.30 \pm 2.47$ | $1.62 \pm 0.11$ | $1.86 \pm 0.15$ |

region of small loss (0–10 pp) and high F1 (65–80 %), which shows that several defenses preserve task performance while enabling ownership verification.

*First, BackdoorWM consistently lies near the top left*, which indicates that it preserves accuracy while enabling strong ownership verification; its spread is tight across datasets, which suggests stable behavior under our protocol. *Second, ImperceptibleWM occupies a similar region but with higher cost*, which matches the efficiency results in RQ4 and reflects the overhead of representation-level optimization. *Third, RandomWM, SurviveWM, and Integrity form broader clusters*, which shows that their balance depends more on data characteristics: RandomWM has moderate loss and verification with larger variance; SurviveWM has the smallest median loss but weak verification; Integrity preserves utility while yielding a binary ownership signal whose effectiveness varies across datasets. These observations imply that backdoor-style triggers provide the most reliable protection–utility balance in our setting, while representation-level watermarks trade efficiency for verification strength and the remaining methods are more sensitive to the data distribution.

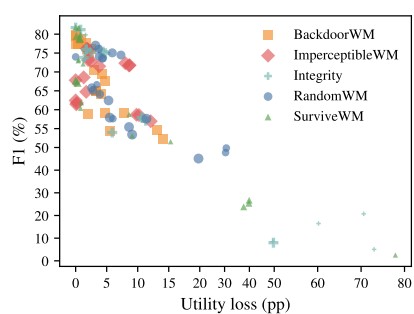

Figure 5: Protection–utility scatter across seven datasets and three seeds per defense. Each point is a dataset–seed run. The horizontal axis shows utility loss (pp) and the vertical axis shows F1 (%). Points farther left and higher indicate a better balance.

### 4.4 COMPUTATIONAL COST OF ATTACKS AND DEFENSES (RQ4)

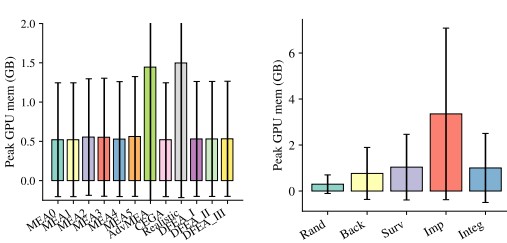

(a) Attacks: peak memory. (b) Defenses: peak memory.

Figure 6: Peak GPU memory (GB). Bars show mean and error bars show standard deviation.

To answer RQ4, we profile empirical efficiency on NVIDIA A100 hardware under a unified software stack. For attacks we fix the budget at $1.00\times$ in the features and structure regime and measure total attack time, query time, surrogate training time, minutes per fidelity point, and peak memory. For defenses we measure total defense time and peak GPU memory. All values are mean $\pm$ standard deviation over three seeds. Figure 6 reports peak memory for attacks and defenses, and Tables 3–4 list total time across seven datasets.

*First, most attacks complete within minutes and have similar memory footprints, while structure reconstruction is prohibitively expensive.* Across datasets the MEA family and CEGA finish in 0.7–2.5 minutes per run and use sub-GB memory, whereas the Realistic pipeline takes hundreds to thousands of minutes (Table 3) and uses more memory because it trains an auxiliary edge model and reconstructs structure (Figure 6). This gap is

Table 4: RQ4: efficiency of defenses across datasets. Each cell reports *total defense time (s)* as mean $\pm$ standard deviation over three seeds (lower is better).

| Defense | Cora | CiteSeer | CoauthorCS | CoauthorPhys | Computers | Photo | PubMed |
|---|---|---|---|---|---|---|---|
| RandomWM | $24.3 \pm 0.07$ | $23.9 \pm 0.09$ | $57.3 \pm 2.58$ | $34.8 \pm 0.77$ | $41.3 \pm 0.23$ | $36.0 \pm 0.30$ | $21.8 \pm 0.13$ |
| BackdoorWM | $1.88 \pm 0.01$ | $2.17 \pm 0.23$ | $2.54 \pm 0.01$ | $3.89 \pm 0.06$ | $1.98 \pm 0.00$ | $1.92 \pm 0.01$ | $1.88 \pm 0.02$ |
| SurviveWM | $1.59 \pm 0.00$ | $1.62 \pm 0.00$ | $2.75 \pm 0.01$ | $5.62 \pm 0.02$ | $2.31 \pm 0.04$ | $1.52 \pm 0.06$ | $2.27 \pm 0.09$ |
| Impercept. | $676 \pm 2.45$ | $709 \pm 2.33$ | $906 \pm 10.1$ | $950 \pm 14.0$ | $461 \pm 7.61$ | $209 \pm 11.0$ | $196 \pm 14.8$ |
| Integrity | $\mathbf{1.29 \pm 0.01}$ | $\mathbf{1.10 \pm 0.01}$ | $\mathbf{1.52 \pm 0.16}$ | $\mathbf{2.37 \pm 0.07}$ | $\mathbf{1.92 \pm 0.03}$ | $\mathbf{1.38 \pm 0.01}$ | $\mathbf{1.25 \pm 0.19}$ |

consistent with RQ1: the budget–metric curves show that fidelity improves rapidly at $0.05\times$–$0.25\times$ and then increases only slightly from $0.50\times$ to $1.00\times$. When the marginal gain per additional query becomes small, extra preprocessing such as structure reconstruction adds substantial time and memory but yields little benefit in extraction quality. *Second, adaptive or policy-based attacks incur higher and more variable time.* AdvMEA is slower and has larger variance across datasets, which matches the overhead of policy search and its non-monotone trajectories in RQ1. *Third, defenses separate into a lightweight group and a high-overhead group.* BackdoorWM, SurviveWM, and Integrity train in 1–6 seconds and maintain low memory, which reflects direct objectives without heavy auxiliary modules; ImperceptibleWM requires much longer time and more memory (Table 4, Figure 6) because representation-level watermarking adds extra losses and larger buffers. *In practice, the primary cost driver for attacks is not raw GPU memory but auxiliary modeling such as structure reconstruction or policy search, which rarely improves the cost–benefit once budgets surpass the small range; for defenses, backdoor triggers and query based integrity offer the most favorable time–memory profiles.*

## 5 RELATED WORK

**Existing Surveys And Benchmarks.** Prior work has surveyed intellectual-property protection for machine learning in vision and language, which includes watermarking, fingerprinting, and differential privacy, and it has summarized open problems in tracing stolen models (Zhao et al., 2025c; Li et al., 2025; Peng et al., 2023; Xue et al., 2021; Sun et al., 2023b; Zhao et al., 2025a;b). Parallel surveys for graph learning catalogue privacy, robustness, and fairness risks, and they review adversarial manipulation and information leakage on graphs (Dai et al., 2024b; Wu et al., 2022c; Zhang et al., 2022a; Wu et al., 2022b; Sun et al., 2024; Zheng et al., 2021). Recent testbeds standardize parts of graph security evaluation, yet they emphasize either robustness or privacy and do not evaluate model extraction together with ownership verification techniques such as watermarking and fingerprinting under a single protocol (Sun et al., 2023a; Zheng et al., 2021).

**Gaps in Technique Evaluation.** Existing studies propose random-query, adaptive, and data-free model-extraction attacks (Wu et al., 2021; Zhuang et al., 2024; Zhang et al., 2022b), information-limiting defenses that perturb outputs or filter queries (Liang et al., 2024; Kesarwani et al., 2018; Hu et al., 2024), and graph-specific ownership schemes for post-hoc verification (Wang et al., 2023; Xu et al., 2023; Dai et al., 2024a; You et al., 2024; Wu et al., 2024a; Waheed et al., 2024). However, results are reported on different datasets with incompatible threat assumptions and metrics, which prevents reliable comparison. We address this gap with, to our knowledge, the first unified benchmark that evaluates graph model extraction and ownership defenses under a single reproducible protocol. We fix public splits and shared query sets, standardize budgets and endpoint assumptions, separate an extraction track from an ownership-verification track, and report protection–utility trade-offs and computational cost with released reference implementations, which enables objective apples-to-apples comparison across methods, including data-free settings.

## 6 CONCLUSION

We presented *GraphIP-Bench*, the first benchmark and library that standardizes the evaluation of extraction attacks and ownership defenses for graph models under a single, reproducible protocol. The suite aligns datasets, query budgets, and endpoint assumptions and reports fidelity, utility, and efficiency for fair comparison, and our study across seven datasets maps the protection, utility and cost trade-offs that guide the design and deployment of trustworthy graph learning.

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

## A  IMPLEMENTATION DETAILS

We conduct all experiments on a single Linux node running Ubuntu 22.04.5 with NVIDIA driver 550.107.02 and CUDA runtime 12.4. Unless otherwise stated, every run uses a single NVIDIA A100 80 GB PCIe GPU. The system has 1.0 TiB of system memory. Our software stack consists of Python 3.11.13 in a dedicated Conda environment, PyTorch 2.3.0, DGL 2.5.0 + cu121, PyTorch Geometric 2.6.1, NumPy 2.3.3, SciPy 1.16.1, and NetworkX 3.3. We rely on prebuilt CUDA wheels and do not require the CUDA toolkit compiler. For each method we fix four disjoint splits per dataset (train, validation, test, query) and apply the same preprocessing pipeline and hyperparameter search protocol. We repeat all measurements with three random seeds and report the mean and the standard deviation. Wall–clock time is recorded with high–resolution timers in the training loop; peak GPU memory is obtained through the CUDA runtime and cross–checked with `nvidia_smi`. All experiments share identical hardware, software, and configuration defaults so that results are directly comparable across attacks, defenses, datasets, regimes, and budgets.

## B  DATASET STATISTICS

We summarize the graph datasets used in our experiments in Table 5. Edges are counted as undirected (unique). The average degree is computed as $2E/N$. For the Planetoid datasets (Cora, CiteSeer, PubMed) we follow the standard splits; for Amazon (Computers, Photo) and Coauthor (CoauthorCS, CoauthorPhysics) we use 100 training nodes per class with fixed validation and test sizes in our loader.

Table 5: Dataset statistics. Edges are undirected and unique. Avg. degree is $2E/N$.

| Dataset | # Nodes | # Edges | Avg. degree | # Classes | Node Text | Domain |
|---|---|---|---|---|---|---|
| Cora | 2,708 | 5,429 | 4.01 | 7 | Paper content | Citation |
| CiteSeer | 3,186 | 4,277 | 2.68 | 6 | Paper content | Citation |
| PubMed | 19,717 | 44,338 | 4.50 | 3 | Paper content | Citation |
| Computers | 13,752 | 245,861 | 35.75 | 10 | Entity description | Web link |
| Photo | 7,650 | 119,081 | 31.14 | 8 | Entity description | Web link |
| CoauthorCS | 18,333 | 163,788 | 17.87 | 15 | Paper content | Citation |
| CoauthorPhysics | 34,493 | 247,962 | 14.38 | 5 | Paper content | Citation |

## C  REPRODUCIBILITY AND CONFIGURATIONS

**Scope and averaging.** We release scripts, fixed random seeds, and per-method configurations to reproduce all tables and figures. Results are averaged over three seeds (0, 1, 2) and reported as mean ± standard deviation unless stated otherwise.

**Seeds and determinism.** For each run we set the Python and CUDA random states and propagate the seed through data loading and sampling. GPU runs use deterministic kernels when available.

**Device selection.** A command-line flag *–device* selects the process-visible GPU (via the environment) or the CPU; all modules use the same device setting throughout the run.

**Dataset loading and splits.** A unified loader normalizes dataset aliases and returns DGL graphs with node features and masks. Planetoid datasets (Cora, CiteSeer, PubMed) use the standard train/validation/test masks. Amazon and Coauthor datasets (Computers, Photo, CoauthorCS, CoauthorPhysics) use a canonical per-class sampling scheme with 100 training nodes per class and fixed validation/test sizes.

**Directory layout and logs.** *Attack track (RQ1)* writes newline-delimited JSON files under *outputs/RQ1_final/<Dataset>/<Dataset>.jsonl*. Each record contains header fields (track, dataset, attack, configuration index, constructor/run configuration, budget multiplier, node fraction induced by the budget, regime, feature/adjacency ratios, seed), performance metrics (accuracy, F1, precision, recall, fidelity), and compute metrics (train target time, query time, surrogate training time, total attack time, per-query inference latency for target and surrogate, peak GPU memory, GPU hours). *Ownership/defense track (RQ2/RQ3)* writes *outputs/RQ2_RQ3_best/<Dataset>.jsonl* with fields (track, dataset, defense, configuration index, configuration, seed), performance metrics (ac-

Table 6: RQ1 overview: regimes × metrics (%). Means across datasets and the five budgets (0.05–1.00).

| Attack | both | | | x_only | | | a_only | | | data_free | | |
|---|---|---|---|---|---|---|---|---|---|---|---|---|
| | Acc | F1 | Fidelity | Acc | F1 | Fidelity | Acc | F1 | Fidelity | Acc | F1 | Fidelity |
| MEA0 | 78.07 | 64.77 | 83.75 | 78.97 | 64.72 | 84.72 | 79.35 | 65.97 | 85.24 | 20.23 | 5.72 | 19.58 |
| MEA1 | 57.08 | 43.79 | 60.33 | 57.07 | 43.96 | 60.34 | 57.08 | 43.80 | 60.32 | 21.06 | 5.68 | 18.98 |
| MEA2 | 37.52 | 22.21 | 38.01 | 35.66 | 21.91 | 36.96 | 35.78 | 21.81 | 36.75 | 18.54 | 6.82 | 17.85 |
| MEA3 | 78.74 | 65.31 | 83.26 | 79.19 | 65.28 | 83.35 | 78.65 | 64.29 | 83.16 | 22.88 | 6.59 | 21.56 |
| MEA4 | 73.71 | 54.80 | 78.54 | 71.38 | 53.02 | 76.36 | 72.08 | 53.58 | 76.71 | 16.69 | 5.01 | 16.23 |
| MEA5 | 79.34 | 65.77 | 83.92 | 79.64 | 65.72 | 84.31 | 79.30 | 65.67 | 84.03 | 21.20 | 6.34 | 20.19 |
| AdvMEA | 62.16 | 53.63 | 63.77 | 62.92 | 53.97 | 64.21 | 62.08 | 53.35 | 63.49 | 62.94 | 54.37 | 64.37 |
| CEGA | 75.01 | 64.61 | 83.08 | 74.94 | 64.64 | 82.68 | 75.58 | 64.71 | 83.80 | 74.02 | 63.83 | 82.08 |
| Realistic | 59.20 | 47.72 | 76.70 | 59.76 | 47.30 | 77.40 | 59.03 | 46.86 | 76.59 | 58.71 | 47.25 | 77.16 |
| DFEA_I | 45.55 | 29.13 | 46.27 | 45.59 | 29.10 | 46.09 | 46.17 | 29.50 | 46.76 | 35.06 | 20.91 | 35.61 |
| DFEA_II | 30.60 | 16.49 | 29.50 | 31.18 | 16.59 | 30.23 | 32.02 | 16.45 | 30.90 | 24.80 | 11.98 | 23.45 |
| DFEA_III | 34.29 | 18.11 | 33.47 | 36.06 | 19.68 | 35.61 | 33.30 | 17.61 | 32.43 | 27.04 | 12.99 | 26.27 |

curacy, F1, precision, recall, watermark accuracy), and compute metrics (train target time, defense training time, defense inference time, total defense time, peak GPU memory, GPU hours). Leaderboards and LaTeX tables are exported with selection and formatting utilities.

**RQ1 (attacks) configuration.** The attack runner sweeps query budgets $\{0.05, 0.10, 0.25, 0.50, 1.00\}$ and four regimes (features only, structure only, both available, data free). Per-method training schedules are recorded in logs. For the MEA family, CEGA, and Realistic we use 200 epochs per cycle; CEGA additionally uses learning rate 0.01 and 200 target/surrogate epochs. The AdvMEA implementation uses its internal fixed epoch schedule; any external epoch parameter appears in logs for uniformity but does not affect training.

**RQ2/RQ3 (defenses) configuration.** We provide a best-configuration runner that replays the top settings discovered by a prior grid search. Table **??** lists the fixed constructor parameters used per dataset and defense; other runtime arguments remain at method defaults.

**Common search spaces.** *Attacks:* budget, regime, and per-method cycles/epochs (when applicable). Learning rate and dropout follow method defaults unless a method requires explicit settings (CEGA uses learning rate 0.01). *Defenses:* grid search over typical ranges; a single best configuration per dataset/defense is fixed as in Table **??**.

**How to re-run.** *RQ1 (attacks):* call the attack runner with dataset, attack, budget, regime, and seed. Logs are saved under *outputs/RQ1_final* with the exact budget multiplier recorded in each line. *RQ2/RQ3 (defenses):* call the best-configuration runner with seeds $[0, 1, 2]$; outputs are saved under *outputs/RQ2_RQ3_best*. This setup unifies scripts, seeds, and logging across datasets and methods, enabling direct regeneration of all tables from the released outputs.

# D SUPPLEMENTARY EXPERIMENTAL RESULTS AND DISCUSSION

## D.1 ATTACK EFFECTIVENESS RESULTS

This appendix reports the full results for attacks across seven datasets. We first present one compact overview tables which summarize, for each metric (Accuracy, F1, and Fidelity), the mean performance of twelve attacks across four query regimes (with an additional Overall column). Each number is averaged over the seven datasets and five budget levels defined in the main text. These tables allow the reader to identify, at a glance, which attack performs best under each regime. We then provide the complete per-dataset matrices. For each dataset and each metric, we show a $2 \times 2$ panel that contains four sub-tables (one per regime). Each sub-table is a $12 \times 5$ matrix whose rows are the attacks and whose columns are the five query budgets. Bold font marks the best score in each budget column. All splits, budgets, and aggregation rules match the protocol in the main paper.

## D.2 DEFENSE PERFORMANCE RESULTS

We report per–dataset defense performance in three complementary tables. Table 28 presents task utility measured by macro F1 (%), averaged across three seeds. Table 29 reports behavioral align-

Table 7: RQ1 detailed for dataset=Cora, metric=Acc (%). Rows are attacks; columns are budgets. Mean ± std across seeds; best per column is bold.

(a) Regime=both

| Attack | 0.05 | 0.10 | 0.25 | 0.50 | 1.00 |
|---|---|---|---|---|---|
| MEA0 | 67.4±4.4 | 75.4±1.6 | 79.7±0.7 | 80.8±0.2 | 81.7±0.3 |
| MEA1 | 52.7±0.5 | 63.5±2.0 | **81.5±0.7** | 80.8±0.1 | **82.3±0.2** |
| MEA2 | 21.5±9.2 | 22.5±6.6 | 33.3±6.2 | 29.7±5.6 | 34.3±10.5 |
| MEA3 | 67.1±3.4 | 74.3±1.9 | 79.6±1.5 | 81.2±0.6 | 81.1±0.3 |
| MEA4 | 45.4±6.1 | 65.7±1.9 | 77.3±0.9 | 78.5±0.2 | 80.5±0.0 |
| MEA5 | **72.1±2.9** | 75.4±0.8 | 79.9±0.7 | **81.4±0.6** | 81.2±0.3 |
| AdvMEA | 64.8±4.2 | 66.8±2.2 | 65.1±1.4 | 67.7±4.7 | 66.7±2.3 |
| CEGA | 71.0±5.2 | **76.5±0.9** | 79.7±0.2 | 79.3±0.6 | 79.3±0.3 |
| Realistic | 55.4±8.2 | 67.1±2.8 | 71.0±4.4 | 74.8±1.8 | 75.9±2.4 |
| DFEA_I | 44.9±2.3 | 59.2±3.7 | 47.2±3.2 | 32.3±0.4 | 31.9±0.0 |
| DFEA_II | 32.7±7.1 | 40.9±2.4 | 35.4±4.4 | 32.3±0.6 | 31.9±0.0 |
| DFEA_III | 38.9±1.8 | 42.1±4.1 | 41.0±3.6 | 31.9±0.0 | 31.9±0.0 |

(b) Regime=x_only

| Attack | 0.05 | 0.10 | 0.25 | 0.50 | 1.00 |
|---|---|---|---|---|---|
| MEA0 | 68.7±4.2 | 75.1±0.7 | 80.1±0.2 | 80.4±1.3 | 81.2±0.5 |
| MEA1 | 52.7±0.5 | 63.5±2.0 | **81.5±0.7** | 80.8±0.1 | **82.3±0.2** |
| MEA2 | 13.7±4.7 | 23.6±4.4 | 39.1±9.5 | 35.0±9.5 | 34.0±9.5 |
| MEA3 | 64.7±1.7 | 74.0±0.4 | 79.7±0.4 | 81.4±0.4 | 81.1±0.1 |
| MEA4 | 42.9±8.4 | 63.7±3.7 | 76.8±0.4 | 78.5±0.2 | 80.5±0.0 |
| MEA5 | 69.9±3.4 | 77.0±1.7 | 80.5±0.3 | **81.5±0.5** | 81.2±0.1 |
| AdvMEA | 65.5±3.3 | 68.8±2.5 | 65.4±2.0 | 63.6±0.8 | 65.0±3.0 |
| CEGA | **74.1±0.5** | **77.6±0.4** | 77.7±1.2 | 78.7±1.1 | 79.4±0.3 |
| Realistic | 55.7±8.3 | 64.2±5.0 | 71.0±4.3 | 74.8±3.4 | 75.5±2.7 |
| DFEA_I | 41.4±10.9 | 50.4±0.7 | 44.4±2.0 | 32.1±0.3 | 31.9±0.0 |
| DFEA_II | 33.1±4.5 | 41.3±2.2 | 33.2±0.9 | 32.5±0.3 | 31.9±0.0 |
| DFEA_III | 39.8±6.1 | 50.9±5.2 | 37.2±6.6 | 31.9±0.0 | 31.9±0.0 |

(c) Regime=a_only

| Attack | 0.05 | 0.10 | 0.25 | 0.50 | 1.00 |
|---|---|---|---|---|---|
| MEA0 | 69.4±0.3 | 75.6±0.3 | 78.7±1.6 | 80.1±1.3 | 80.8±0.6 |
| MEA1 | 52.7±0.5 | 63.5±2.0 | **81.5±0.7** | 80.8±0.1 | **82.3±0.2** |
| MEA2 | 23.4±8.3 | 21.1±6.4 | 37.9±10.0 | 42.0±1.7 | 32.6±9.2 |
| MEA3 | 63.0±1.3 | 75.2±1.8 | 79.7±0.8 | 81.0±0.6 | 81.3±0.2 |
| MEA4 | 45.9±5.8 | 63.4±1.4 | 77.4±0.6 | 78.5±0.2 | 80.5±0.0 |
| MEA5 | 65.6±6.9 | 76.5±1.5 | 79.6±1.1 | **81.5±0.5** | 81.3±0.1 |
| AdvMEA | 65.4±2.7 | 64.0±2.3 | 65.8±1.7 | 68.1±1.9 | 60.5±5.7 |
| CEGA | **69.6±3.6** | **78.0±0.4** | 78.0±0.8 | 79.6±0.5 | 79.2±0.6 |
| Realistic | 51.5±4.3 | 64.5±6.0 | 72.0±2.3 | 75.3±2.5 | 76.3±2.7 |
| DFEA_I | 51.2±7.2 | 55.4±5.1 | 53.1±5.3 | 32.7±1.1 | 31.9±0.0 |
| DFEA_II | 35.8±1.3 | 34.8±4.8 | 35.9±1.3 | 32.5±0.8 | 31.9±0.0 |
| DFEA_III | 40.0±1.5 | 48.2±4.4 | 34.3±3.3 | 31.9±0.0 | 31.9±0.0 |

(d) Regime=data_free

| Attack | 0.05 | 0.10 | 0.25 | 0.50 | 1.00 |
|---|---|---|---|---|---|
| MEA0 | 10.9±2.5 | 14.6±0.2 | 19.3±8.9 | 20.2±8.2 | 19.0±9.3 |
| MEA1 | 14.4±0.0 | 14.4±0.0 | 14.4±0.0 | 14.4±0.0 | 14.4±0.0 |
| MEA2 | 13.7±1.3 | 13.3±1.9 | 13.3±1.9 | 13.7±1.3 | 13.7±1.3 |
| MEA3 | 17.6±10.6 | 12.3±2.4 | 17.1±10.8 | 13.5±0.7 | 26.2±8.0 |
| MEA4 | 17.1±10.8 | 19.9±8.5 | 18.5±9.7 | 10.9±2.5 | 18.0±10.0 |
| MEA5 | 24.6±13.0 | 11.9±2.7 | 21.1±7.7 | 20.4±8.1 | 18.7±9.5 |
| AdvMEA | 68.6±2.4 | 68.5±1.5 | 65.1±6.3 | 65.8±2.9 | 67.9±3.7 |
| CEGA | **71.0±1.8** | **77.0±0.7** | **78.8±1.9** | **79.2±0.9** | **79.2±0.1** |
| Realistic | 67.9±2.3 | 70.8±1.6 | 69.0±3.6 | 66.5±2.6 | 69.1±1.6 |
| DFEA_I | 54.1±2.5 | 53.9±6.3 | 51.5±1.4 | 52.1±8.2 | 50.1±6.4 |
| DFEA_II | 37.5±2.9 | 39.9±1.3 | 45.2±2.8 | 39.6±2.1 | 36.2±0.6 |
| DFEA_III | 43.5±5.8 | 47.8±1.7 | 42.4±7.5 | 41.5±3.7 | 38.7±4.7 |

Table 8: RQ1 detailed for dataset=Cora, metric=F1 (%). Rows are attacks; columns are budgets. Mean ± std across seeds; best per column is bold.

(a) Regime=both

| Attack | 0.05 | 0.10 | 0.25 | 0.50 | 1.00 |
|---|---|---|---|---|---|
| MEA0 | 61.0±6.6 | 73.9±1.5 | 77.9±1.2 | 79.6±0.4 | 81.0±0.1 |
| MEA1 | 50.5±0.5 | 58.2±4.1 | **80.3±0.7** | 80.2±0.2 | **81.8±0.0** |
| MEA2 | 12.3±1.9 | 14.1±4.3 | 30.1±4.5 | 24.7±4.6 | 20.8±7.8 |
| MEA3 | 60.6±5.9 | 70.4±2.8 | 78.4±1.6 | 79.5±0.9 | 80.2±0.2 |
| MEA4 | 44.3±6.5 | 65.8±1.2 | 76.1±0.6 | 77.3±0.0 | 79.2±0.3 |
| MEA5 | 67.9±4.1 | 70.3±3.8 | 78.8±0.8 | **80.3±0.6** | 80.2±0.4 |
| AdvMEA | 65.2±2.9 | 67.3±2.3 | 66.6±0.7 | 67.7±3.9 | 67.3±2.0 |
| CEGA | **69.9±4.4** | **75.6±0.9** | 79.1±0.1 | 78.6±0.3 | 78.4±0.2 |
| Realistic | 53.1±6.4 | 61.1±4.6 | 69.3±3.8 | 73.9±1.2 | 75.0±2.0 |
| DFEA_I | 34.6±6.2 | 50.9±4.9 | 27.9±5.3 | 7.6±0.9 | 6.9±0.0 |
| DFEA_II | 15.6±5.8 | 25.5±3.9 | 13.0±6.9 | 7.7±1.2 | 7.0±0.1 |
| DFEA_III | 23.8±6.3 | 22.8±4.4 | 18.7±4.8 | 6.9±0.0 | 6.9±0.0 |

(b) Regime=x_only

| Attack | 0.05 | 0.10 | 0.25 | 0.50 | 1.00 |
|---|---|---|---|---|---|
| MEA0 | 63.2±4.5 | 68.7±0.4 | 78.8±0.2 | 79.0±1.8 | 80.4±0.6 |
| MEA1 | 50.5±0.5 | 58.2±4.1 | **80.3±0.7** | 80.2±0.2 | **81.8±0.0** |
| MEA2 | 6.9±1.9 | 16.5±1.6 | 35.7±7.2 | 33.6±8.4 | 20.8±6.6 |
| MEA3 | 58.0±2.6 | 70.0±1.8 | 78.5±0.5 | **80.5±0.3** | 80.2±0.1 |
| MEA4 | 27.9±6.9 | 62.2±4.5 | 75.8±0.5 | 77.3±0.0 | 79.2±0.3 |
| MEA5 | 66.5±1.3 | 74.2±2.4 | 79.2±0.5 | 80.3±0.4 | 80.3±0.3 |
| AdvMEA | 66.3±2.8 | 67.0±3.0 | 65.3±3.1 | 64.0±0.5 | 66.4±2.2 |
| CEGA | **73.6±1.1** | **77.2±0.4** | 77.4±0.7 | 78.1±0.7 | 78.6±0.2 |
| Realistic | 50.9±5.8 | 58.8±4.9 | 68.2±6.1 | 74.0±3.1 | 75.2±2.6 |
| DFEA_I | 33.3±7.3 | 38.2±0.4 | 23.9±5.1 | 7.4±0.7 | 6.9±0.0 |
| DFEA_II | 18.1±4.8 | 26.4±2.0 | 12.2±3.1 | 8.4±0.8 | 6.9±0.0 |
| DFEA_III | 22.9±10.3 | 39.7±9.1 | 14.2±7.2 | 6.9±0.0 | 6.9±0.0 |

(c) Regime=a_only

| Attack | 0.05 | 0.10 | 0.25 | 0.50 | 1.00 |
|---|---|---|---|---|---|
| MEA0 | 64.3±0.6 | 72.9±0.8 | 77.5±1.9 | 78.7±1.8 | 80.0±0.7 |
| MEA1 | 50.4±0.5 | 58.2±4.1 | **80.3±0.7** | 80.2±0.2 | **81.8±0.0** |
| MEA2 | 13.8±2.5 | 13.2±3.6 | 35.0±6.6 | 36.4±2.4 | 21.2±8.6 |
| MEA3 | 55.7±3.0 | 70.3±3.0 | 78.3±1.4 | 80.0±0.3 | 80.4±0.2 |
| MEA4 | 34.6±8.6 | 60.4±4.2 | 76.4±0.2 | 77.3±0.0 | 79.2±0.3 |
| MEA5 | 58.8±11.2 | 73.7±2.4 | 77.7±1.0 | **80.7±0.4** | 80.2±0.2 |
| AdvMEA | 67.0±2.0 | 64.7±3.1 | 65.8±1.1 | 67.0±1.4 | 62.6±3.9 |
| CEGA | **68.7±1.4** | **77.5±0.6** | 77.4±0.9 | 79.0±0.8 | 78.4±0.2 |
| Realistic | 46.3±5.0 | 62.2±8.0 | 70.8±1.2 | 74.6±2.2 | 76.0±2.0 |
| DFEA_I | 48.9±3.6 | 47.4±6.6 | 37.5±9.6 | 8.4±2.2 | 6.9±0.0 |
| DFEA_II | 17.0±2.4 | 19.0±3.7 | 14.4±2.2 | 8.0±1.4 | 6.9±0.0 |
| DFEA_III | 25.8±3.0 | 33.7±6.6 | 9.5±3.7 | 6.9±0.0 | 6.9±0.0 |

(d) Regime=data_free

| Attack | 0.05 | 0.10 | 0.25 | 0.50 | 1.00 |
|---|---|---|---|---|---|
| MEA0 | 2.8±0.6 | 3.6±0.1 | 4.5±1.7 | 4.7±1.6 | 4.4±1.8 |
| MEA1 | 3.6±0.0 | 3.6±0.0 | 3.6±0.0 | 3.6±0.0 | 3.6±0.0 |
| MEA2 | 4.3±1.0 | 4.1±0.7 | 4.1±0.7 | 4.1±1.0 | 4.3±1.0 |
| MEA3 | 4.1±2.1 | 3.1±0.6 | 4.0±2.2 | 3.4±0.1 | 5.8±1.5 |
| MEA4 | 4.0±2.2 | 4.6±1.6 | 4.3±1.9 | 2.8±0.6 | 4.2±2.0 |
| MEA5 | 7.4±4.8 | 3.6±0.4 | 6.1±1.7 | 4.7±1.5 | 4.5±1.7 |
| AdvMEA | 68.4±0.6 | 68.4±1.1 | 64.7±6.1 | 66.3±2.2 | 68.0±2.7 |
| CEGA | **69.5±1.5** | **76.3±0.5** | **78.4±1.6** | **78.5±0.9** | **78.7±0.2** |
| Realistic | 63.1±3.3 | 67.8±3.2 | 67.1±3.7 | 63.0±4.0 | 67.2±1.8 |
| DFEA_I | 41.3±6.0 | 42.1±11.8 | 43.7±1.3 | 40.6±11.9 | 36.6±12.1 |
| DFEA_II | 17.7±4.6 | 21.2±3.2 | 34.1±5.0 | 22.9±4.1 | 16.3±1.9 |
| DFEA_III | 26.1±6.8 | 31.9±4.9 | 22.7±12.1 | 27.8±6.5 | 20.6±9.4 |

ment with the original target, measured by fidelity (%) on the same test inputs. Table 30 summarizes ownership verification on a standardized verification set, where values reflect verification accuracy for watermarking methods (and the integrity checker) under our evaluation protocol. Taken together, these views separate downstream utility, behavioral consistency, and ownership verification so that the trade–offs across defenses are explicit on every dataset.

Table 9: RQ1 detailed for dataset=Cora, metric=Fidelity (%). Rows are attacks; columns are budgets. Mean ± std across seeds; best per column is bold.

(a) Regime=both

| Attack | 0.05 | 0.10 | 0.25 | 0.50 | 1.00 |
|---|---|---|---|---|---|
| MEA0 | 69.3±3.1 | 82.6±1.6 | 87.5±0.9 | 89.7±0.7 | 92.4±0.6 |
| MEA1 | 57.3±0.2 | 70.6±2.0 | 89.9±0.9 | 90.4±0.6 | 91.3±0.4 |
| MEA2 | 21.1±7.0 | 24.7±7.4 | 36.4±6.5 | 31.9±6.1 | 36.4±10.9 |
| MEA3 | 71.2±3.6 | 79.3±2.7 | 88.0±0.8 | 90.1±1.4 | 92.7±0.9 |
| MEA4 | 49.2±6.2 | 71.9±1.7 | 85.0±1.3 | 87.6±0.6 | 89.5±0.2 |
| MEA5 | 75.8±4.8 | 82.1±4.5 | 88.3±0.6 | **92.8±1.1** | 92.5±0.7 |
| AdvMEA | 67.6±4.0 | 68.8±2.8 | 67.4±1.6 | 69.0±4.9 | 68.2±4.0 |
| CEGA | **78.4±4.4** | **86.5±1.0** | **91.5±0.4** | 92.5±0.4 | **93.7±0.5** |
| Realistic | 62.0±4.5 | 68.1±1.8 | 78.6±0.7 | 83.7±0.6 | 88.5±1.7 |
| DFEA_I | 47.6±2.9 | 59.2±4.5 | 45.0±3.5 | 26.5±0.5 | 26.2±0.0 |
| DFEA_II | 30.5±5.3 | 38.8±2.7 | 31.8±7.1 | 26.5±0.5 | 26.2±0.1 |
| DFEA_III | 38.5±5.0 | 40.6±3.0 | 37.9±4.3 | 26.2±0.0 | 26.2±0.0 |

(b) Regime=x_only

| Attack | 0.05 | 0.10 | 0.25 | 0.50 | 1.00 |
|---|---|---|---|---|---|
| MEA0 | 72.2±3.9 | 79.5±0.9 | 88.2±1.2 | 90.9±1.2 | 92.1±0.8 |
| MEA1 | 57.3±0.2 | 70.6±2.0 | 89.9±0.9 | 90.4±0.5 | 91.3±0.4 |
| MEA2 | 15.0±4.9 | 26.4±5.0 | 42.5±10.5 | 37.9±11.4 | 35.8±10.0 |
| MEA3 | 69.1±0.8 | 77.7±0.5 | 88.0±0.9 | 90.9±1.1 | 92.5±1.3 |
| MEA4 | 45.3±8.6 | 69.2±3.7 | 84.6±0.7 | 87.6±0.6 | 89.6±0.2 |
| MEA5 | 73.9±1.8 | 83.2±0.3 | 88.8±0.8 | 91.9±0.5 | 92.3±1.1 |
| AdvMEA | 66.7±2.0 | 71.2±3.0 | 66.5±1.5 | 67.1±1.5 | 67.4±3.1 |
| CEGA | **82.5±1.7** | **86.8±0.8** | **90.8±1.1** | **92.8±0.8** | **93.7±0.1** |
| Realistic | 62.9±4.6 | 69.8±1.7 | 79.4±2.4 | 84.1±0.9 | 87.9±0.2 |
| DFEA_I | 42.9±10.0 | 50.1±2.9 | 41.4±3.3 | 26.5±0.4 | 26.2±0.0 |
| DFEA_II | 30.4±3.7 | 39.6±2.2 | 29.9±2.2 | 26.9±0.4 | 26.2±0.0 |
| DFEA_III | 38.6±10.6 | 51.7±7.4 | 33.8±8.8 | 26.2±0.0 | 26.2±0.0 |

(c) Regime=a_only

| Attack | 0.05 | 0.10 | 0.25 | 0.50 | 1.00 |
|---|---|---|---|---|---|
| MEA0 | 74.1±0.8 | 81.9±1.4 | 87.2±1.1 | 89.5±1.3 | 92.4±1.2 |
| MEA1 | 57.2±0.2 | 70.6±2.0 | 89.9±0.9 | 90.4±0.6 | 91.3±0.4 |
| MEA2 | 23.8±7.1 | 22.6±7.2 | 41.1±11.0 | 45.7±2.2 | 34.3±10.0 |
| MEA3 | 64.3±0.6 | 80.9±1.2 | 88.1±0.3 | 92.0±0.5 | 92.7±1.1 |
| MEA4 | 48.3±4.1 | 68.2±1.8 | 85.6±0.5 | 87.6±0.6 | 89.5±0.2 |
| MEA5 | 68.8±8.4 | 82.1±1.6 | 88.0±0.9 | 92.2±0.7 | 92.7±1.4 |
| AdvMEA | 67.5±3.0 | 66.1±2.3 | 67.4±3.6 | 68.9±2.5 | 62.7±4.3 |
| CEGA | **78.2±3.8** | **88.4±0.9** | **91.0±0.7** | **92.3±0.3** | **93.7±0.3** |
| Realistic | 56.5±4.1 | 69.3±3.9 | 79.8±0.3 | 84.3±1.9 | 89.1±0.9 |
| DFEA_I | 55.2±6.2 | 57.0±5.8 | 52.4±7.4 | 27.0±1.2 | 26.2±0.0 |
| DFEA_II | 32.0±2.2 | 34.1±3.1 | 30.8±1.8 | 26.8±0.8 | 26.2±0.0 |
| DFEA_III | 39.8±2.0 | 48.4±5.3 | 29.1±4.1 | 26.2±0.0 | 26.2±0.0 |

(d) Regime=data_free

| Attack | 0.05 | 0.10 | 0.25 | 0.50 | 1.00 |
|---|---|---|---|---|---|
| MEA0 | 12.3±2.0 | 15.9±0.8 | 17.3±6.2 | 19.0±5.0 | 18.0±6.3 |
| MEA1 | 15.3±0.2 | 15.3±0.2 | 15.3±0.2 | 15.3±0.1 | 15.3±0.2 |
| MEA2 | 15.1±1.7 | 14.5±2.4 | 14.5±2.4 | 15.1±1.7 | 15.1±1.7 |
| MEA3 | 16.2±7.9 | 13.3±2.5 | 15.3±8.0 | 13.8±1.0 | 23.1±4.3 |
| MEA4 | 15.5±8.0 | 18.7±5.5 | 17.3±6.6 | 12.3±2.0 | 16.8±6.7 |
| MEA5 | 21.9±10.9 | 12.5±2.6 | 20.2±4.3 | 19.5±4.8 | 17.2±6.6 |
| AdvMEA | 69.4±2.7 | 70.1±0.9 | 67.4±5.8 | 67.4±2.1 | 70.5±3.5 |
| CEGA | **79.8±2.7** | **88.0±0.4** | **91.0±1.5** | **92.5±0.0** | **93.5±0.6** |
| Realistic | 73.0±3.5 | 72.9±1.6 | 74.3±1.4 | 72.9±1.8 | 73.1±5.0 |
| DFEA_I | 56.1±3.4 | 55.6±8.9 | 54.5±1.2 | 54.3±8.5 | 52.4±8.8 |
| DFEA_II | 34.0±4.0 | 35.3±1.8 | 44.4±3.5 | 39.4±3.4 | 32.5±2.1 |
| DFEA_III | 42.5±9.3 | 45.5±1.9 | 40.3±10.4 | 41.3±5.2 | 36.3±6.7 |

Table 10: RQ1 detailed for dataset=CiteSeer, metric=Acc (%). Rows are attacks; columns are budgets. Mean ± std across seeds; best per column is bold.

(a) Regime=both

| Attack | 0.05 | 0.10 | 0.25 | 0.50 | 1.00 |
|---|---|---|---|---|---|
| MEA0 | 55.4±6.9 | 62.4±2.4 | 69.0±0.7 | **70.1±0.9** | 70.1±0.6 |
| MEA1 | 41.6±1.4 | 41.8±2.6 | 54.0±7.9 | 66.3±1.4 | 69.0±0.6 |
| MEA2 | 14.7±4.9 | 16.0±6.1 | 25.5±5.5 | 28.5±2.4 | 24.4±7.1 |
| MEA3 | 50.7±2.8 | 59.7±3.3 | 69.4±1.0 | 69.6±0.0 | 70.2±0.4 |
| MEA4 | 44.9±1.0 | 57.9±3.7 | 67.1±0.2 | 69.9±0.4 | **70.6±0.4** |
| MEA5 | 49.0±2.1 | 62.8±1.8 | **69.5±1.1** | 69.8±0.2 | 70.4±0.8 |
| AdvMEA | 51.8±8.1 | 46.3±4.2 | 53.2±0.7 | 52.3±1.7 | 51.3±5.1 |
| CEGA | **59.9±3.2** | **64.6±2.1** | 66.5±1.1 | 67.0±1.5 | 67.2±0.3 |
| Realistic | 47.2±2.0 | 57.7±4.2 | 64.3±2.3 | 64.5±0.4 | 65.2±1.1 |
| DFEA_I | 39.2±3.7 | 48.7±1.5 | 47.7±2.6 | 32.0±9.0 | 32.6±2.8 |
| DFEA_II | 27.7±2.4 | 29.6±4.4 | 32.5±7.6 | 29.5±7.1 | 26.1±2.3 |
| DFEA_III | 37.4±2.6 | 41.6±10.7 | 31.3±5.4 | 29.4±9.3 | 19.6±2.1 |

(b) Regime=x_only

| Attack | 0.05 | 0.10 | 0.25 | 0.50 | 1.00 |
|---|---|---|---|---|---|
| MEA0 | 54.1±3.3 | 63.2±2.4 | 69.8±1.2 | 70.0±0.9 | 69.6±0.2 |
| MEA1 | 41.6±1.4 | 41.8±2.6 | 54.0±7.9 | 66.3±1.4 | 68.9±0.6 |
| MEA2 | 14.1±3.1 | 15.1±5.1 | 19.0±8.0 | 31.7±2.2 | 23.9±5.0 |
| MEA3 | 57.9±4.1 | 64.6±1.8 | 68.7±1.0 | 69.9±0.8 | 69.9±0.7 |
| MEA4 | 39.9±4.7 | 59.0±2.3 | 66.6±0.5 | 69.9±0.4 | **70.6±0.4** |
| MEA5 | 55.3±1.9 | **65.9±0.7** | **70.3±0.6** | **70.3±0.7** | 70.0±0.5 |
| AdvMEA | 48.9±0.9 | 51.4±2.4 | 53.9±4.3 | 48.2±6.8 | 45.9±2.2 |
| CEGA | **61.3±1.9** | 65.2±1.1 | 66.7±0.6 | 67.3±0.6 | 67.8±0.5 |
| Realistic | 48.1±3.5 | 53.2±4.3 | 61.5±1.9 | 62.7±2.0 | 65.4±1.4 |
| DFEA_I | 34.6±4.5 | 51.3±1.8 | 49.9±4.4 | 41.9±8.6 | 28.3±0.4 |
| DFEA_II | 34.2±4.6 | 38.2±5.7 | 30.3±5.4 | 28.2±5.7 | 24.9±1.7 |
| DFEA_III | 40.1±2.7 | 40.7±11.1 | 35.8±4.7 | 28.0±7.2 | 20.8±3.0 |

(c) Regime=a_only

| Attack | 0.05 | 0.10 | 0.25 | 0.50 | 1.00 |
|---|---|---|---|---|---|
| MEA0 | 52.5±3.1 | 64.5±1.5 | **70.2±0.9** | 69.3±0.8 | 70.1±1.1 |
| MEA1 | 41.6±1.4 | 41.8±2.6 | 54.0±7.9 | 66.3±1.4 | 69.0±0.6 |
| MEA2 | 15.1±4.5 | 15.4±4.1 | 18.9±9.1 | 24.4±4.0 | 24.5±6.4 |
| MEA3 | 48.1±5.0 | 62.3±3.3 | 68.4±1.5 | **70.1±0.5** | 70.0±0.5 |
| MEA4 | 43.9±1.3 | 56.8±1.3 | 66.2±0.6 | 69.9±0.4 | 70.6±0.4 |
| MEA5 | 52.7±3.1 | 63.5±1.9 | 69.9±0.6 | 69.7±0.4 | **70.8±1.1** |
| AdvMEA | 47.5±4.0 | 51.2±4.3 | 54.0±4.7 | 49.9±5.4 | 46.5±0.3 |
| CEGA | **58.4±3.2** | **64.6±0.4** | 66.8±0.8 | 68.0±1.1 | 67.4±0.7 |
| Realistic | 40.7±7.3 | 55.1±4.9 | 62.8±0.2 | 63.2±2.2 | 64.1±1.2 |
| DFEA_I | 36.9±4.8 | 41.6±6.1 | 48.4±5.5 | 37.4±9.8 | 30.3±3.1 |
| DFEA_II | 24.5±3.7 | 34.8±9.3 | 33.7±3.4 | 27.5±11.4 | 25.9±4.9 |
| DFEA_III | 31.1±1.5 | 41.5±4.9 | 26.0±5.1 | 28.9±0.7 | 18.2±0.0 |

(d) Regime=data_free

| Attack | 0.05 | 0.10 | 0.25 | 0.50 | 1.00 |
|---|---|---|---|---|---|
| MEA0 | 12.8±7.3 | 17.4±1.0 | 14.7±4.9 | 14.7±4.9 | 17.4±1.0 |
| MEA1 | 21.4±2.4 | 21.4±2.4 | 21.4±2.4 | 21.4±2.4 | 21.4±2.4 |
| MEA2 | 11.5±3.4 | 16.5±4.9 | 11.5±3.4 | 16.5±4.9 | 16.5±4.9 |
| MEA3 | 18.1±0.0 | 19.5±2.6 | 16.7±5.4 | 17.7±0.6 | 16.6±0.4 |
| MEA4 | 21.5±2.3 | 14.6±4.9 | 14.3±4.7 | 16.7±1.0 | 17.5±1.0 |
| MEA5 | 18.8±3.1 | 17.3±0.6 | 19.8±2.3 | 17.3±0.6 | 14.0±4.5 |
| AdvMEA | 48.9±2.9 | 47.6±1.7 | 53.1±8.2 | 50.1±2.5 | 50.1±5.8 |
| CEGA | **60.9±2.4** | **64.4±0.8** | **67.8±0.8** | **67.6±0.7** | **67.5±0.6** |
| Realistic | 59.0±2.0 | 58.2±2.6 | 59.0±1.5 | 57.0±1.5 | 60.1±2.5 |
| DFEA_I | 48.6±1.3 | 51.8±1.5 | 45.9±3.1 | 51.5±3.4 | 52.2±2.9 |
| DFEA_II | 36.4±1.4 | 34.4±5.8 | 30.3±1.2 | 26.9±4.7 | 33.7±2.4 |
| DFEA_III | 44.6±8.3 | 34.1±6.2 | 41.6±3.4 | 40.6±1.7 | 35.6±4.2 |

Table 11: RQ1 detailed for dataset=CiteSeer, metric=F1 (%). Rows are attacks; columns are budgets. Mean ± std across seeds; best per column is bold.

(a) Regime=both

| Attack | 0.05 | 0.10 | 0.25 | 0.50 | 1.00 |
|---|---|---|---|---|---|
| MEA0 | 50.9±4.6 | 55.4±1.7 | 63.7±0.6 | 66.0±0.8 | 66.6±0.2 |
| MEA1 | 34.4±1.3 | 35.0±4.3 | 51.0±8.3 | 63.7±1.0 | 65.9±0.6 |
| MEA2 | 5.2±2.4 | 7.5±3.5 | 19.0±4.5 | 19.7±2.9 | 11.3±3.3 |
| MEA3 | 43.3±5.1 | 54.1±3.0 | 65.4±0.5 | 65.8±1.0 | 66.5±0.1 |
| MEA4 | 36.5±2.6 | 55.1±4.6 | 63.8±0.1 | **66.4±0.5** | **67.2±0.3** |
| MEA5 | 42.1±0.6 | 56.9±1.8 | **65.8±1.1** | 65.8±0.5 | 66.8±0.5 |
| AdvMEA | 47.8±5.1 | 41.3±5.4 | 46.7±0.4 | 45.6±1.6 | 46.8±6.4 |
| CEGA | **54.7±2.3** | **61.0±1.5** | 63.4±1.1 | 63.9±1.4 | 63.9±0.2 |
| Realistic | 42.4±3.0 | 53.0±3.3 | 59.2±2.6 | 60.4±1.1 | 61.3±1.2 |
| DFEA_I | 35.0±4.1 | 45.8±1.5 | 42.1±2.8 | 25.4±11.0 | 24.1±5.3 |
| DFEA_II | 17.4±1.8 | 21.8±5.4 | 26.0±5.2 | 20.1±7.8 | 17.5±1.6 |
| DFEA_III | 31.8±0.6 | 33.9±12.0 | 21.9±4.8 | 16.4±10.3 | 7.1±2.8 |

(b) Regime=x_only

| Attack | 0.05 | 0.10 | 0.25 | 0.50 | 1.00 |
|---|---|---|---|---|---|
| MEA0 | 46.1±2.9 | 58.8±0.4 | 64.8±1.4 | 66.2±0.5 | 65.7±0.4 |
| MEA1 | 34.4±1.3 | 35.0±4.3 | 51.0±8.3 | 63.7±1.0 | 65.8±0.6 |
| MEA2 | 6.7±1.4 | 5.2±1.8 | 14.6±5.8 | 22.9±2.1 | 10.7±3.9 |
| MEA3 | 51.9±4.4 | 57.7±0.5 | 63.4±1.0 | 65.0±0.4 | 66.3±0.7 |
| MEA4 | 34.7±3.4 | 53.7±1.9 | 63.3±0.5 | **66.4±0.5** | **67.2±0.3** |
| MEA5 | 47.0±5.4 | 59.9±1.6 | **65.4±0.7** | 65.8±0.7 | 66.3±0.5 |
| AdvMEA | 43.3±0.4 | 47.5±0.5 | 50.3±3.0 | 44.4±6.0 | 42.3±2.5 |
| CEGA | **57.4±1.1** | **61.4±0.8** | 63.3±0.5 | 63.9±0.5 | 64.5±0.8 |
| Realistic | 42.1±5.1 | 47.4±5.2 | 58.1±1.9 | 59.4±2.3 | 61.6±1.9 |
| DFEA_I | 27.2±4.8 | 46.7±1.2 | 45.9±5.7 | 37.8±9.0 | 20.5±1.8 |
| DFEA_II | 24.9±3.4 | 30.7±7.8 | 23.7±7.4 | 18.1±6.7 | 15.8±2.2 |
| DFEA_III | 34.1±5.9 | 34.9±11.7 | 26.0±5.8 | 15.9±7.7 | 7.9±2.7 |

(c) Regime=a_only

| Attack | 0.05 | 0.10 | 0.25 | 0.50 | 1.00 |
|---|---|---|---|---|---|
| MEA0 | 46.0±3.1 | 57.1±3.2 | **65.8±1.3** | 65.6±0.6 | 66.6±1.1 |
| MEA1 | 34.4±1.3 | 35.0±4.3 | 51.0±8.3 | 63.7±1.0 | 65.9±0.6 |
| MEA2 | 7.3±1.9 | 8.9±1.5 | 12.1±8.0 | 18.0±3.2 | 11.1±3.3 |
| MEA3 | 39.5±6.3 | 56.5±1.3 | 63.6±2.1 | 66.1±0.8 | 66.4±0.4 |
| MEA4 | 33.0±4.6 | 54.5±1.7 | 63.1±0.7 | **66.4±0.5** | **67.2±0.3** |
| MEA5 | 45.2±2.5 | 59.0±2.1 | 65.1±1.5 | 65.8±0.3 | 67.1±0.9 |
| AdvMEA | 44.2±6.2 | 46.2±4.9 | 47.9±7.8 | 47.3±3.3 | 41.5±1.2 |
| CEGA | **54.1±1.9** | **61.4±0.1** | 63.4±0.7 | 65.0±0.9 | 64.1±0.9 |
| Realistic | 33.2±10.0 | 50.1±4.6 | 58.5±0.3 | 58.9±2.7 | 60.6±1.0 |
| DFEA_I | 30.8±4.4 | 38.3±6.9 | 44.5±5.3 | 29.9±11.4 | 23.3±5.7 |
| DFEA_II | 15.8±1.8 | 28.6±9.5 | 25.8±6.1 | 19.3±12.0 | 15.3±6.1 |
| DFEA_III | 22.5±2.2 | 34.5±5.2 | 16.1±4.1 | 14.3±0.3 | 5.1±0.0 |

(d) Regime=data_free

| Attack | 0.05 | 0.10 | 0.25 | 0.50 | 1.00 |
|---|---|---|---|---|---|
| MEA0 | 3.7±1.8 | 4.9±0.2 | 4.2±1.3 | 4.2±1.3 | 4.9±0.2 |
| MEA1 | 5.9±0.5 | 5.9±0.5 | 5.9±0.5 | 5.9±0.5 | 5.9±0.5 |
| MEA2 | 4.3±1.5 | 5.9±1.0 | 4.3±1.5 | 5.9±1.0 | 6.5±1.9 |
| MEA3 | 5.1±0.0 | 6.0±0.6 | 5.6±0.6 | 5.0±1.3 | 4.7±0.1 |
| MEA4 | 5.9±0.5 | 4.2±1.3 | 4.1±1.2 | 4.8±0.2 | 5.0±0.3 |
| MEA5 | 5.4±0.7 | 4.9±0.1 | 5.5±0.5 | 4.9±0.1 | 4.0±1.2 |
| AdvMEA | 45.1±0.8 | 42.0±1.2 | 48.5±4.5 | 47.1±1.8 | 47.1±4.9 |
| CEGA | **56.0±2.0** | **61.0±1.1** | **64.3±1.1** | **64.2±0.7** | **64.2±0.4** |
| Realistic | 53.9±1.4 | 53.4±3.4 | 54.4±0.7 | 51.7±3.7 | 54.9±2.8 |
| DFEA_I | 44.7±2.4 | 48.1±2.1 | 42.0±4.3 | 47.7±4.1 | 48.4±3.6 |
| DFEA_II | 27.3±2.9 | 24.7±7.2 | 22.9±2.4 | 21.7±4.4 | 26.9±1.5 |
| DFEA_III | 37.5±9.6 | 27.5±6.0 | 33.4±6.5 | 33.3±3.9 | 24.2±7.2 |

Table 12: RQ1 detailed for dataset=CiteSeer, metric=Fidelity (%). Rows are attacks; columns are budgets. Mean ± std across seeds; best per column is bold.

(a) Regime=both

| Attack | 0.05 | 0.10 | 0.25 | 0.50 | 1.00 |
|---|---|---|---|---|---|
| MEA0 | 62.9±6.1 | 69.9±3.6 | 82.8±1.5 | 86.7±1.0 | **90.0±0.7** |
| MEA1 | 47.3±2.1 | 47.4±2.3 | 64.7±7.6 | 81.0±1.7 | 83.4±0.9 |
| MEA2 | 15.7±4.1 | 17.2±5.6 | 31.9±6.0 | 33.4±5.6 | 23.0±4.9 |
| MEA3 | 55.6±3.6 | 69.0±3.6 | 82.5±1.5 | 86.3±2.2 | 88.5±0.5 |
| MEA4 | 48.3±2.9 | 65.4±5.3 | 79.4±0.8 | 83.3±1.5 | 84.1±1.3 |
| MEA5 | 56.5±1.5 | 72.1±2.9 | 83.0±0.7 | **87.2±0.4** | 87.9±0.8 |
| AdvMEA | 55.9±6.8 | 50.0±4.2 | 54.5±2.5 | 57.2±1.0 | 52.7±6.0 |
| CEGA | **68.6±2.0** | **79.8±0.7** | **83.2±0.9** | 86.3±0.8 | 87.2±1.0 |
| Realistic | 54.1±3.8 | 64.5±1.5 | 73.3±1.4 | 78.3±3.5 | 84.1±0.8 |
| DFEA_I | 46.4±5.1 | 59.2±1.4 | 57.5±0.9 | 39.3±11.5 | 36.6±4.3 |
| DFEA_II | 26.9±3.4 | 30.1±4.2 | 34.6±6.2 | 32.0±8.1 | 29.9±1.3 |
| DFEA_III | 39.7±2.7 | 43.8±10.2 | 35.3±4.9 | 33.3±12.0 | 20.9±2.8 |

(b) Regime=x_only

| Attack | 0.05 | 0.10 | 0.25 | 0.50 | 1.00 |
|---|---|---|---|---|---|
| MEA0 | 58.5±2.3 | 73.2±0.6 | 82.0±0.2 | 85.8±0.5 | **89.2±0.8** |
| MEA1 | 47.3±2.1 | 47.4±2.3 | 64.7±7.6 | 81.0±1.7 | 83.4±1.0 |
| MEA2 | 16.2±2.1 | 16.2±4.5 | 23.0±9.4 | 36.5±3.9 | 23.0±4.0 |
| MEA3 | 63.4±5.5 | 69.8±0.5 | 81.0±0.6 | 85.2±1.1 | 88.8±0.4 |
| MEA4 | 43.5±4.9 | 67.1±2.3 | 79.7±1.8 | 83.3±1.5 | 84.1±1.3 |
| MEA5 | 61.9±3.9 | 72.7±0.7 | 83.4±0.5 | 86.2±1.6 | 88.5±0.5 |
| AdvMEA | 52.4±3.0 | 55.2±3.2 | 57.7±3.7 | 52.7±6.8 | 50.7±3.0 |
| CEGA | **73.1±1.8** | **78.6±1.8** | **83.7±0.9** | **86.4±0.9** | 87.0±1.1 |
| Realistic | 54.5±3.8 | 60.3±2.7 | 74.5±1.8 | 80.0±2.9 | 84.4±1.1 |
| DFEA_I | 39.7±4.6 | 60.5±1.5 | 61.7±5.2 | 50.0±10.2 | 31.5±1.4 |
| DFEA_II | 34.8±5.2 | 40.6±5.4 | 31.0±5.9 | 30.0±6.3 | 27.2±3.3 |
| DFEA_III | 44.9±3.6 | 46.7±11.4 | 40.3±3.7 | 30.9±8.8 | 20.8±2.0 |

(c) Regime=a_only

| Attack | 0.05 | 0.10 | 0.25 | 0.50 | 1.00 |
|---|---|---|---|---|---|
| MEA0 | 57.6±3.5 | 73.1±4.4 | 82.9±0.5 | 85.8±0.5 | **89.8±0.7** |
| MEA1 | 47.3±2.1 | 47.4±2.3 | 64.7±7.6 | 81.0±1.7 | 83.4±0.9 |
| MEA2 | 16.3±4.0 | 17.2±3.7 | 23.1±10.0 | 28.6±4.5 | 23.8±4.2 |
| MEA3 | 54.4±4.3 | 72.5±3.2 | 80.8±0.7 | 87.1±0.3 | 87.9±0.5 |
| MEA4 | 44.5±4.4 | 66.6±2.9 | 80.0±1.6 | 83.3±1.5 | 84.1±1.3 |
| MEA5 | 58.2±2.7 | 73.7±2.3 | **84.0±1.7** | 86.2±1.5 | 88.5±0.8 |
| AdvMEA | 52.2±3.8 | 53.6±3.0 | 56.1±6.6 | 53.5±5.3 | 50.6±2.2 |
| CEGA | **69.9±3.5** | **79.6±2.4** | 81.9±0.9 | **87.2±0.3** | 88.8±0.5 |
| Realistic | 48.1±7.3 | 61.3±4.2 | 73.8±0.7 | 81.1±0.8 | 84.2±0.3 |
| DFEA_I | 43.4±6.6 | 51.1±9.7 | 59.3±7.3 | 43.6±11.6 | 34.4±4.3 |
| DFEA_II | 27.8±3.9 | 38.7±9.1 | 36.0±3.0 | 28.4±10.1 | 28.5±6.0 |
| DFEA_III | 31.9±3.2 | 45.6±5.5 | 29.4±4.3 | 31.7±2.3 | 18.9±0.3 |

(d) Regime=data_free

| Attack | 0.05 | 0.10 | 0.25 | 0.50 | 1.00 |
|---|---|---|---|---|---|
| MEA0 | 12.2±3.6 | 18.4±0.9 | 16.1±4.7 | 15.9±4.5 | 18.6±0.9 |
| MEA1 | 18.1±0.7 | 18.1±0.7 | 18.1±0.7 | 18.1±0.7 | 18.1±0.7 |
| MEA2 | 12.5±3.4 | 15.2±3.6 | 12.5±3.4 | 15.2±3.6 | 15.1±3.5 |
| MEA3 | 19.0±0.4 | 18.0±0.9 | 16.1±2.0 | 18.3±1.1 | 17.2±0.3 |
| MEA4 | 17.6±0.6 | 15.9±4.5 | 15.4±4.3 | 17.9±0.8 | 18.7±1.0 |
| MEA5 | 17.6±0.5 | 17.9±1.0 | 19.0±0.6 | 17.6±0.7 | 15.3±4.3 |
| AdvMEA | 54.3±3.5 | 52.7±1.0 | 56.3±6.5 | 53.3±2.1 | 54.6±3.1 |
| CEGA | **71.9±2.1** | **77.4±0.1** | **85.2±0.7** | **86.4±1.0** | **86.9±0.9** |
| Realistic | 70.6±2.6 | 69.8±3.7 | 67.3±1.1 | 67.2±2.3 | 70.8±3.1 |
| DFEA_I | 57.6±2.3 | 61.1±4.4 | 56.3±4.0 | 62.4±3.7 | 63.3±2.5 |
| DFEA_II | 38.0±3.9 | 34.4±5.2 | 32.9±0.6 | 29.5±3.8 | 36.8±2.1 |
| DFEA_III | 50.9±5.9 | 41.1±6.6 | 48.3±4.2 | 46.3±2.3 | 37.8±5.0 |

Table 13: RQ1 detailed for dataset=`CoauthorCS`, metric=Acc (%). Rows are attacks; columns are budgets. Mean ± std across seeds; best per column is bold.

(a) Regime=`both`

| Attack | 0.05 | 0.10 | 0.25 | 0.50 | 1.00 |
|---|---|---|---|---|---|
| MEA0 | 84.3±3.8 | 88.4±1.8 | 90.8±1.1 | 91.2±1.2 | 92.2±0.9 |
| MEA1 | 71.1±0.8 | 74.4±0.7 | 77.2±0.3 | 89.2±0.7 | 89.5±0.4 |
| MEA2 | 2.2±1.1 | 2.0±0.6 | 71.3±8.3 | 76.9±9.3 | 83.1±4.2 |
| MEA3 | 86.2±0.6 | 88.8±0.7 | 90.8±0.5 | 91.8±0.3 | **92.8±0.3** |
| MEA4 | 68.4±8.5 | 79.4±7.5 | 85.8±3.7 | 90.3±0.5 | 91.0±0.2 |
| MEA5 | 87.3±0.7 | 88.5±0.3 | 91.4±0.8 | **92.2±0.1** | 92.0±0.4 |
| AdvMEA | **92.0±0.4** | **91.8±0.2** | 91.7±0.4 | 91.8±0.4 | 91.9±0.3 |
| CEGA | 89.0±1.0 | 90.4±0.7 | 91.6±0.4 | 91.8±0.0 | 91.7±0.6 |
| Realistic | 71.9±5.8 | 81.1±4.4 | 81.0±3.4 | 79.1±3.4 | 77.7±2.1 |
| DFEA_I | 73.0±3.7 | 82.3±4.8 | 80.2±5.3 | 55.5±24.7 | 62.4±11.6 |
| DFEA_II | 56.4±10.8 | 58.3±3.2 | 44.4±23.0 | 26.5±21.1 | 21.7±20.3 |
| DFEA_III | 53.3±19.2 | 49.2±18.6 | 55.1±18.9 | 52.3±11.5 | 38.9±17.4 |

(b) Regime=`x_only`

| Attack | 0.05 | 0.10 | 0.25 | 0.50 | 1.00 |
|---|---|---|---|---|---|
| MEA0 | 83.5±2.7 | 89.1±0.2 | 91.5±0.6 | **92.3±0.2** | **92.2±1.6** |
| MEA1 | 71.1±0.8 | 74.4±0.7 | 77.2±0.3 | 89.2±0.7 | 89.5±0.4 |
| MEA2 | 5.5±1.5 | 3.4±1.4 | 71.4±9.4 | 80.5±6.7 | 84.6±4.5 |
| MEA3 | 79.2±3.8 | 89.0±1.0 | 90.2±0.7 | 91.2±0.5 | 91.6±0.3 |
| MEA4 | 65.3±10.4 | 73.3±0.6 | 86.9±1.5 | 89.7±0.2 | 91.0±0.3 |
| MEA5 | 83.2±2.8 | 89.7±1.9 | 91.0±0.4 | 91.6±1.2 | 91.7±1.1 |
| AdvMEA | **91.7±0.3** | **91.8±0.6** | **91.8±0.6** | 91.6±0.4 | 91.8±0.4 |
| CEGA | 88.7±0.8 | 90.3±0.4 | 91.0±0.1 | 91.7±0.5 | 91.9±0.4 |
| Realistic | 68.8±8.3 | 77.2±4.3 | 81.2±2.2 | 78.9±2.4 | 79.3±1.2 |
| DFEA_I | 74.4±4.4 | 83.5±3.5 | 81.6±7.2 | 65.8±19.6 | 54.7±22.6 |
| DFEA_II | 60.0±6.2 | 39.6±15.9 | 42.7±17.1 | 36.8±19.7 | 24.3±24.2 |
| DFEA_III | 70.0±2.5 | 65.9±4.0 | 50.7±21.4 | 45.6±20.8 | 31.5±18.7 |

(c) Regime=`a_only`

| Attack | 0.05 | 0.10 | 0.25 | 0.50 | 1.00 |
|---|---|---|---|---|---|
| MEA0 | 85.1±2.1 | 89.0±1.4 | 90.5±0.5 | 91.0±0.3 | 91.4±0.7 |
| MEA1 | 71.1±0.8 | 74.4±0.7 | 77.2±0.3 | 89.2±0.7 | 89.5±0.4 |
| MEA2 | 4.2±2.4 | 2.2±0.8 | 71.6±8.6 | 76.4±10.2 | 82.4±5.2 |
| MEA3 | 87.6±0.4 | 89.5±1.3 | 90.8±0.7 | 91.9±0.3 | **92.5±0.4** |
| MEA4 | 58.5±15.4 | 82.6±3.3 | 87.2±0.2 | 90.0±0.7 | 91.0±0.4 |
| MEA5 | 86.5±1.2 | 89.2±0.7 | 91.3±0.0 | **92.0±0.9** | 92.0±0.6 |
| AdvMEA | **91.8±0.2** | **91.7±0.3** | **91.9±0.2** | 91.8±0.4 | 91.6±0.5 |
| CEGA | 90.0±0.2 | 90.4±0.2 | 91.8±0.5 | 91.3±0.8 | 91.7±0.7 |
| Realistic | 74.5±3.2 | 80.8±2.0 | 84.4±4.4 | 77.1±3.0 | 77.3±2.3 |
| DFEA_I | 52.7±15.3 | 85.8±1.8 | 84.5±4.7 | 70.4±6.9 | 53.0±23.9 |
| DFEA_II | 66.0±6.5 | 48.0±17.1 | 35.1±17.2 | 43.0±16.5 | 20.0±21.3 |
| DFEA_III | 35.5±13.6 | 46.8±17.5 | 57.6±14.9 | 37.8±17.0 | 26.8±23.0 |

(d) Regime=`data_free`

| Attack | 0.05 | 0.10 | 0.25 | 0.50 | 1.00 |
|---|---|---|---|---|---|
| MEA0 | 7.4±2.7 | 33.5±23.5 | 7.4±3.8 | 3.5±2.7 | 22.1±19.8 |
| MEA1 | 0.5±0.4 | 0.5±0.4 | 0.5±0.4 | 0.5±0.4 | 0.5±0.4 |
| MEA2 | 4.2±4.4 | 4.2±4.4 | 4.6±4.1 | 4.2±4.4 | 4.2±4.4 |
| MEA3 | 21.5±20.3 | 22.1±19.9 | 17.5±23.1 | 37.2±18.2 | 18.9±19.5 |
| MEA4 | 4.0±2.4 | 22.6±19.5 | 18.4±22.5 | 20.1±21.5 | 7.1±3.0 |
| MEA5 | 17.0±12.7 | 26.4±23.8 | 18.1±22.7 | 6.2±4.3 | 7.8±5.0 |
| AdvMEA | **91.7±0.3** | **91.9±0.2** | **91.8±0.3** | **91.8±0.4** | **91.8±0.3** |
| CEGA | 88.8±1.0 | 91.1±0.6 | 91.4±0.3 | 91.3±0.8 | 91.1±0.6 |
| Realistic | 74.9±2.2 | 77.4±2.8 | 75.1±2.0 | 79.7±1.1 | 79.9±1.9 |
| DFEA_I | 49.8±24.9 | 53.0±22.2 | 58.7±24.1 | 64.8±2.2 | 49.4±23.6 |
| DFEA_II | 18.9±9.2 | 29.2±16.9 | 23.5±20.7 | 27.4±23.9 | 23.2±19.1 |
| DFEA_III | 38.8±20.3 | 43.0±21.2 | 26.8±17.1 | 41.6±22.6 | 26.7±20.9 |

Table 14: RQ1 detailed for dataset=`CoauthorCS`, metric=F1 (%). Rows are attacks; columns are budgets. Mean ± std across seeds; best per column is bold.

(a) Regime=`both`

| Attack | 0.05 | 0.10 | 0.25 | 0.50 | 1.00 |
|---|---|---|---|---|---|
| MEA0 | 51.8±1.1 | 61.1±3.9 | 70.1±3.4 | 73.7±3.8 | 77.1±2.5 |
| MEA1 | 34.5±0.3 | 36.7±0.4 | 42.1±1.7 | 61.2±0.9 | 64.3±1.3 |
| MEA2 | 0.6±0.5 | 0.8±0.5 | 36.7±12.0 | 48.1±14.5 | 53.3±7.0 |
| MEA3 | 54.8±2.8 | 64.2±4.6 | 71.6±4.2 | 76.8±0.6 | 76.4±1.5 |
| MEA4 | 25.2±4.7 | 38.0±4.8 | 52.3±8.8 | 67.1±1.9 | 71.3±1.1 |
| MEA5 | 54.6±1.0 | 59.5±1.4 | 71.8±5.1 | **77.6±1.4** | **77.7±0.1** |
| AdvMEA | **75.0±1.6** | **73.7±0.8** | 74.0±1.8 | 74.9±2.2 | 75.3±1.0 |
| CEGA | 67.6±2.1 | 72.6±1.5 | **74.7±2.0** | 76.5±0.9 | 74.1±2.4 |
| Realistic | 42.9±6.9 | 53.4±2.3 | 61.3±3.9 | 56.6±5.6 | 54.2±3.5 |
| DFEA_I | 38.1±4.8 | 47.2±7.2 | 42.0±4.5 | 23.4±7.4 | 22.5±7.3 |
| DFEA_II | 23.8±3.0 | 26.1±2.4 | 16.3±6.8 | 8.2±2.8 | 4.0±1.2 |
| DFEA_III | 22.6±6.7 | 27.6±4.3 | 24.5±5.5 | 15.0±3.4 | 8.0±2.7 |

(b) Regime=`x_only`

| Attack | 0.05 | 0.10 | 0.25 | 0.50 | 1.00 |
|---|---|---|---|---|---|
| MEA0 | 44.4±3.8 | 60.9±4.1 | 74.5±0.4 | **76.1±1.7** | 74.8±4.7 |
| MEA1 | 34.5±0.3 | 36.7±0.4 | 42.1±1.7 | 61.2±0.9 | 64.3±1.3 |
| MEA2 | 1.4±0.5 | 1.2±0.7 | 37.6±14.5 | 49.2±7.2 | 52.4±5.3 |
| MEA3 | 45.6±0.8 | 62.5±6.8 | 67.6±3.0 | 74.1±1.6 | 73.5±1.6 |
| MEA4 | 20.8±8.6 | 36.1±6.9 | 52.2±4.5 | 64.5±0.7 | 71.1±0.8 |
| MEA5 | 50.4±3.0 | 66.6±5.2 | 71.6±4.0 | 73.1±6.9 | 74.4±3.6 |
| AdvMEA | **74.7±0.7** | **74.7±2.0** | **74.6±1.6** | 74.3±1.0 | 73.2±1.9 |
| CEGA | 70.0±0.9 | 70.8±1.5 | 74.0±2.8 | 72.5±1.3 | 72.8±1.4 |
| Realistic | 46.1±1.1 | 46.4±3.0 | 56.5±1.7 | 56.9±3.7 | 55.8±0.6 |
| DFEA_I | 40.0±2.4 | 54.4±3.2 | 45.1±8.7 | 38.1±5.0 | 23.4±4.4 |
| DFEA_II | 20.2±5.9 | 22.7±3.0 | 15.9±3.3 | 11.5±4.4 | 7.9±3.3 |
| DFEA_III | 33.0±6.0 | 27.0±3.0 | 21.7±5.2 | 14.6±4.4 | 8.8±3.2 |

(c) Regime=`a_only`

| Attack | 0.05 | 0.10 | 0.25 | 0.50 | 1.00 |
|---|---|---|---|---|---|
| MEA0 | 56.0±6.3 | 65.7±4.6 | 71.3±2.8 | 73.2±1.6 | 73.4±3.3 |
| MEA1 | 34.5±0.3 | 36.7±0.4 | 42.1±1.7 | 61.2±0.9 | 64.3±1.3 |
| MEA2 | 0.8±0.7 | 1.0±0.6 | 35.9±9.2 | 48.4±15.0 | 50.9±4.4 |
| MEA3 | 57.1±0.9 | 64.4±7.1 | 70.5±4.1 | 74.8±2.2 | **75.8±0.9** |
| MEA4 | 21.3±4.2 | 47.2±8.5 | 52.4±0.9 | 66.1±2.5 | 71.3±1.4 |
| MEA5 | 53.6±3.3 | 67.3±4.9 | 74.1±1.1 | **77.3±1.9** | 74.1±2.0 |
| AdvMEA | **75.0±0.8** | **75.2±1.0** | **74.7±1.1** | 74.6±1.1 | 74.4±1.0 |
| CEGA | 69.4±2.1 | 71.2±0.9 | 73.9±0.9 | 73.3±3.0 | 72.3±1.3 |
| Realistic | 41.5±6.9 | 56.2±2.7 | 61.6±5.3 | 54.4±5.0 | 54.5±0.8 |
| DFEA_I | 33.4±2.1 | 56.0±3.9 | 50.9±10.0 | 26.2±11.4 | 20.7±10.3 |
| DFEA_II | 29.3±0.7 | 22.3±5.1 | 14.9±2.7 | 13.4±2.4 | 4.5±0.5 |
| DFEA_III | 22.2±7.3 | 25.6±9.0 | 28.6±2.3 | 15.9±1.8 | 9.3±4.4 |

(d) Regime=`data_free`

| Attack | 0.05 | 0.10 | 0.25 | 0.50 | 1.00 |
|---|---|---|---|---|---|
| MEA0 | 0.9±0.3 | 3.0±2.1 | 0.9±0.5 | 0.4±0.3 | 2.1±1.6 |
| MEA1 | 0.1±0.0 | 0.1±0.0 | 0.1±0.0 | 0.1±0.0 | 0.1±0.0 |
| MEA2 | 0.7±0.7 | 0.7±0.7 | 0.8±0.6 | 0.7±0.7 | 0.7±0.7 |
| MEA3 | 2.4±1.5 | 3.4±1.9 | 1.6±2.0 | 3.9±0.8 | 2.9±2.8 |
| MEA4 | 0.5±0.3 | 2.2±1.6 | 1.7±2.0 | 1.9±1.8 | 0.9±0.4 |
| MEA5 | 2.7±2.2 | 5.5±5.0 | 1.8±1.9 | 0.8±0.5 | 3.6±2.6 |
| AdvMEA | **75.1±1.0** | **74.9±0.8** | **73.9±0.9** | 74.7±0.9 | **74.6±0.7** |
| CEGA | 65.1±3.0 | 72.6±2.4 | 71.9±1.0 | **75.5±0.8** | 72.0±1.0 |
| Realistic | 52.0±1.7 | 55.1±3.3 | 54.1±1.5 | 59.5±0.8 | 56.9±2.8 |
| DFEA_I | 17.0±9.1 | 19.8±5.7 | 25.5±6.6 | 22.2±4.6 | 16.4±7.5 |
| DFEA_II | 6.1±1.5 | 6.3±1.8 | 5.6±1.2 | 10.7±6.9 | 5.4±1.0 |
| DFEA_III | 7.8±2.3 | 11.1±2.7 | 7.3±2.8 | 10.2±6.1 | 8.1±1.1 |

Table 15: RQ1 detailed for dataset=CoauthorCS, metric=Fidelity (%). Rows are attacks; columns are budgets. Mean ± std across seeds; best per column is bold.

(a) Regime=both

| Attack | 0.05 | 0.10 | 0.25 | 0.50 | 1.00 |
|---|---|---|---|---|---|
| MEA0 | 83.7±4.0 | 87.9±1.6 | 91.2±1.2 | 92.8±0.3 | 94.5±0.3 |
| MEA1 | 71.0±0.8 | 74.2±0.5 | 76.9±0.5 | 89.8±0.8 | 90.2±0.7 |
| MEA2 | 2.9±1.2 | 3.0±0.8 | 72.0±7.0 | 77.6±8.6 | 83.4±3.7 |
| MEA3 | 85.2±1.8 | 88.2±1.3 | 90.8±1.0 | 91.7±0.2 | 93.1±0.6 |
| MEA4 | 66.7±8.4 | 79.1±7.2 | 86.9±3.4 | 93.5±0.7 | **96.3±0.5** |
| MEA5 | 85.8±0.2 | 87.8±0.3 | 91.1±0.2 | 92.4±0.4 | 93.6±0.8 |
| AdvMEA | **92.3±0.5** | **92.3±0.4** | 92.0±0.4 | 92.1±0.3 | 92.3±0.4 |
| CEGA | 90.3±0.6 | 91.7±0.7 | **93.5±0.7** | **94.9±0.8** | 94.8±0.5 |
| Realistic | 72.6±5.2 | 81.5±4.3 | 82.5±3.2 | 80.4±3.7 | 80.9±2.0 |
| DFEA_I | 72.5±3.4 | 81.5±5.5 | 79.4±5.5 | 54.8±25.0 | 61.5±10.8 |
| DFEA_II | 55.1±10.0 | 57.4±3.1 | 43.1±22.6 | 25.8±20.6 | 20.8±19.7 |
| DFEA_III | 52.6±19.2 | 49.0±17.4 | 54.9±18.8 | 51.3±11.6 | 37.7±16.8 |

(b) Regime=x_only

| Attack | 0.05 | 0.10 | 0.25 | 0.50 | 1.00 |
|---|---|---|---|---|---|
| MEA0 | 83.3±2.8 | 88.8±0.5 | 92.3±0.7 | 93.3±0.5 | 93.7±0.2 |
| MEA1 | 71.0±0.8 | 74.2±0.5 | 76.9±0.5 | 89.8±0.8 | 90.2±0.7 |
| MEA2 | 5.8±1.6 | 4.3±1.3 | 72.3±8.3 | 81.3±6.0 | 85.1±3.9 |
| MEA3 | 77.8±4.5 | 88.6±0.9 | 90.0±0.6 | 91.7±0.4 | 93.4±0.5 |
| MEA4 | 63.8±10.0 | 72.2±0.8 | 87.7±1.2 | 92.9±0.3 | **96.4±0.4** |
| MEA5 | 83.5±2.4 | 89.2±2.0 | 91.0±0.3 | 92.0±0.7 | 93.1±1.0 |
| AdvMEA | **92.0±0.4** | 92.1±0.3 | 92.2±0.6 | 92.0±0.3 | 92.2±0.2 |
| CEGA | 90.2±0.6 | **92.7±0.3** | **92.8±0.4** | **94.5±0.3** | 96.2±0.3 |
| Realistic | 68.4±8.8 | 78.1±4.7 | 81.4±1.7 | 80.8±1.9 | 80.5±1.1 |
| DFEA_I | 73.4±3.8 | 82.6±3.6 | 81.2±7.4 | 66.0±19.5 | 53.9±21.9 |
| DFEA_II | 59.0±5.3 | 39.5±16.3 | 42.3±16.3 | 36.6±18.6 | 24.2±23.8 |
| DFEA_III | 68.9±2.1 | 65.7±3.7 | 50.7±21.0 | 45.4±21.2 | 30.8±18.3 |

(c) Regime=a_only

| Attack | 0.05 | 0.10 | 0.25 | 0.50 | 1.00 |
|---|---|---|---|---|---|
| MEA0 | 84.6±2.4 | 89.4±1.4 | 91.9±0.6 | 92.9±0.4 | 94.6±0.4 |
| MEA1 | 71.0±0.8 | 74.2±0.5 | 76.9±0.5 | 89.8±0.8 | 90.2±0.7 |
| MEA2 | 4.3±2.2 | 2.4±0.6 | 72.6±7.8 | 77.8±9.3 | 82.9±4.4 |
| MEA3 | 86.5±0.5 | 89.5±1.4 | 90.8±0.9 | 92.0±0.4 | 93.5±0.1 |
| MEA4 | 57.4±15.1 | 81.4±2.8 | 88.5±0.3 | 92.8±1.3 | **96.6±0.4** |
| MEA5 | 86.2±1.6 | 89.0±0.6 | 91.1±0.4 | 92.4±0.7 | 93.7±1.0 |
| AdvMEA | **92.0±0.3** | 92.0±0.4 | 92.3±0.4 | 91.9±0.3 | 91.9±0.3 |
| CEGA | 90.2±1.0 | **92.5±0.5** | **93.8±0.5** | **94.2±0.3** | 95.8±0.3 |
| Realistic | 73.9±3.3 | 82.2±1.5 | 85.7±4.5 | 78.6±3.7 | 79.0±2.6 |
| DFEA_I | 52.0±14.9 | 85.7±1.2 | 83.8±4.8 | 69.7±7.4 | 51.8±23.1 |
| DFEA_II | 65.0±5.8 | 47.0±16.4 | 35.2±16.4 | 42.0±16.0 | 19.1±20.8 |
| DFEA_III | 35.6±13.5 | 46.8±17.2 | 57.9±14.1 | 37.6±16.4 | 26.6±23.3 |

(d) Regime=data_free

| Attack | 0.05 | 0.10 | 0.25 | 0.50 | 1.00 |
|---|---|---|---|---|---|
| MEA0 | 6.6±2.5 | 32.2±22.7 | 7.0±3.9 | 3.4±2.8 | 21.2±19.1 |
| MEA1 | 0.6±0.7 | 0.6±0.7 | 0.6±0.7 | 0.6±0.7 | 0.6±0.7 |
| MEA2 | 4.9±4.4 | 4.9±4.4 | 5.6±3.9 | 4.9±4.4 | 4.9±4.4 |
| MEA3 | 20.4±19.6 | 21.2±19.1 | 16.8±22.4 | 36.1±17.3 | 18.7±18.9 |
| MEA4 | 3.9±2.2 | 22.1±18.8 | 17.4±21.7 | 19.4±20.5 | 7.0±3.1 |
| MEA5 | 16.3±12.3 | 25.8±23.0 | 17.3±21.8 | 5.8±3.7 | 8.0±5.0 |
| AdvMEA | **91.9±0.4** | 92.2±0.4 | 92.2±0.4 | 92.0±0.5 | 92.1±0.5 |
| CEGA | 90.0±1.2 | **92.3±0.6** | **93.1±0.3** | **93.8±0.7** | **95.5±0.5** |
| Realistic | 77.8±2.5 | 78.9±2.1 | 76.4±2.5 | 80.3±1.9 | 81.5±2.1 |
| DFEA_I | 49.0±24.4 | 52.4±21.7 | 57.9±23.4 | 64.8±2.4 | 48.7±23.6 |
| DFEA_II | 18.4±9.0 | 28.2±16.5 | 22.7±20.0 | 26.6±23.4 | 22.4±18.6 |
| DFEA_III | 38.5±20.5 | 42.0±20.9 | 26.5±16.0 | 41.1±22.8 | 26.0±20.5 |

Table 16: RQ1 detailed for dataset=CoauthorPhysics, metric=Acc (%). Rows are attacks; columns are budgets. Mean ± std across seeds; best per column is bold.

(a) Regime=both

| Attack | 0.05 | 0.10 | 0.25 | 0.50 | 1.00 |
|---|---|---|---|---|---|
| MEA0 | 82.2±4.9 | 89.6±0.4 | 90.9±0.2 | 90.7±0.4 | **92.8±1.0** |
| MEA1 | 3.5±0.0 | 10.1±0.6 | 17.2±0.2 | 44.0±1.9 | 79.5±7.9 |
| MEA2 | 20.4±11.9 | 34.7±7.8 | 35.8±5.3 | 30.3±0.4 | 50.0±28.2 |
| MEA3 | 74.0±1.4 | 81.1±2.6 | 89.5±1.4 | 89.8±0.7 | 90.6±1.8 |
| MEA4 | 70.1±10.5 | 74.0±10.0 | 87.7±2.0 | 89.6±0.5 | 89.5±0.6 |
| MEA5 | 75.4±4.8 | 78.4±4.7 | 89.8±1.6 | **91.0±0.8** | 91.3±1.0 |
| AdvMEA | **91.4±1.2** | **91.6±0.7** | 91.3±0.2 | 90.6±1.3 | 91.4±0.9 |
| CEGA | 90.9±1.4 | 90.4±2.8 | **91.5±0.6** | 91.0±0.5 | 90.8±0.3 |
| Realistic | 79.6±3.8 | 77.0±6.5 | 79.3±1.7 | 76.1±3.1 | 73.0±1.5 |
| DFEA_I | 64.6±19.3 | 70.4±9.2 | 67.5±16.0 | 69.5±11.1 | 21.1±9.8 |
| DFEA_II | 52.7±24.6 | 53.5±13.3 | 16.8±12.9 | 16.0±14.5 | 3.9±0.4 |
| DFEA_III | 52.7±26.7 | 72.8±3.9 | 70.0±7.3 | 22.2±20.9 | 3.5±0.0 |

(b) Regime=x_only

| Attack | 0.05 | 0.10 | 0.25 | 0.50 | 1.00 |
|---|---|---|---|---|---|
| MEA0 | 80.9±4.2 | 90.4±1.8 | 89.3±1.5 | 91.5±0.9 | 91.3±0.8 |
| MEA1 | 3.5±0.0 | 10.1±0.6 | 17.2±0.2 | 44.0±1.9 | 79.5±7.9 |
| MEA2 | 17.6±9.1 | 7.7±1.1 | 40.8±11.4 | 29.4±3.5 | 32.1±2.6 |
| MEA3 | 75.3±1.5 | 84.0±2.4 | 89.8±0.9 | **91.6±0.5** | **91.4±0.5** |
| MEA4 | 39.5±17.4 | 83.4±4.5 | 88.7±1.6 | 90.2±0.6 | 89.9±0.3 |
| MEA5 | 76.2±5.5 | 78.9±5.5 | 89.0±1.5 | 91.5±0.7 | 91.2±1.4 |
| AdvMEA | 90.8±1.2 | 91.2±1.0 | 90.7±0.4 | 91.6±0.4 | 90.9±0.9 |
| CEGA | **90.9±0.4** | **91.2±1.0** | **91.6±0.3** | 91.2±0.5 | 91.2±0.4 |
| Realistic | 79.0±5.6 | 80.3±4.6 | 79.4±2.7 | 78.6±2.6 | 74.4±0.5 |
| DFEA_I | 75.8±11.5 | 86.0±0.9 | 77.8±15.1 | 50.5±25.5 | 33.9±16.7 |
| DFEA_II | 51.6±5.7 | 72.3±6.5 | 24.7±13.5 | 18.7±15.6 | 5.7±3.1 |
| DFEA_III | 70.6±10.5 | 73.4±1.4 | 63.2±9.2 | 26.6±16.8 | 3.5±0.0 |

(c) Regime=a_only

| Attack | 0.05 | 0.10 | 0.25 | 0.50 | 1.00 |
|---|---|---|---|---|---|
| MEA0 | 85.3±2.4 | 89.0±0.8 | 91.1±0.4 | 90.1±2.0 | **92.0±0.6** |
| MEA1 | 3.5±0.0 | 10.1±0.6 | 17.2±0.2 | 44.0±1.9 | 79.5±7.9 |
| MEA2 | 20.2±18.0 | 14.1±11.3 | 32.7±21.1 | 31.2±1.7 | 31.2±1.7 |
| MEA3 | 70.5±6.3 | 83.9±2.6 | 89.6±1.0 | 90.7±1.1 | 91.3±0.5 |
| MEA4 | 61.6±24.8 | 77.9±6.2 | 86.0±3.0 | 89.4±0.5 | 89.7±0.6 |
| MEA5 | 73.1±9.3 | 83.9±1.0 | 89.4±1.5 | 90.7±0.7 | 90.4±0.6 |
| AdvMEA | **91.4±0.4** | 91.0±0.7 | 91.6±0.7 | 91.5±0.6 | 91.5±0.8 |
| CEGA | 90.7±1.0 | **91.8±0.6** | **91.6±0.3** | **91.3±0.1** | 91.5±0.8 |
| Realistic | 72.2±5.2 | 84.5±1.3 | 72.7±7.7 | 74.1±1.2 | 69.8±2.1 |
| DFEA_I | 70.6±6.4 | 82.7±2.1 | 75.7±12.9 | 58.9±28.2 | 29.8±14.2 |
| DFEA_II | 57.5±6.2 | 68.4±10.8 | 25.6±20.0 | 8.8±5.3 | 3.5±0.0 |
| DFEA_III | 63.7±2.8 | 74.4±3.3 | 53.6±12.3 | 33.4±36.4 | 3.8±0.4 |

(d) Regime=data_free

| Attack | 0.05 | 0.10 | 0.25 | 0.50 | 1.00 |
|---|---|---|---|---|---|
| MEA0 | 5.5±2.9 | 27.6±18.8 | 3.5±0.0 | 15.7±10.1 | 20.8±20.3 |
| MEA1 | 3.5±0.0 | 3.5±0.0 | 3.5±0.0 | 3.5±0.0 | 3.5±0.0 |
| MEA2 | 29.7±16.5 | 29.7±16.5 | 29.7±16.5 | 29.7±16.5 | 29.7±16.5 |
| MEA3 | 6.7±2.3 | 19.5±21.1 | 28.6±19.9 | 35.8±22.7 | 18.8±21.6 |
| MEA4 | 4.9±1.9 | 13.7±11.6 | 5.5±2.9 | 14.4±11.3 | 12.3±12.5 |
| MEA5 | 3.5±0.0 | 34.0±21.6 | 29.9±12.2 | 25.0±18.8 | 18.8±21.6 |
| AdvMEA | **91.1±0.5** | **91.6±0.7** | 91.2±0.9 | **91.2±0.8** | 91.0±0.4 |
| CEGA | 89.0±1.5 | 90.7±0.7 | **91.6±0.5** | 91.0±0.2 | **91.4±0.3** |
| Realistic | 70.1±1.6 | 73.5±1.7 | 71.5±1.4 | 71.5±2.7 | 69.6±3.6 |
| DFEA_I | 8.0±3.2 | 8.6±3.6 | 28.1±29.9 | 16.7±18.7 | 7.7±3.3 |
| DFEA_II | 3.5±0.0 | 3.5±0.0 | 3.5±0.0 | 3.5±0.0 | 3.5±0.0 |
| DFEA_III | 3.5±0.0 | 3.5±0.0 | 3.5±0.0 | 3.5±0.0 | 3.5±0.0 |

Table 17: RQ1 detailed for dataset=CoauthorPhysics, metric=F1 (%). Rows are attacks; columns are budgets. Mean ± std across seeds; best per column is bold.

(a) Regime=both

| Attack | 0.05 | 0.10 | 0.25 | 0.50 | 1.00 |
|---|---|---|---|---|---|
| MEA0 | 71.3±6.1 | 78.7±2.0 | 80.3±0.6 | 80.9±0.8 | **84.2±1.5** |
| MEA1 | 1.4±0.0 | 10.4±1.8 | 18.5±1.1 | 44.0±1.2 | 71.6±5.2 |
| MEA2 | 9.2±5.3 | 14.3±6.6 | 13.8±3.3 | 13.9±6.6 | 33.5±34.4 |
| MEA3 | 61.1±3.7 | 70.1±2.5 | 79.9±2.3 | 79.5±0.9 | 80.4±3.1 |
| MEA4 | 38.8±7.7 | 48.9±6.7 | 68.6±1.4 | 74.8±1.7 | 76.1±0.9 |
| MEA5 | 62.6±6.6 | 67.6±4.2 | 78.6±2.6 | 79.3±1.6 | 81.6±0.6 |
| AdvMEA | **82.9±2.3** | **82.9±1.7** | **82.9±1.0** | 81.5±2.2 | 82.5±2.0 |
| CEGA | 80.5±1.4 | 80.3±3.9 | 80.3±1.6 | 79.8±1.3 | 80.2±0.2 |
| Realistic | 64.7±4.0 | 64.9±5.4 | 65.2±1.1 | 62.9±3.2 | 60.0±1.1 |
| DFEA_I | 52.5±12.8 | 57.7±5.2 | 50.6±19.7 | 47.2±11.9 | 17.8±7.9 |
| DFEA_II | 41.2±15.8 | 40.9±7.3 | 13.2±8.9 | 10.7±8.9 | 2.8±1.3 |
| DFEA_III | 37.1±15.1 | 54.1±3.2 | 49.2±5.5 | 10.2±8.1 | 1.4±0.0 |

(b) Regime=x_only

| Attack | 0.05 | 0.10 | 0.25 | 0.50 | 1.00 |
|---|---|---|---|---|---|
| MEA0 | 63.9±4.3 | 78.5±3.6 | 77.5±2.1 | 79.4±1.4 | **81.9±1.0** |
| MEA1 | 1.4±0.0 | 10.4±1.8 | 18.5±1.1 | 44.0±1.2 | 71.6±5.2 |
| MEA2 | 7.6±1.7 | 5.2±1.8 | 18.1±6.3 | 15.0±5.2 | 13.7±4.5 |
| MEA3 | 58.1±4.4 | 71.1±2.8 | 79.4±0.5 | 81.4±0.8 | 81.5±1.6 |
| MEA4 | 21.8±5.2 | 61.3±8.0 | 69.3±4.0 | 75.1±1.9 | 76.9±0.6 |
| MEA5 | 63.0±6.6 | 68.0±3.4 | 78.6±2.6 | 81.1±1.4 | 81.0±1.6 |
| AdvMEA | 81.9±2.2 | 82.1±1.9 | 81.7±1.1 | **83.2±0.9** | 81.6±2.2 |
| CEGA | 79.8±1.9 | 81.6±1.2 | **81.8±0.5** | 81.2±1.0 | 81.1±0.7 |
| Realistic | 63.4±4.6 | 68.1±5.3 | 65.6±4.9 | 64.1±3.9 | 60.6±0.3 |
| DFEA_I | 55.2±12.2 | 73.0±3.4 | 58.9±19.4 | 31.6±20.4 | 18.9±8.8 |
| DFEA_II | 34.0±4.4 | 48.5±9.6 | 20.7±9.4 | 13.6±11.0 | 3.4±2.9 |
| DFEA_III | 50.1±10.6 | 54.0±8.3 | 41.2±7.7 | 22.0±13.6 | 1.4±0.0 |

(c) Regime=a_only

| Attack | 0.05 | 0.10 | 0.25 | 0.50 | 1.00 |
|---|---|---|---|---|---|
| MEA0 | 70.1±6.9 | 78.0±1.8 | 80.6±0.4 | 79.9±3.5 | **82.7±1.6** |
| MEA1 | 1.4±0.0 | 10.4±1.8 | 18.5±1.1 | 44.0±1.2 | 71.6±5.2 |
| MEA2 | 9.4±3.9 | 6.2±3.3 | 18.4±9.8 | 13.7±6.3 | 13.2±3.5 |
| MEA3 | 62.6±2.9 | 72.7±1.8 | 78.8±1.7 | 79.9±3.0 | 81.4±1.0 |
| MEA4 | 42.8±15.8 | 51.2±5.7 | 64.2±3.1 | 76.4±2.3 | 76.6±0.9 |
| MEA5 | 60.1±9.1 | 72.1±0.8 | 78.0±2.2 | 79.9±1.2 | 79.1±1.1 |
| AdvMEA | **82.8±1.1** | **82.1±1.7** | **82.9±2.2** | **81.8±1.8** | 82.1±1.5 |
| CEGA | 79.2±2.1 | 80.7±0.7 | 81.6±0.7 | 81.3±1.2 | 80.6±0.5 |
| Realistic | 52.4±8.7 | 70.5±2.6 | 60.2±6.0 | 60.1±2.4 | 56.2±1.3 |
| DFEA_I | 54.1±3.3 | 65.7±6.7 | 57.0±16.3 | 42.2±20.2 | 17.7±7.3 |
| DFEA_II | 37.1±9.0 | 44.0±8.6 | 18.6±10.8 | 6.9±3.9 | 1.4±0.0 |
| DFEA_III | 40.5±2.6 | 55.0±8.9 | 39.1±5.2 | 22.6±25.6 | 2.4±1.5 |

(d) Regime=data_free

| Attack | 0.05 | 0.10 | 0.25 | 0.50 | 1.00 |
|---|---|---|---|---|---|
| MEA0 | 2.1±1.0 | 7.9±4.9 | 1.4±0.0 | 5.2±2.9 | 6.0±5.2 |
| MEA1 | 1.4±0.0 | 1.4±0.0 | 1.4±0.0 | 1.4±0.0 | 1.4±0.0 |
| MEA2 | 11.1±6.8 | 11.1±6.8 | 11.1±6.8 | 11.1±6.8 | 11.1±6.8 |
| MEA3 | 3.5±2.1 | 6.2±5.1 | 10.2±7.7 | 17.1±9.7 | 5.3±5.6 |
| MEA4 | 1.8±0.7 | 4.5±3.4 | 2.1±1.0 | 4.7±3.3 | 4.0±3.7 |
| MEA5 | 1.4±0.0 | 9.3±5.6 | 12.3±4.0 | 9.0±5.4 | 5.3±5.6 |
| AdvMEA | **82.5±1.3** | **82.7±1.6** | **82.2±2.0** | **82.7±1.6** | **82.3±0.9** |
| CEGA | 77.8±1.5 | 79.5±2.5 | 80.4±0.9 | 80.4±0.5 | 80.8±1.0 |
| Realistic | 56.4±3.0 | 58.3±1.0 | 58.5±1.9 | 56.0±2.7 | 55.1±2.9 |
| DFEA_I | 7.3±5.0 | 7.0±4.3 | 20.1±22.6 | 12.3±15.4 | 7.5±4.9 |
| DFEA_II | 1.4±0.0 | 1.4±0.0 | 1.4±0.0 | 1.4±0.0 | 1.4±0.0 |
| DFEA_III | 1.4±0.0 | 1.4±0.0 | 1.4±0.0 | 1.4±0.0 | 1.4±0.0 |

Table 18: RQ1 detailed for dataset=CoauthorPhysics, metric=Fidelity (%). Rows are attacks; columns are budgets. Mean ± std across seeds; best per column is bold.

(a) Regime=both

| Attack | 0.05 | 0.10 | 0.25 | 0.50 | 1.00 |
|---|---|---|---|---|---|
| MEA0 | 81.9±4.5 | 89.3±0.9 | 92.5±0.7 | 92.8±0.9 | 94.2±0.1 |
| MEA1 | 2.7±0.5 | 9.5±0.3 | 15.3±0.7 | 44.4±1.6 | 78.6±7.3 |
| MEA2 | 21.3±13.1 | 36.3±6.7 | 37.3±4.9 | 32.3±0.1 | 51.5±27.2 |
| MEA3 | 74.4±2.1 | 80.8±3.0 | 89.1±1.0 | 90.4±1.7 | 93.0±0.7 |
| MEA4 | 69.2±11.7 | 75.4±11.2 | 90.7±0.9 | **95.5±0.6** | **97.8±0.4** |
| MEA5 | 74.3±5.8 | 77.5±4.9 | 89.5±1.5 | 91.2±1.4 | 91.9±0.2 |
| AdvMEA | 90.0±0.5 | 90.2±0.5 | 89.7±0.8 | 89.5±0.9 | 90.2±0.4 |
| CEGA | **91.0±1.4** | **91.6±1.3** | **94.1±0.0** | 94.9±0.3 | 96.8±0.3 |
| Realistic | 80.5±4.1 | 77.5±6.4 | 82.2±1.6 | 77.4±2.2 | 74.3±1.4 |
| DFEA_I | 64.0±19.5 | 70.9±10.1 | 69.1±16.7 | 69.1±10.8 | 19.8±9.4 |
| DFEA_II | 51.9±24.3 | 52.1±13.1 | 16.2±12.9 | 15.1±14.4 | 3.1±0.5 |
| DFEA_III | 51.1±26.4 | 70.5±5.6 | 70.0±7.8 | 21.3±21.1 | 2.7±0.5 |

(b) Regime=x_only

| Attack | 0.05 | 0.10 | 0.25 | 0.50 | 1.00 |
|---|---|---|---|---|---|
| MEA0 | 80.4±4.7 | 89.9±1.2 | 92.0±1.5 | 93.6±0.7 | 93.3±0.8 |
| MEA1 | 2.7±0.5 | 9.5±0.3 | 15.3±0.7 | 44.4±1.6 | 78.6±7.3 |
| MEA2 | 19.0±9.6 | 7.5±2.3 | 42.3±11.5 | 31.6±3.9 | 33.9±2.7 |
| MEA3 | 74.5±1.4 | 83.1±2.9 | 90.1±1.2 | 92.6±1.4 | 92.9±0.4 |
| MEA4 | 40.3±18.5 | 84.0±4.2 | 90.9±0.1 | 94.4±0.4 | **98.1±0.6** |
| MEA5 | 75.8±5.7 | 77.8±6.2 | 89.1±1.4 | 92.4±1.1 | 93.4±0.9 |
| AdvMEA | 89.8±0.6 | 90.4±0.2 | 89.8±0.2 | 90.0±0.3 | 89.9±0.3 |
| CEGA | **92.3±1.3** | **93.0±1.3** | **94.1±0.6** | 94.7±0.8 | 96.3±0.8 |
| Realistic | 79.1±6.6 | 84.3±4.3 | 81.0±2.9 | 79.2±2.2 | 76.7±1.1 |
| DFEA_I | 75.7±11.4 | 86.9±2.0 | 78.3±15.6 | 49.7±26.6 | 33.0±16.4 |
| DFEA_II | 51.8±5.2 | 72.2±7.3 | 24.1±14.0 | 17.4±14.4 | 4.6±2.7 |
| DFEA_III | 70.9±11.9 | 74.2±1.1 | 44.1±9.6 | 26.9±17.9 | 2.7±0.5 |

(c) Regime=a_only

| Attack | 0.05 | 0.10 | 0.25 | 0.50 | 1.00 |
|---|---|---|---|---|---|
| MEA0 | 84.5±3.8 | 88.6±0.5 | 91.9±0.8 | 91.8±1.0 | 94.5±0.4 |
| MEA1 | 2.7±0.5 | 9.5±0.3 | 15.3±0.7 | 44.4±1.6 | 78.6±7.3 |
| MEA2 | 19.8±19.3 | 15.3±12.6 | 33.4±21.2 | 34.2±2.8 | 32.8±1.8 |
| MEA3 | 69.7±6.2 | 84.3±3.2 | 89.8±0.5 | 90.7±0.3 | 93.7±0.8 |
| MEA4 | 61.1±25.0 | 78.4±6.6 | 89.0±0.9 | 94.6±0.7 | **97.9±0.3** |
| MEA5 | 72.1±9.2 | 84.0±1.2 | 90.4±1.9 | 92.6±0.9 | 94.4±0.3 |
| AdvMEA | 90.0±0.0 | 90.1±0.1 | 90.1±0.4 | 89.9±0.8 | 89.7±0.3 |
| CEGA | **91.5±0.4** | **93.3±0.6** | **93.5±0.4** | **95.6±1.1** | 96.2±0.4 |
| Realistic | 72.8±4.6 | 85.6±2.4 | 73.0±7.3 | 75.6±1.1 | 72.0±3.2 |
| DFEA_I | 70.8±0.4 | 84.3±1.5 | 77.4±13.5 | 57.7±29.0 | 29.0±14.4 |
| DFEA_II | 56.8±5.5 | 66.9±10.9 | 24.0±19.5 | 8.1±5.4 | 2.7±0.5 |
| DFEA_III | 62.4±3.3 | 73.4±2.5 | 52.3±11.6 | 33.5±37.9 | 3.0±0.2 |

(d) Regime=data_free

| Attack | 0.05 | 0.10 | 0.25 | 0.50 | 1.00 |
|---|---|---|---|---|---|
| MEA0 | 4.6±2.8 | 27.6±18.8 | 2.7±0.5 | 16.3±11.3 | 19.5±20.3 |
| MEA1 | 2.7±0.5 | 2.7±0.5 | 2.7±0.5 | 2.7±0.5 | 2.7±0.5 |
| MEA2 | 29.9±17.3 | 29.9±17.3 | 29.9±17.3 | 29.9±17.3 | 29.9±17.3 |
| MEA3 | 6.2±2.2 | 18.9±20.8 | 28.8±20.2 | 35.8±22.6 | 17.8±21.4 |
| MEA4 | 4.6±2.7 | 14.6±12.7 | 4.6±2.8 | 14.7±12.6 | 12.5±13.9 |
| MEA5 | 2.7±0.5 | 33.3±21.6 | 30.5±12.9 | 24.6±18.9 | 17.9±21.9 |
| AdvMEA | **89.7±0.6** | 90.5±0.9 | 89.8±0.6 | 90.0±0.7 | 89.7±0.3 |
| CEGA | 89.6±1.7 | **92.8±0.7** | **95.0±0.2** | **95.1±1.0** | **96.5±0.4** |
| Realistic | 72.7±2.0 | 75.2±2.1 | 73.5±0.7 | 73.5±2.0 | 71.6±4.1 |
| DFEA_I | 7.2±3.2 | 7.7±3.2 | 27.9±30.2 | 15.9±18.2 | 6.6±2.9 |
| DFEA_II | 2.7±0.5 | 2.7±0.5 | 2.7±0.5 | 2.7±0.5 | 2.7±0.5 |
| DFEA_III | 2.7±0.5 | 2.7±0.5 | 2.7±0.5 | 2.7±0.5 | 2.7±0.5 |

Table 19: RQ1 detailed for dataset=Computers, metric=Acc (%). Rows are attacks; columns are budgets. Mean ± std across seeds; best per column is bold.

(a) Regime=both

| Attack | 0.05 | 0.10 | 0.25 | 0.50 | 1.00 |
|---|---|---|---|---|---|
| MEA0 | 51.7±7.0 | 60.4±4.2 | 65.7±2.4 | 67.3±2.3 | 71.4±1.4 |
| MEA1 | 46.4±19.1 | 31.3±14.7 | 48.1±2.6 | 51.9±1.4 | 53.0±5.3 |
| MEA2 | 20.2±6.9 | 35.8±13.6 | 43.4±15.0 | 27.1±21.0 | 57.1±5.5 |
| MEA3 | 61.6±2.2 | **69.8±2.0** | 66.0±1.3 | 69.3±2.5 | 70.7±3.1 |
| MEA4 | 41.7±14.8 | 35.0±19.2 | 65.9±4.1 | 66.5±3.7 | **71.7±0.3** |
| MEA5 | **67.8±2.0** | 68.8±2.9 | **69.4±3.1** | **70.5±3.3** | 70.1±4.2 |
| AdvMEA | 34.7±17.3 | 20.8±22.8 | 22.4±23.3 | 46.8±13.1 | 16.8±11.4 |
| CEGA | 29.8±21.9 | 27.8±20.7 | 34.8±14.0 | 45.4±24.1 | 39.6±25.5 |
| Realistic | 2.4±1.1 | 1.1±0.7 | 1.1±0.7 | 1.5±1.2 | 1.1±0.7 |
| DFEA_I | 49.0±5.6 | 49.3±1.9 | 48.5±12.9 | 19.8±12.5 | 26.6±21.9 |
| DFEA_II | 25.1±16.4 | 29.5±11.1 | 16.8±7.0 | 26.5±20.9 | 26.6±21.9 |
| DFEA_III | 8.1±8.6 | 22.2±20.0 | 26.6±21.9 | 26.6±21.9 | 26.6±21.9 |

(b) Regime=x_only

| Attack | 0.05 | 0.10 | 0.25 | 0.50 | 1.00 |
|---|---|---|---|---|---|
| MEA0 | 54.7±8.8 | 63.8±1.1 | 64.4±5.4 | 64.7±6.9 | 67.0±4.9 |
| MEA1 | 46.4±19.1 | 31.3±14.7 | 48.0±2.5 | 51.7±1.3 | 52.9±5.6 |
| MEA2 | 15.4±12.4 | 11.4±7.5 | 25.3±21.1 | 29.8±15.4 | 56.1±6.4 |
| MEA3 | 63.7±3.0 | **68.4±1.8** | 69.2±3.4 | 70.3±3.9 | 70.1±1.8 |
| MEA4 | 31.7±4.2 | 37.0±22.1 | 61.4±4.3 | 67.2±0.9 | 71.4±1.1 |
| MEA5 | **69.9±2.3** | 67.7±2.3 | 67.9±3.1 | **71.1±3.3** | **72.1±1.9** |
| AdvMEA | 27.6±23.9 | 33.1±21.1 | 31.6±22.4 | 29.5±21.4 | 40.4±26.8 |
| CEGA | 27.9±17.2 | 27.5±20.0 | 43.6±25.0 | 25.1±22.5 | 41.6±29.1 |
| Realistic | 12.9±15.6 | 13.2±16.7 | 1.1±0.7 | 13.6±16.6 | 10.9±13.6 |
| DFEA_I | 37.3±9.0 | 48.6±9.2 | 36.5±11.8 | 38.6±14.4 | 26.3±22.0 |
| DFEA_II | 20.4±18.4 | 21.1±8.8 | 30.2±18.2 | 26.6±21.8 | 26.6±21.9 |
| DFEA_III | 27.2±19.9 | 31.2±18.3 | 26.6±21.9 | 26.6±21.9 | 26.6±21.9 |

(c) Regime=a_only

| Attack | 0.05 | 0.10 | 0.25 | 0.50 | 1.00 |
|---|---|---|---|---|---|
| MEA0 | 62.9±5.2 | 64.4±2.9 | 62.3±1.6 | 67.7±2.6 | 67.6±3.3 |
| MEA1 | 46.4±19.1 | 31.3±14.7 | 48.1±2.6 | 51.9±1.4 | 53.0±5.3 |
| MEA2 | 26.1±15.1 | 25.1±21.0 | 14.1±11.1 | 40.2±22.0 | 57.5±7.1 |
| MEA3 | 62.1±2.2 | 67.2±1.0 | 68.1±4.2 | 65.2±5.8 | 71.7±1.7 |
| MEA4 | 40.2±16.5 | 44.6±19.3 | 54.0±17.1 | 62.2±7.8 | 68.4±0.5 |
| MEA5 | **64.5±4.1** | **69.9±0.5** | **68.3±1.8** | **71.5±4.9** | **71.7±1.6** |
| AdvMEA | 36.9±14.9 | 40.3±16.2 | 27.0±23.9 | 31.6±24.1 | 17.2±8.3 |
| CEGA | 29.1±16.6 | 31.0±20.7 | 39.7±23.2 | 36.1±25.4 | 39.3±28.1 |
| Realistic | 12.7±16.1 | 1.5±1.2 | 15.0±19.4 | 1.1±0.7 | 14.6±18.4 |
| DFEA_I | 34.9±13.6 | 56.2±7.0 | 51.8±4.5 | 38.7±23.5 | 27.3±21.3 |
| DFEA_II | 20.4±14.0 | 41.3±7.2 | 35.3±12.9 | 27.2±21.6 | 26.6±21.9 |
| DFEA_III | 17.0±1.5 | 41.0±13.8 | 26.6±21.9 | 26.6±21.9 | 26.6±21.9 |

(d) Regime=data_free

| Attack | 0.05 | 0.10 | 0.25 | 0.50 | 1.00 |
|---|---|---|---|---|---|
| MEA0 | 25.7±22.9 | 11.7±7.5 | 29.7±18.9 | 44.1±16.9 | 8.6±8.2 |
| MEA1 | **56.1±0.0** | **56.1±0.0** | **56.1±0.0** | **56.1±0.0** | **56.1±0.0** |
| MEA2 | 22.4±21.6 | 15.3±18.6 | 32.1±21.5 | 32.1±21.5 | 32.1±21.5 |
| MEA3 | 38.6±4.4 | 41.7±20.4 | 27.3±20.9 | 25.6±23.0 | 19.8±25.7 |
| MEA4 | 25.6±23.0 | 4.7±5.8 | 19.0±26.2 | 5.7±5.3 | 4.7±5.8 |
| MEA5 | 19.3±26.0 | 20.9±25.0 | 25.4±22.0 | 26.5±21.2 | 24.8±22.8 |
| AdvMEA | 42.6±13.0 | 32.0±16.6 | 23.6±21.7 | 20.7±17.0 | 39.4±10.4 |
| CEGA | 30.4±15.3 | 17.2±17.0 | 39.6±28.3 | 34.9±26.8 | 36.4±24.5 |
| Realistic | 13.6±17.4 | 1.3±0.5 | 10.8±13.4 | 8.2±9.7 | 1.0±0.6 |
| DFEA_I | 27.3±21.1 | 31.3±17.5 | 27.1±21.1 | 28.7±19.8 | 26.3±22.0 |
| DFEA_II | 27.3±21.7 | 26.8±21.8 | 26.3±21.5 | 27.0±21.3 | 26.7±22.0 |
| DFEA_III | 26.6±21.9 | 26.6±21.9 | 26.6±21.9 | 26.6±21.9 | 26.6±21.9 |

Table 20: RQ1 detailed for dataset=Computers, metric=F1 (%). Rows are attacks; columns are budgets. Mean ± std across seeds; best per column is bold.

(a) Regime=both

| Attack | 0.05 | 0.10 | 0.25 | 0.50 | 1.00 |
|---|---|---|---|---|---|
| MEA0 | 22.0±6.0 | **44.3±4.0** | 46.4±8.0 | 44.1±2.0 | **60.0±0.6** |
| MEA1 | 13.0±6.0 | 7.6±3.7 | 17.2±5.5 | 19.3±7.8 | 22.7±4.6 |
| MEA2 | 7.0±2.5 | 12.5±1.8 | 8.4±1.1 | 7.7±5.0 | 11.4±5.9 |
| MEA3 | 20.6±4.3 | 41.7±4.6 | 44.0±4.2 | **54.5±4.0** | 58.8±2.5 |
| MEA4 | 13.8±1.8 | 16.5±9.6 | 28.0±6.6 | 36.7±9.7 | 46.8±4.4 |
| MEA5 | **29.8±1.1** | 39.5±4.5 | **56.3±4.8** | 53.5±1.9 | 53.7±8.7 |
| AdvMEA | 21.7±13.1 | 14.9±14.0 | 15.5±16.6 | 30.0±8.0 | 12.0±9.9 |
| CEGA | 26.8±18.9 | 22.8±15.0 | 31.3±15.1 | 39.5±23.5 | 33.4±21 |
| Realistic | 2.0±1.2 | 0.2±0.1 | 0.2±0.1 | 0.5±0.3 | 0.2±0.1 |
| DFEA_I | 19.1±2.9 | 17.9±3.3 | 12.2±3.3 | 3.8±2.6 | 3.8±2.6 |
| DFEA_II | 7.2±4.6 | 7.8±2.3 | 3.3±1.1 | 4.8±3.7 | 3.7±2.7 |
| DFEA_III | 1.4±1.4 | 4.8±4.7 | 3.7±2.7 | 3.7±2.7 | 3.7±2.7 |

(b) Regime=x_only

| Attack | 0.05 | 0.10 | 0.25 | 0.50 | 1.00 |
|---|---|---|---|---|---|
| MEA0 | 29.9±7.4 | **41.9±4.5** | 43.1±8.7 | 42.5±18.6 | 50.8±7.5 |
| MEA1 | 13.0±6.0 | 7.6±3.7 | 17.1±5.5 | 19.9±8.5 | 27.8±2.7 |
| MEA2 | 6.0±3.8 | 6.4±4.3 | 6.5±3.7 | 6.8±2.9 | 12.3±7.3 |
| MEA3 | 29.0±3.6 | 40.5±2.5 | 50.5±7.4 | 51.0±12.1 | 56.9±3.9 |
| MEA4 | 12.7±4.4 | 15.6±5.3 | 24.6±5.1 | 34.5±9.6 | 47.8±1.1 |
| MEA5 | **32.9±4.8** | 39.0±4.6 | **50.6±8.6** | **52.5±4.6** | **59.5±1.9** |
| AdvMEA | 17.3±13.6 | 17.5±12.9 | 22.8±16.0 | 18.8±11.7 | 27.5±18.8 |
| CEGA | 29.1±17.2 | 27.1±19.3 | 36.9±20.2 | 21.0±20.3 | 35.8±25.5 |
| Realistic | 3.9±4.3 | 5.1±6.8 | 0.2±0.1 | 5.5±7.2 | 3.2±4.1 |
| DFEA_I | 16.0±2.5 | 21.1±6.0 | 7.8±0.5 | 7.0±2.2 | 3.8±2.7 |
| DFEA_II | 6.5±4.4 | 7.0±2.0 | 6.2±1.9 | 3.7±2.7 | 3.7±2.7 |
| DFEA_III | 5.4±3.9 | 6.0±1.8 | 3.7±2.7 | 3.7±2.7 | 3.7±2.7 |

(c) Regime=a_only

| Attack | 0.05 | 0.10 | 0.25 | 0.50 | 1.00 |
|---|---|---|---|---|---|
| MEA0 | **33.3±3.9** | **44.3±5.7** | 35.8±10.0 | 55.1±0.6 | 52.9±5.0 |
| MEA1 | 13.0±6.0 | 7.6±3.7 | 17.2±5.5 | 19.3±7.8 | 22.7±4.6 |
| MEA2 | 10.4±3.6 | 11.9±4.9 | 3.7±1.9 | 11.7±8.5 | 11.5±5.7 |
| MEA3 | 24.9±1.6 | 34.9±1.8 | 45.5±9.0 | 42.4±18.4 | **60.8±2.4** |
| MEA4 | 15.1±4.4 | 16.4±9.1 | 27.8±5.7 | 32.1±8.9 | 40.6±5.5 |
| MEA5 | 32.5±5.3 | 38.5±1.8 | **48.9±5.2** | **57.0±9.1** | 58.4±1.5 |
| AdvMEA | 23.1±12.3 | 25.9±11.9 | 17.8±16.5 | 20.0±15.2 | 11.9±6.3 |
| CEGA | 21.5±15.5 | 22.0±23.1 | 33.2±19.2 | 31.4±19.8 | 34.6±25.0 |
| Realistic | 4.0±5.2 | 0.4±0.3 | 4.3±5.6 | 0.2±0.1 | 4.8±5.8 |
| DFEA_I | 15.3±3.1 | 23.0±6.6 | 12.4±3.7 | 6.9±4.4 | 4.0±2.4 |
| DFEA_II | 6.7±5.0 | 14.0±1.4 | 7.7±2.5 | 4.0±2.6 | 3.7±2.7 |
| DFEA_III | 4.7±0.3 | 6.3±1.3 | 3.7±2.7 | 3.7±2.7 | 3.7±2.7 |

(d) Regime=data_free

| Attack | 0.05 | 0.10 | 0.25 | 0.50 | 1.00 |
|---|---|---|---|---|---|
| MEA0 | 3.6±2.9 | 2.0±1.2 | 4.3±2.1 | 5.9±1.8 | 1.5±1.3 |
| MEA1 | 7.2±0.0 | 7.2±0.0 | 7.2±0.0 | 7.2±0.0 | 7.2±0.0 |
| MEA2 | 4.7±3.1 | 3.4±3.6 | 6.2±3.4 | 6.2±3.4 | 6.2±3.4 |
| MEA3 | 5.0±3.1 | 5.6±2.3 | 5.1±3.5 | 3.5±2.9 | 2.7±3.2 |
| MEA4 | 3.5±2.9 | 0.8±1.0 | 2.5±3.3 | 1.0±0.9 | 0.8±1.0 |
| MEA5 | 2.7±3.2 | 3.3±2.9 | 4.8±3.0 | 4.4±2.1 | 4.4±2.9 |
| AdvMEA | **31.8±11.4** | **21.1±13.9** | 13.9±11.5 | 16.2±13.4 | 23.3±8.1 |
| CEGA | 21.7±13.3 | 15.7±18.4 | **32.8±22.6** | **28.1±20.6** | **35.5±22.6** |
| Realistic | 5.2±6.9 | 1.0±1.1 | 3.6±4.7 | 2.2±2.7 | 0.2±0.1 |
| DFEA_I | 4.2±2.2 | 5.3±1.4 | 4.3±2.2 | 4.5±1.9 | 3.8±2.7 |
| DFEA_II | 4.0±2.7 | 3.8±2.7 | 5.2±4.7 | 3.9±2.5 | 4.3±3.4 |
| DFEA_III | 3.7±2.7 | 3.7±2.7 | 3.7±2.7 | 3.7±2.7 | 3.7±2.7 |

Table 21: RQ1 detailed for dataset=Computers, metric=Fidelity (%). Rows are attacks; columns are budgets. Mean ± std across seeds; best per column is bold.

(a) Regime=both

| Attack | 0.05 | 0.10 | 0.25 | 0.50 | 1.00 |
|---|---|---|---|---|---|
| MEA0 | 51.0±7.0 | 67.3±3.1 | 67.0±1.5 | 71.4±2.7 | 80.5±4.6 |
| MEA1 | 41.5±15.5 | 27.7±14.0 | 46.2±5.5 | 49.4±5.6 | 56.3±12.6 |
| MEA2 | 17.0±7.1 | 31.5±8.2 | 32.2±10.4 | 26.3±12.5 | 43.1±6.3 |
| MEA3 | 53.8±3.0 | 64.8±5.5 | 67.6±6.2 | **74.9±4.3** | 79.5±5.2 |
| MEA4 | 38.4±13.4 | 37.9±12.6 | 63.6±4.0 | 74.3±5.0 | **83.7±4.3** |
| MEA5 | 61.3±4.3 | 65.6±6.4 | 66.5±2.1 | 73.9±3.5 | 76.3±4.2 |
| AdvMEA | 35.5±17.8 | 25.7±24.7 | 26.9±22.4 | 46.1±12.5 | 22.8±11.8 |
| CEGA | 36.0±25.0 | 36.7±23.0 | 43.4±17.5 | 53.9±29.2 | 54.1±33.4 |
| Realistic | **64.5±33.7** | **67.9±45.4** | **67.7±45.3** | 67.4±44.0 | 67.0±44.8 |
| DFEA_I | 51.7±7.2 | 49.2±4.6 | 43.6±4.9 | 16.6±8.3 | 21.4±14.8 |
| DFEA_II | 21.4±12.4 | 25.0±6.9 | 14.4±4.2 | 22.5±15.5 | 21.5±14.7 |
| DFEA_III | 8.4±6.9 | 18.1±15.8 | 21.5±14.7 | 21.5±14.7 | 21.5±14.7 |

(b) Regime=x_only

| Attack | 0.05 | 0.10 | 0.25 | 0.50 | 1.00 |
|---|---|---|---|---|---|
| MEA0 | 54.8±10.5 | 65.4±7.0 | 70.6±2.9 | 69.7±10.9 | 79.8±3.1 |
| MEA1 | 41.5±15.5 | 27.7±14.0 | 46.2±5.5 | 49.1±5.7 | 57.1±11.4 |
| MEA2 | 19.7±13.0 | 12.4±4.3 | 23.7±14.0 | 26.8±11.5 | 43.3±6.4 |
| MEA3 | 56.5±7.4 | 61.9±5.6 | 68.2±4.2 | 71.7±4.9 | 77.6±3.1 |
| MEA4 | 33.7±0.4 | 34.7±18.5 | 65.6±4.0 | 75.7±0.8 | **84.6±3.4** |
| MEA5 | 59.4±3.4 | 65.5±5.2 | **71.0±3.6** | 75.4±5.6 | 79.5±3.5 |
| AdvMEA | 29.0±24.1 | 33.6±19.9 | 34.5±23.4 | 28.7±16.6 | 39.8±25.8 |
| CEGA | 30.9±16.1 | 33.6±25.2 | 49.2±24.3 | 33.1±30.1 | 49.8±33.8 |
| Realistic | **75.1±25.8** | **78.7±30.0** | 67.9±45.4 | **81.0±25.5** | 78.6±25.1 |
| DFEA_I | 35.5±6.9 | 45.1±7.7 | 31.0±8.2 | 31.8±10.2 | 21.5±14.7 |
| DFEA_II | 20.9±16.3 | 18.9±6.0 | 25.1±11.3 | 21.4±14.6 | 21.5±14.7 |
| DFEA_III | 23.2±17.1 | 24.4±12.5 | 21.4±14.7 | 21.5±14.7 | 21.5±14.7 |

(c) Regime=a_only

| Attack | 0.05 | 0.10 | 0.25 | 0.50 | 1.00 |
|---|---|---|---|---|---|
| MEA0 | 60.7±2.6 | **66.5±6.5** | 66.1±1.6 | **77.1±2.9** | 78.8±0.9 |
| MEA1 | 41.5±15.5 | 27.7±14.0 | 46.2±5.5 | 49.4±5.6 | 56.3±12.6 |
| MEA2 | 27.9±11.1 | 21.3±15.0 | 14.0±6.1 | 36.1±19.4 | 44.5±6.6 |
| MEA3 | 58.0±4.7 | 62.7±5.5 | 68.4±4.2 | 65.1±4.4 | 79.5±4.4 |
| MEA4 | 42.3±14.5 | 38.0±15.4 | 54.9±16.2 | 64.1±4.0 | **81.2±2.2** |
| MEA5 | 57.7±4.0 | 62.8±3.9 | 69.7±5.2 | 72.5±3.9 | 80.9±3.3 |
| AdvMEA | 39.9±16.5 | 40.0±16.3 | 32.3±22.2 | 32.2±23.3 | 20.2±8.3 |
| CEGA | 37.9±19.9 | 37.9±23.5 | 50.1±27.4 | 45.4±30.8 | 51.6±32.9 |
| Realistic | **69.5±26.3** | 64.5±43.1 | **77.2±30.2** | 67.8±45.4 | 80.8±26.0 |
| DFEA_I | 36.7±14.3 | 57.7±13.2 | 43.1±6.5 | 30.6±16.2 | 22.0±14.2 |
| DFEA_II | 20.2±11.1 | 37.9±5.0 | 27.8±8.5 | 21.8±14.4 | 21.4±14.7 |
| DFEA_III | 17.2±6.3 | 33.0±9.7 | 21.5±14.7 | 21.5±14.7 | 21.5±14.7 |

(d) Regime=data_free

| Attack | 0.05 | 0.10 | 0.25 | 0.50 | 1.00 |
|---|---|---|---|---|---|
| MEA0 | 22.2±16.0 | 9.5±6.5 | 22.7±14.2 | 33.7±11.0 | 14.2±10.6 |
| MEA1 | 40.4±1.6 | 40.4±1.6 | 40.4±1.6 | 40.4±1.6 | 40.4±1.6 |
| MEA2 | 16.9±16.5 | 14.2±14.3 | 26.0±16.1 | 26.0±16.1 | 26.0±16.1 |
| MEA3 | 31.8±13.7 | 29.5±15.1 | 22.4±13.8 | 19.9±16.5 | 15.6±18.0 |
| MEA4 | 23.9±16.7 | 3.3±3.6 | 15.1±19.0 | 6.7±5.1 | 4.4±4.7 |
| MEA5 | 16.1±17.6 | 15.7±16.2 | 18.4±13.6 | 22.6±12.8 | 23.1±14.1 |
| AdvMEA | 44.9±13.6 | 34.2±18.1 | 21.6±17.4 | 24.3±18.1 | 40.7±12.2 |
| CEGA | 33.6±15.6 | 21.0±18.4 | 49.4±31.1 | 46.2±32.9 | 50.0±30.4 |
| Realistic | **82.1±22.2** | **70.2±41.7** | **82.5±24.8** | **75.5±34.6** | **66.5±44.5** |
| DFEA_I | 22.2±13.9 | 26.2±10.5 | 22.1±14.0 | 23.4±12.7 | 21.5±14.7 |
| DFEA_II | 21.7±14.7 | 21.6±14.7 | 22.1±15.6 | 22.0±14.0 | 21.5±14.7 |
| DFEA_III | 21.5±14.7 | 21.5±14.7 | 21.5±14.7 | 21.5±14.7 | 21.5±14.7 |

Table 22: RQ1 detailed for dataset=Photo, metric=Acc (%). Rows are attacks; columns are budgets. Mean ± std across seeds; best per column is bold.

(a) Regime=both

| Attack | 0.05 | 0.10 | 0.25 | 0.50 | 1.00 |
|---|---|---|---|---|---|
| MEA0 | 55.1±30.4 | 90.2±3.8 | 94.9±0.6 | 96.4±0.2 | 95.5±0.4 |
| MEA1 | 2.0±0.2 | 39.5±26.2 | 42.2±28.9 | 62.7±33.5 | 67.0±36.5 |
| MEA2 | 20.5±2.7 | 25.6±13.6 | 76.6±21.2 | 63.5±20.0 | 34.9±32.2 |
| MEA3 | 78.9±14.6 | **93.9±2.4** | **96.2±0.4** | **96.5±0.2** | 96.3±0.5 |
| MEA4 | 74.6±2.4 | 83.0±8.4 | 95.1±0.3 | 95.5±0.4 | 96.0±0.5 |
| MEA5 | **80.2±7.1** | 92.8±1.3 | 95.0±1.6 | 95.5±0.8 | **96.4±0.2** |
| AdvMEA | 49.0±22.8 | 38.0±24.8 | 47.3±27.6 | 36.8±26.7 | 40.9±24.8 |
| CEGA | 79.6±5.4 | 83.0±14.5 | 89.8±6.4 | 94.8±0.6 | 88.9±9.3 |
| Realistic | 49.0±1.1 | 47.7±8.3 | 64.5±19.4 | 63.3±20.6 | 50.3±7.0 |
| DFEA_I | 29.0±19.9 | 32.3±20.4 | 9.1±12.5 | 0.3±0.3 | 0.3±0.3 |
| DFEA_II | 12.7±7.8 | 9.4±6.5 | 2.6±3.5 | 6.0±4.4 | 0.3±0.3 |
| DFEA_III | 12.2±7.3 | 11.5±8.1 | 5.9±7.8 | 0.6±0.4 | 0.3±0.3 |

(b) Regime=x_only

| Attack | 0.05 | 0.10 | 0.25 | 0.50 | 1.00 |
|---|---|---|---|---|---|
| MEA0 | 84.5±5.9 | 91.1±2.3 | 95.4±0.8 | 95.9±0.8 | 96.0±0.4 |
| MEA1 | 2.0±0.2 | 39.6±26.4 | 42.2±28.9 | 62.7±33.5 | 67.0±36.5 |
| MEA2 | 27.2±22.0 | 18.5±11.6 | 69.1±19.9 | 75.5±13.1 | 34.5±32.3 |
| MEA3 | 83.0±5.1 | 92.8±2.1 | 95.7±0.1 | 95.8±0.8 | **96.2±0.3** |
| MEA4 | 67.2±18.6 | 86.0±2.1 | 87.8±8.1 | 96.0±0.4 | 96.0±0.1 |
| MEA5 | 82.5±13.8 | **94.5±1.7** | **95.9±0.1** | **96.4±0.2** | 96.1±0.6 |
| AdvMEA | 48.7±30.4 | 40.1±24.6 | 42.7±26.8 | 44.0±25.6 | 57.3±17.0 |
| CEGA | **84.9±6.6** | 85.9±6.1 | 82.7±7.0 | 93.8±1.9 | 94.1±3.0 |
| Realistic | 53.5±5.7 | 34.5±15.1 | 51.7±5.8 | 61.5±14.2 | 56.9±15.7 |
| DFEA_I | 19.6±9.9 | 11.2±7.0 | 8.6±7.1 | 0.6±0.4 | 0.5±0.3 |
| DFEA_II | 11.3±7.0 | 7.8±5.4 | 1.9±2.6 | 3.2±3.6 | 0.3±0.3 |
| DFEA_III | 6.8±6.9 | 3.9±5.0 | 0.3±0.3 | 0.3±0.3 | 0.3±0.3 |

(c) Regime=a_only

| Attack | 0.05 | 0.10 | 0.25 | 0.50 | 1.00 |
|---|---|---|---|---|---|
| MEA0 | 87.2±5.3 | 92.2±1.8 | 93.9±1.5 | 95.9±0.2 | 95.7±0.5 |
| MEA1 | 2.0±0.2 | 39.4±26.0 | 42.2±28.9 | 62.7±33.5 | 67.0±36.5 |
| MEA2 | 17.1±10.3 | 20.2±4.7 | 74.2±17.2 | 69.5±17.2 | 34.4±32.0 |
| MEA3 | 81.6±3.9 | 92.5±1.8 | 95.6±0.2 | **96.0±0.2** | **96.7±0.3** |
| MEA4 | 69.2±15.5 | 75.7±19.8 | 90.5±2.4 | 95.9±0.2 | 96.0±0.3 |
| MEA5 | 75.5±4.4 | **94.2±1.3** | **95.7±0.4** | 96.0±0.2 | 96.6±0.4 |
| AdvMEA | 47.8±27.0 | 52.5±30.7 | 33.5±28.0 | 47.0±31.9 | 38.9±16.2 |
| CEGA | **90.0±3.4** | 84.9±8.2 | 95.1±0.7 | 95.5±0.9 | 94.2±2.5 |
| Realistic | 47.0±7.1 | 33.5±13.4 | 51.8±7.0 | 59.2±19.6 | 66.4±19.5 |
| DFEA_I | 21.0±14.5 | 41.0±35.7 | 0.6±0.4 | 0.5±0.3 | 0.4±0.2 |
| DFEA_II | 14.3±4.9 | 8.5±7.0 | 10.2±7.4 | 2.5±2.4 | 0.3±0.3 |
| DFEA_III | 10.2±7.2 | 6.3±7.5 | 6.4±8.5 | 0.3±0.3 | 0.3±0.3 |

(d) Regime=data_free

| Attack | 0.05 | 0.10 | 0.25 | 0.50 | 1.00 |
|---|---|---|---|---|---|
| MEA0 | 10.4±6.9 | 28.0±13.5 | 15.8±0.8 | 16.9±0.0 | 11.7±7.5 |
| MEA1 | 10.2±7.2 | 10.2±7.2 | 10.2±7.2 | 10.2±7.2 | 10.2±7.2 |
| MEA2 | 17.7±2.5 | 17.7±2.5 | 17.7±2.5 | 17.7±2.5 | 17.7±2.5 |
| MEA3 | 21.2±19.2 | 10.8±7.5 | 5.7±6.8 | 6.1±7.7 | 15.3±0.0 |
| MEA4 | 11.3±8.0 | 11.2±7.0 | 11.0±7.3 | 11.7±7.4 | 5.9±7.8 |
| MEA5 | 32.0±21.4 | 27.1±14.2 | 10.9±6.3 | 10.4±6.4 | 22.1±19.1 |
| AdvMEA | 49.3±29.1 | 50.8±30.7 | 36.9±25.8 | 46.9±29.3 | 40.9±24.7 |
| CEGA | **51.2±32.0** | **89.9±7.7** | **95.1±0.5** | **92.7±2.8** | **92.8±1.5** |
| Realistic | 42.1±18.6 | 63.2±19.3 | 61.1±20.5 | 33.7±13.0 | 57.9±17.0 |
| DFEA_I | 0.4±0.4 | 7.0±5.8 | 3.1±4.0 | 5.3±7.1 | 10.3±7.3 |
| DFEA_II | 3.2±2.2 | 0.3±0.3 | 6.4±4.5 | 10.7±8.0 | 4.1±5.2 |
| DFEA_III | 0.3±0.3 | 0.8±0.6 | 0.3±0.3 | 3.1±3.9 | 0.3±0.3 |

Table 23: RQ1 detailed for dataset=`Photo`, metric=`F1` (%). Rows are attacks; columns are budgets. Mean ± std across seeds; best per column is bold.

(a) Regime=`both`

| Attack | 0.05 | 0.10 | 0.25 | 0.50 | 1.00 |
|---|---|---|---|---|---|
| MEA0 | 31.9±15.5 | 54.4±4.2 | 59.6±4.6 | 64.4±2.4 | 62.0±3.0 |
| MEA1 | 2.0±0.2 | 22.6±14.3 | 25.7±19.3 | 33.5±21.6 | 41.6±27.2 |
| MEA2 | 14.5±5.1 | 16.9±8.4 | 47.1±18.5 | 35.7±14.4 | 21.6±15.4 |
| MEA3 | 52.0±9.1 | **66.9±7.6** | 64.5±1.7 | **66.9±4.1** | 64.9±5.5 |
| MEA4 | 32.6±2.7 | 37.6±7.0 | 50.1±2.0 | 49.6±1.5 | 55.3±2.8 |
| MEA5 | **56.8±9.0** | 66.4±4.1 | 60.2±3.6 | 61.7±5.3 | 62.7±0.7 |
| AdvMEA | 28.5±14.3 | 25.6±14.7 | 28.6±15.2 | 20.1±16.2 | 21.8±13.3 |
| CEGA | 45.7±3.0 | 50.3±11.3 | 55.4±4.3 | 58.5±1.2 | 55.1±8.4 |
| Realistic | 15.4±4.7 | 15.9±4.8 | 28.0±18.4 | 28.6±19.9 | 17.0±5.2 |
| DFEA_I | 21.1±13.3 | 26.9±12.3 | 9.8±13.3 | 0.1±0.1 | 0.1±0.1 |
| DFEA_II | 6.2±3.3 | 6.8±3.0 | 2.2±3.0 | 3.4±2.5 | 0.1±0.1 |
| DFEA_III | 6.8±5.4 | 3.0±2.2 | 1.5±1.9 | 0.6±0.7 | 0.1±0.1 |

(b) Regime=`x_only`

| Attack | 0.05 | 0.10 | 0.25 | 0.50 | 1.00 |
|---|---|---|---|---|---|
| MEA0 | 46.4±7.7 | 59.5±6.0 | 58.4±2.4 | 66.3±2.7 | 65.7±1.0 |
| MEA1 | 2.0±0.2 | 22.7±14.5 | 25.7±19.3 | 33.5±21.6 | 41.6±27.2 |
| MEA2 | 12.9±8.9 | 9.1±5.6 | 44.5±15.2 | 42.5±11.7 | 21.4±15.3 |
| MEA3 | **54.0±9.8** | 61.5±5.0 | **63.3±5.3** | 68.0±2.2 | **69.5±4.5** |
| MEA4 | 24.8±15.5 | 43.3±4.7 | 43.0±5.1 | 60.1±4.7 | 55.9±2.4 |
| MEA5 | 48.2±17.8 | **61.7±4.6** | 60.4±3.9 | **69.9±5.0** | 62.9±2.7 |
| AdvMEA | 28.7±14.8 | 23.3±13.4 | 24.0±15.4 | 29.0±16.3 | 34.3±10.8 |
| CEGA | 48.6±8.1 | 53.1±4.3 | 45.6±7.4 | 59.3±3.3 | 58.5±3.0 |
| Realistic | 15.3±4.5 | 13.0±4.5 | 15.6±5.0 | 26.5±15.7 | 22.5±12.1 |
| DFEA_I | 14.6±3.9 | 7.2±4.9 | 5.5±4.1 | 0.2±0.1 | 0.4±0.4 |
| DFEA_II | 5.8±3.7 | 4.3±2.8 | 1.7±2.4 | 2.2±2.0 | 0.1±0.1 |
| DFEA_III | 3.1±2.1 | 2.4±3.3 | 0.1±0.1 | 0.1±0.1 | 0.1±0.1 |

(c) Regime=`a_only`

| Attack | 0.05 | 0.10 | 0.25 | 0.50 | 1.00 |
|---|---|---|---|---|---|
| MEA0 | **57.9±4.5** | **63.7±7.0** | 57.5±9.1 | 64.2±3.5 | 64.0±4.5 |
| MEA1 | 2.0±0.2 | 22.7±14.5 | 25.7±19.3 | 33.5±21.6 | 41.6±27.2 |
| MEA2 | 18.9±12.4 | 15.2±6.4 | 41.0±15.8 | 39.6±13.3 | 21.4±15.3 |
| MEA3 | 45.1±7.8 | 57.7±3.2 | 60.5±2.4 | **65.9±3.7** | 65.3±2.1 |
| MEA4 | 33.8±6.9 | 33.0±17.0 | 46.4±4.7 | 51.1±3.2 | 56.1±4.1 |
| MEA5 | 43.2±7.8 | 62.0±1.4 | **64.6±5.2** | 64.1±5.3 | **65.4±5.1** |
| AdvMEA | 29.5±14.9 | 32.1±17.1 | 19.4±17.3 | 26.8±17.8 | 21.6±3.4 |
| CEGA | 54.2±3.2 | 49.3±8.7 | 62.0±2.1 | 62.0±3.6 | 58.2±3.0 |
| Realistic | 15.0±3.0 | 10.0±5.7 | 17.1±5.6 | 26.5±17.5 | 32.6±24.6 |
| DFEA_I | 14.1±5.6 | 25.8±18.8 | 0.5±0.4 | 0.6±0.7 | 1.0±1.3 |
| DFEA_II | 9.9±0.4 | 5.6±4.3 | 5.0±3.7 | 1.9±1.8 | 0.1±0.1 |
| DFEA_III | 3.0±2.3 | 1.6±1.5 | 1.4±1.8 | 0.1±0.1 | 0.1±0.1 |

(d) Regime=`data_free`

| Attack | 0.05 | 0.10 | 0.25 | 0.50 | 1.00 |
|---|---|---|---|---|---|
| MEA0 | 2.3±1.5 | 5.3±1.9 | 3.4±0.1 | 3.6±0.0 | 2.5±1.6 |
| MEA1 | 2.2±1.6 | 2.2±1.6 | 2.2±1.6 | 2.2±1.6 | 2.2±1.6 |
| MEA2 | 5.7±0.9 | 5.7±0.9 | 5.7±0.9 | 5.7±0.9 | 5.7±0.9 |
| MEA3 | 3.9±3.2 | 2.9±0.8 | 1.6±1.6 | 1.3±1.6 | 3.3±0.0 |
| MEA4 | 2.4±1.7 | 2.4±1.5 | 2.4±1.6 | 2.5±1.6 | 1.3±1.7 |
| MEA5 | 5.8±3.1 | 7.0±2.2 | 3.1±0.3 | 2.6±1.7 | 4.0±3.2 |
| AdvMEA | 29.9±15.9 | 30.8±17.4 | 22.9±14.2 | 29.2±19.0 | 23.3±11.7 |
| CEGA | **31.0±17.9** | **55.3±6.1** | **62.7±1.6** | **59.0±3.2** | **60.0±4.9** |
| Realistic | 13.7±4.1 | 28.6±18.2 | 27.0±17.7 | 10.3±6.3 | 24.6±15.1 |
| DFEA_I | 0.2±0.2 | 3.8±2.8 | 4.0±5.1 | 2.6±3.0 | 5.7±5.0 |
| DFEA_II | 2.9±2.0 | 0.1±0.1 | 5.2±4.0 | 6.4±5.1 | 1.4±1.7 |
| DFEA_III | 0.1±0.1 | 0.5±0.6 | 0.1±0.1 | 2.5±3.4 | 0.1±0.1 |

Table 24: RQ1 detailed for dataset=`Photo`, metric=`Fidelity` (%). Rows are attacks; columns are budgets. Mean ± std across seeds; best per column is bold.

(a) Regime=`both`

| Attack | 0.05 | 0.10 | 0.25 | 0.50 | 1.00 |
|---|---|---|---|---|---|
| MEA0 | 54.7±30.8 | 90.8±4.1 | 95.4±1.4 | 96.6±1.1 | 96.8±0.1 |
| MEA1 | 1.9±0.3 | 39.2±26.6 | 42.0±29.2 | 62.1±33.8 | 67.0±37.2 |
| MEA2 | 20.7±2.0 | 26.3±14.1 | 77.2±21.3 | 64.4±20.1 | 35.4±31.7 |
| MEA3 | 78.3±15.0 | **93.9±3.0** | **96.5±0.8** | **96.9±0.9** | **97.8±1.0** |
| MEA4 | 74.3±2.6 | 83.1±8.2 | 95.2±0.4 | 96.7±1.1 | 97.4±0.8 |
| MEA5 | 81.1±6.6 | 92.4±0.8 | 95.0±2.3 | 95.9±0.9 | 97.5±0.6 |
| AdvMEA | 49.5±22.9 | 38.6±24.6 | 48.0±27.4 | 37.3±26.7 | 41.3±24.7 |
| CEGA | **80.6±5.0** | 83.7±15.1 | 91.4±6.8 | 95.9±1.2 | 90.4±9.5 |
| Realistic | 70.7±15.3 | 72.5±13.9 | 83.6±10.3 | 86.9±5.1 | 77.5±11.1 |
| DFEA_I | 29.3±19.3 | 33.3±20.8 | 9.1±11.9 | 0.5±0.3 | 0.5±0.4 |
| DFEA_II | 13.1±7.5 | 9.6±6.1 | 2.5±2.8 | 6.2±3.8 | 0.5±0.4 |
| DFEA_III | 12.1±6.7 | 11.5±7.4 | 6.1±7.6 | 0.7±0.2 | 0.5±0.4 |

(b) Regime=`x_only`

| Attack | 0.05 | 0.10 | 0.25 | 0.50 | 1.00 |
|---|---|---|---|---|---|
| MEA0 | 84.2±5.8 | 91.0±3.3 | 96.0±0.8 | 96.8±1.0 | 96.6±1.2 |
| MEA1 | 1.9±0.3 | 39.3±26.8 | 42.0±29.2 | 62.1±33.8 | 67.0±37.2 |
| MEA2 | 27.7±21.7 | 19.1±12.1 | 69.7±19.9 | 75.8±13.7 | 35.0±31.7 |
| MEA3 | 82.4±5.2 | 92.6±0.8 | 96.1±0.7 | 96.3±1.1 | 96.7±0.8 |
| MEA4 | 67.3±18.7 | 85.8±1.8 | 88.5±7.8 | **97.0±0.9** | 97.3±0.9 |
| MEA5 | 81.9±13.2 | **94.0±1.1** | **96.2±0.7** | 96.9±1.4 | **97.8±0.8** |
| AdvMEA | 49.5±30.4 | 40.3±24.6 | 43.1±26.5 | 44.7±25.5 | 57.6±17.4 |
| CEGA | **85.5±7.2** | 86.6±5.8 | 83.7±7.5 | 94.8±2.3 | 95.5±2.9 |
| Realistic | 76.1±11.1 | 53.1±24.5 | 72.3±16.7 | 81.6±9.4 | 85.5±5.4 |
| DFEA_I | 20.3±9.6 | 11.7±6.8 | 8.5±6.3 | 0.7±0.2 | 0.6±0.3 |
| DFEA_II | 11.3±6.2 | 8.0±4.9 | 2.0±2.1 | 3.3±3.1 | 0.5±0.4 |
| DFEA_III | 6.9±6.5 | 3.9±4.5 | 0.5±0.4 | 0.5±0.4 | 0.5±0.4 |

(c) Regime=`a_only`

| Attack | 0.05 | 0.10 | 0.25 | 0.50 | 1.00 |
|---|---|---|---|---|---|
| MEA0 | 87.3±5.7 | 92.4±1.1 | 94.3±2.3 | 96.5±0.9 | 97.5±0.9 |
| MEA1 | 1.9±0.3 | 39.1±26.5 | 42.0±29.2 | 62.1±33.8 | 67.0±37.2 |
| MEA2 | 17.5±10.3 | 21.1±4.9 | 74.7±17.0 | 70.2±17.6 | 35.2±31.8 |
| MEA3 | 81.4±3.8 | 92.4±2.3 | 95.8±1.0 | 96.8±0.8 | 97.6±0.7 |
| MEA4 | 69.0±16.2 | 76.0±19.4 | 90.9±2.0 | 96.1±0.6 | **98.2±0.5** |
| MEA5 | 75.2±4.1 | **94.2±1.4** | **95.9±0.5** | 96.8±0.7 | 97.7±1.1 |
| AdvMEA | 48.3±26.7 | 53.1±30.6 | 34.1±28.0 | 47.7±31.8 | 39.1±6.0 |
| CEGA | **90.6±3.9** | 85.5±8.8 | 95.4±1.1 | **97.3±0.8** | 96.4±1.8 |
| Realistic | 69.3±14.6 | 62.2±32.2 | 78.4±11.1 | 84.9±7.0 | 85.2±8.0 |
| DFEA_I | 21.3±14.3 | 41.5±35.7 | 0.7±0.2 | 0.7±0.2 | 0.6±0.3 |
| DFEA_II | 13.9±4.0 | 8.8±6.7 | 10.4±7.1 | 2.6±2.2 | 0.5±0.4 |
| DFEA_III | 10.1±6.4 | 6.5±7.0 | 6.9±8.7 | 0.5±0.4 | 0.5±0.4 |

(d) Regime=`data_free`

| Attack | 0.05 | 0.10 | 0.25 | 0.50 | 1.00 |
|---|---|---|---|---|---|
| MEA0 | 10.3±7.0 | 28.3±13.4 | 15.6±0.6 | 16.2±0.6 | 11.6±7.6 |
| MEA1 | 10.3±6.6 | 10.3±6.6 | 10.3±6.6 | 10.3±6.6 | 10.3±6.6 |
| MEA2 | 18.0±2.8 | 18.0±2.8 | 18.0±2.8 | 18.0±2.8 | 18.0±2.8 |
| MEA3 | 21.1±19.3 | 10.3±7.2 | 5.3±7.2 | 6.2±7.2 | 14.9±0.5 |
| MEA4 | 11.1±8.0 | 10.7±6.7 | 10.0±6.9 | 10.9±7.0 | 5.9±7.4 |
| MEA5 | 31.9±21.7 | 26.9±14.6 | 10.1±6.4 | 10.3±6.5 | 22.2±19.2 |
| AdvMEA | 50.3±29.2 | 51.6±30.7 | 37.6±25.6 | 47.6±29.1 | 41.1±24.3 |
| CEGA | 51.7±32.2 | **91.0±7.8** | **95.9±1.0** | **93.8±2.4** | **94.2±1.4** |
| Realistic | **70.4±15.5** | 86.8±4.3 | 86.0±5.6 | 61.8±31.4 | 85.0±5.8 |
| DFEA_I | 0.6±0.3 | 7.1±5.3 | 3.2±3.6 | 5.0±6.3 | 10.5±6.9 |
| DFEA_II | 3.3±1.6 | 0.5±0.4 | 6.5±4.0 | 11.1±7.8 | 4.3±5.0 |
| DFEA_III | 0.5±0.4 | 0.9±0.3 | 0.5±0.4 | 3.3±3.7 | 0.5±0.4 |

Table 25: RQ1 detailed for dataset=PubMed, metric=Acc (%). Rows are attacks; columns are budgets. Mean ± std across seeds; best per column is bold.

(a) Regime=both

| Attack | 0.05 | 0.10 | 0.25 | 0.50 | 1.00 |
|---|---|---|---|---|---|
| MEA0 | 70.6±1.5 | 75.8±1.8 | 77.4±0.3 | 76.7±0.4 | 78.2±0.3 |
| MEA1 | 68.6±2.3 | 70.6±5.0 | 74.8±3.4 | 75.3±3.5 | 75.5±3.9 |
| MEA2 | 40.6±9.4 | 41.6±13.4 | 62.6±10.2 | 64.9±5.3 | 41.6±0.8 |
| MEA3 | 72.1±1.9 | 73.4±1.6 | 76.6±0.7 | 79.0±1.1 | 77.3±1.7 |
| MEA4 | 67.8±0.6 | 71.2±3.8 | 75.3±0.9 | 78.2±0.5 | 78.6±0.8 |
| MEA5 | 72.3±3.4 | 75.9±0.1 | 77.8±0.8 | 77.8±0.7 | 78.1±1.1 |
| AdvMEA | 60.4±3.2 | 66.5±1.9 | 65.5±2.6 | 62.6±6.7 | 66.0±2.6 |
| CEGA | **78.3±0.4** | **77.9±1.6** | **78.2±0.2** | **79.1±0.2** | **78.7±0.7** |
| Realistic | 72.7±1.5 | 71.5±1.0 | 74.7±0.2 | 75.6±1.8 | 76.8±0.8 |
| DFEA_I | 64.5±8.9 | 70.0±1.0 | 50.0±6.7 | 42.7±2.2 | 40.7±0.0 |
| DFEA_II | 55.2±3.1 | 52.2±5.1 | 47.5±4.4 | 51.4±5.1 | 40.7±0.0 |
| DFEA_III | 63.7±4.4 | 58.4±13.6 | 40.7±0.0 | 40.7±0.0 | 40.7±0.0 |

(b) Regime=x_only

| Attack | 0.05 | 0.10 | 0.25 | 0.50 | 1.00 |
|---|---|---|---|---|---|
| MEA0 | 72.1±3.2 | 76.8±1.1 | 76.8±1.7 | 78.0±0.5 | 78.5±0.3 |
| MEA1 | 68.6±2.3 | 70.6±5.0 | 74.8±3.4 | 75.3±3.5 | 75.5±3.9 |
| MEA2 | 39.6±12.1 | 51.2±19.5 | 65.0±6.9 | 63.6±6.5 | 43.9±4.1 |
| MEA3 | 73.2±1.1 | 75.4±1.4 | 77.2±0.6 | 77.4±0.7 | 78.0±0.6 |
| MEA4 | 51.5±4.7 | 65.7±6.3 | 77.0±1.3 | 77.4±0.9 | 78.6±0.8 |
| MEA5 | 70.6±2.8 | 74.5±1.0 | 77.0±1.4 | 77.7±0.5 | 77.5±0.9 |
| AdvMEA | 64.3±9.4 | 64.2±5.4 | 64.0±6.3 | 62.5±3.1 | 61.6±3.1 |
| CEGA | **74.8±1.2** | **78.7±0.7** | **78.7±0.4** | **79.1±0.2** | **78.9±0.3** |
| Realistic | 67.1±1.7 | 74.2±0.6 | 76.7±1.0 | 75.7±0.4 | 76.0±0.8 |
| DFEA_I | 68.0±3.6 | 67.1±4.5 | 55.3±10.3 | 46.4±8.1 | 40.8±0.2 |
| DFEA_II | 49.3±5.0 | 55.5±2.5 | 45.2±5.3 | 47.1±9.0 | 40.7±0.0 |
| DFEA_III | 54.0±10.6 | 64.7±5.3 | 54.2±9.0 | 40.7±0.0 | 40.7±0.0 |

(c) Regime=a_only

| Attack | 0.05 | 0.10 | 0.25 | 0.50 | 1.00 |
|---|---|---|---|---|---|
| MEA0 | 71.1±0.9 | 76.3±0.6 | 77.3±1.0 | 78.4±0.5 | 78.3±0.6 |
| MEA1 | 68.6±2.3 | 70.6±5.0 | 74.8±3.4 | 75.3±3.5 | 75.5±3.9 |
| MEA2 | 36.9±12.0 | 40.7±14.3 | 67.0±4.9 | 64.4±4.2 | 43.1±3.0 |
| MEA3 | 70.7±3.4 | 73.8±1.3 | 77.7±0.4 | 77.6±0.4 | **79.1±0.5** |
| MEA4 | 54.4±7.9 | 71.4±0.5 | 76.6±1.2 | 78.0±1.0 | 78.8±1.0 |
| MEA5 | 72.1±3.5 | 76.9±1.0 | 77.6±1.1 | 77.8±0.9 | 77.8±0.9 |
| AdvMEA | 59.3±7.4 | 63.2±6.2 | 62.5±5.0 | 63.6±2.0 | 63.8±2.6 |
| CEGA | **74.9±2.4** | **77.7±1.3** | **78.3±0.5** | **78.9±0.5** | 78.8±0.3 |
| Realistic | 69.6±1.5 | 72.7±0.8 | 74.9±0.6 | 76.5±0.9 | 76.6±0.4 |
| DFEA_I | 57.9±8.7 | 63.8±2.3 | 55.6±10.6 | 42.8±1.5 | 40.8±0.2 |
| DFEA_II | 50.1±7.5 | 51.0±7.3 | 50.9±7.8 | 48.2±5.0 | 40.7±0.0 |
| DFEA_III | 54.1±5.4 | 55.0±6.1 | 48.4±10.9 | 40.7±0.0 | 40.7±0.0 |

(d) Regime=data_free

| Attack | 0.05 | 0.10 | 0.25 | 0.50 | 1.00 |
|---|---|---|---|---|---|
| MEA0 | 33.1±10.7 | 41.3±0.0 | 40.9±0.3 | 40.9±0.3 | 40.9±0.3 |
| MEA1 | 41.3±0.0 | 41.3±0.0 | 41.3±0.0 | 41.3±0.0 | 41.3±0.0 |
| MEA2 | 23.3±7.4 | 23.3±7.4 | 23.2±7.4 | 23.3±7.4 | 23.3±7.4 |
| MEA3 | 38.3±3.8 | 33.3±10.8 | 33.5±11.0 | 40.9±0.3 | 40.9±0.3 |
| MEA4 | 40.7±0.0 | 33.1±10.7 | 33.3±10.8 | 33.3±10.8 | 41.1±0.3 |
| MEA5 | 18.1±0.1 | 40.9±0.3 | 26.0±10.4 | 25.6±10.7 | 41.2±0.4 |
| AdvMEA | 62.1±6.5 | 66.9±1.9 | 65.3±4.1 | 62.0±6.2 | 62.7±3.6 |
| CEGA | 75.7±2.4 | **78.0±1.1** | **78.6±0.4** | **78.5±0.4** | **78.6±0.3** |
| Realistic | **76.1±1.2** | 77.2±0.4 | 76.6±0.5 | 76.2±0.8 | 75.9±0.5 |
| DFEA_I | 40.7±0.0 | 41.0±0.4 | 40.7±0.0 | 40.7±0.0 | 40.8±0.1 |
| DFEA_II | 40.7±0.0 | 45.6±6.9 | 40.7±0.0 | 41.8±1.5 | 40.7±0.0 |
| DFEA_III | 40.7±0.0 | 40.7±0.0 | 40.7±0.0 | 40.7±0.0 | 40.7±0.0 |

Table 26: RQ1 detailed for dataset=PubMed, metric=F1 (%). Rows are attacks; columns are budgets. Mean ± std across seeds; best per column is bold.

(a) Regime=both

| Attack | 0.05 | 0.10 | 0.25 | 0.50 | 1.00 |
|---|---|---|---|---|---|
| MEA0 | 68.2±2.8 | 74.7±1.4 | 76.3±0.2 | 76.0±0.5 | 77.2±0.4 |
| MEA1 | 50.2±1.7 | 68.3±6.2 | 74.0±3.5 | 74.6±3.5 | 74.9±3.9 |
| MEA2 | 28.5±5.8 | 32.2±11.0 | 61.6±10.3 | 58.3±9.6 | 24.9±7.9 |
| MEA3 | 70.3±2.3 | 72.2±2.0 | 75.4±0.5 | 78.0±0.9 | 76.3±1.6 |
| MEA4 | 64.4±3.3 | 66.2±7.9 | 73.8±0.7 | 76.8±0.6 | 77.0±0.7 |
| MEA5 | 71.3±3.0 | 74.1±0.5 | 76.5±0.5 | 76.7±0.5 | 77.1±1.2 |
| AdvMEA | 55.6±1.7 | 65.8±2.1 | 64.8±2.6 | 60.9±8.1 | 63.9±1.7 |
| CEGA | **77.3±0.3** | **76.9±1.5** | **77.4±0.1** | **78.3±0.1** | **77.7±0.6** |
| Realistic | 71.9±1.0 | 70.6±1.2 | 73.9±0.1 | 74.6±1.7 | 76.2±0.7 |
| DFEA_I | 59.2±14.3 | 69.2±1.8 | 32.6±9.6 | 25.2±4.2 | 19.3±0.0 |
| DFEA_II | 44.3±8.5 | 42.3±6.1 | 32.1±8.1 | 35.0±6.2 | 19.3±0.0 |
| DFEA_III | 59.0±6.0 | 48.0±22.2 | 19.3±0.0 | 19.3±0.0 | 19.3±0.0 |

(b) Regime=x_only

| Attack | 0.05 | 0.10 | 0.25 | 0.50 | 1.00 |
|---|---|---|---|---|---|
| MEA0 | 70.7±4.1 | 75.7±0.8 | 75.9±1.4 | 77.2±0.4 | 77.5±0.6 |
| MEA1 | 50.2±1.7 | 68.3±6.2 | 74.0±3.5 | 74.6±3.5 | 74.9±3.9 |
| MEA2 | 27.6±7.7 | 48.3±20.8 | 63.6±6.4 | 57.4±9.5 | 26.5±10.0 |
| MEA3 | 71.6±1.3 | 74.1±1.4 | 75.7±0.8 | 76.3±0.7 | 76.9±0.9 |
| MEA4 | 46.5±5.0 | 62.1±7.7 | 75.3±1.6 | 75.8±0.8 | 77.0±0.8 |
| MEA5 | 68.2±2.9 | 73.0±0.5 | 75.5±2.1 | 76.5±0.3 | 76.6±0.8 |
| AdvMEA | 62.3±11.3 | 63.1±5.7 | 61.3±6.6 | 59.8±2.0 | 60.4±2.9 |
| CEGA | **74.4±0.9** | **77.8±0.6** | **77.8±0.4** | **78.2±0.2** | **78.1±0.3** |
| Realistic | 66.1±1.2 | 72.6±0.3 | 76.1±0.8 | 75.3±0.4 | 75.4±0.7 |
| DFEA_I | 64.7±3.8 | 60.2±11.0 | 37.0±12.5 | 26.8±10.7 | 19.6±0.4 |
| DFEA_II | 37.1±9.6 | 45.1±4.9 | 26.7±7.6 | 27.7±11.2 | 19.3±0.0 |
| DFEA_III | 45.6±15.0 | 56.5±11.4 | 36.6±10.5 | 19.3±0.0 | 19.3±0.0 |

(c) Regime=a_only

| Attack | 0.05 | 0.10 | 0.25 | 0.50 | 1.00 |
|---|---|---|---|---|---|
| MEA0 | 68.6±2.2 | 75.3±0.5 | 76.0±1.3 | 77.4±0.5 | 77.5±0.4 |
| MEA1 | 50.2±1.7 | 68.3±6.2 | 74.0±3.5 | 74.6±3.5 | 74.9±3.9 |
| MEA2 | 24.9±8.4 | 28.3±10.0 | 66.1±5.2 | 57.4±7.2 | 25.9±9.2 |
| MEA3 | 68.8±5.0 | 72.9±1.2 | 76.4±0.6 | 76.4±0.3 | **78.0±0.5** |
| MEA4 | 43.5±15.2 | 69.3±0.5 | 74.9±1.6 | 76.5±1.1 | 77.2±1.0 |
| MEA5 | 69.1±3.1 | 75.9±1.0 | 76.5±1.1 | 76.8±0.8 | 76.9±0.8 |
| AdvMEA | 56.1±10.3 | 60.4±7.9 | 58.8±5.1 | 62.0±2.4 | 61.8±1.8 |
| CEGA | **74.0±2.1** | **77.1±1.2** | **77.4±0.5** | **77.8±0.5** | 77.8±0.3 |
| Realistic | 69.0±0.9 | 71.2±1.9 | 74.6±0.6 | 75.7±0.9 | 76.1±0.3 |
| DFEA_I | 48.4±12.1 | 59.9±3.0 | 37.3±12.8 | 23.4±3.0 | 19.6±0.4 |
| DFEA_II | 35.0±12.2 | 36.5±11.0 | 33.3±10.2 | 30.7±7.4 | 19.3±0.0 |
| DFEA_III | 42.9±4.9 | 41.3±3.5 | 28.5±13.1 | 19.3±0.0 | 19.3±0.0 |

(d) Regime=data_free

| Attack | 0.05 | 0.10 | 0.25 | 0.50 | 1.00 |
|---|---|---|---|---|---|
| MEA0 | 16.2±4.3 | 19.5±0.0 | 19.4±0.1 | 19.4±0.1 | 19.4±0.1 |
| MEA1 | 19.5±0.0 | 19.5±0.0 | 19.5±0.0 | 19.5±0.0 | 19.5±0.0 |
| MEA2 | 15.3±7.2 | 15.3±7.2 | 15.3±7.2 | 15.3±7.2 | 15.3±7.2 |
| MEA3 | 20.2±1.2 | 16.9±4.8 | 16.4±4.4 | 19.4±0.1 | 19.4±0.1 |
| MEA4 | 19.3±0.0 | 16.2±4.3 | 16.3±4.3 | 16.3±4.3 | 19.4±0.1 |
| MEA5 | 10.3±0.2 | 19.4±0.1 | 14.0±3.9 | 13.3±4.2 | 20.2±1.2 |
| AdvMEA | 59.5±8.3 | 64.7±1.6 | 63.6±5.3 | 60.1±8.6 | 61.3±3.4 |
| CEGA | 75.0±2.2 | **77.3±1.1** | **77.8±0.4** | **77.4±0.4** | **77.7±0.3** |
| Realistic | **75.4±0.4** | 76.6±0.4 | 75.9±0.6 | 75.7±0.7 | 75.4±0.5 |
| DFEA_I | 19.3±0.1 | 22.2±4.1 | 19.3±0.0 | 19.3±0.0 | 22.8±5.0 |
| DFEA_II | 19.3±0.0 | 26.3±9.9 | 19.3±0.0 | 21.2±2.7 | 19.3±0.1 |
| DFEA_III | 19.3±0.0 | 19.3±0.0 | 19.3±0.0 | 19.3±0.0 | 19.3±0.0 |

Table 27: RQ1 detailed for dataset=PubMed, metric=Fidelity (%). Rows are attacks; columns are budgets. Mean ± std across seeds; best per column is bold.

(a) Regime=both

| Attack | 0.05 | 0.10 | 0.25 | 0.50 | 1.00 |
|---|---|---|---|---|---|
| MEA0 | 80.0±0.6 | 87.1±2.4 | 90.6±1.8 | 92.9±0.8 | **94.2±0.6** |
| MEA1 | 72.9±2.0 | 79.0±6.1 | 81.9±3.4 | 83.2±4.5 | 85.3±4.6 |
| MEA2 | 39.5±10.2 | 43.4±15.6 | 68.3±13.2 | 73.6±8.9 | 41.9±3.0 |
| MEA3 | 80.8±2.1 | 83.3±1.5 | 87.0±1.4 | 89.6±0.7 | 89.7±1.3 |
| MEA4 | 71.6±1.9 | 77.7±4.1 | 83.5±1.1 | 87.7±0.9 | 89.1±0.9 |
| MEA5 | 82.2±4.6 | 86.4±1.6 | 89.3±1.3 | 90.6±1.1 | 90.9±0.9 |
| AdvMEA | 63.5±6.4 | 70.6±3.6 | 69.5±4.2 | 65.1±8.9 | 69.8±4.5 |
| CEGA | **89.5±0.4** | **89.9±1.3** | **92.6±0.3** | **94.4±0.4** | 94.2±0.3 |
| Realistic | 81.2±1.3 | 83.5±1.1 | 88.1±1.0 | 89.5±1.6 | 91.4±0.8 |
| DFEA_I | 68.6±12.3 | 79.6±1.2 | 50.7±9.3 | 41.7±2.7 | 37.7±0.5 |
| DFEA_II | 55.9±3.5 | 54.1±7.9 | 48.4±7.0 | 51.8±7.7 | 37.7±0.5 |
| DFEA_III | 70.6±4.4 | 62.5±17.7 | 37.7±0.5 | 37.7±0.5 | 37.7±0.5 |

(b) Regime=x_only

| Attack | 0.05 | 0.10 | 0.25 | 0.50 | 1.00 |
|---|---|---|---|---|---|
| MEA0 | 82.9±0.9 | 87.5±1.5 | 91.7±0.7 | 92.6±0.3 | 93.3±1.1 |
| MEA1 | 72.9±2.0 | 79.0±6.1 | 81.9±3.4 | 83.2±4.5 | 85.3±4.6 |
| MEA2 | 41.6±13.7 | 55.0±22.8 | 73.1±10.4 | 71.5±10.0 | 43.9±4.5 |
| MEA3 | 82.8±2.2 | 84.8±2.4 | 88.7±0.6 | 88.7±2.3 | 90.8±0.8 |
| MEA4 | 52.8±4.6 | 72.6±5.9 | 86.0±1.4 | 87.2±1.4 | 89.2±0.9 |
| MEA5 | 80.0±3.1 | 84.7±2.4 | 88.0±0.8 | 89.9±0.8 | 90.4±0.5 |
| AdvMEA | 67.8±11.8 | 66.1±8.4 | 65.7±8.6 | 65.2±6.3 | 64.1±5.7 |
| CEGA | **85.4±2.4** | **91.0±0.9** | **93.5±0.4** | **93.5±1.3** | **95.3±0.7** |
| Realistic | 77.6±3.0 | 83.8±2.8 | 88.1±2.9 | 90.8±0.9 | 91.7±0.4 |
| DFEA_I | 75.4±2.7 | 75.8±3.6 | 59.5±15.1 | 46.3±11.7 | 37.9±0.3 |
| DFEA_II | 49.6±7.1 | 58.0±3.8 | 44.8±7.9 | 47.4±13.0 | 37.7±0.5 |
| DFEA_III | 57.0±12.8 | 71.7±6.4 | 57.5±12.0 | 37.7±0.5 | 37.7±0.5 |

(c) Regime=a_only

| Attack | 0.05 | 0.10 | 0.25 | 0.50 | 1.00 |
|---|---|---|---|---|---|
| MEA0 | 79.6±1.8 | 89.5±0.4 | 92.7±0.9 | 91.8±0.5 | 93.4±0.6 |
| MEA1 | 72.9±2.0 | 79.0±6.1 | 81.9±3.4 | 83.2±4.5 | 85.3±4.6 |
| MEA2 | 38.2±13.9 | 42.3±17.9 | 73.6±5.1 | 74.2±7.5 | 43.5±4.1 |
| MEA3 | 79.1±2.3 | 84.5±1.3 | 88.1±0.9 | 89.3±0.4 | 90.7±0.2 |
| MEA4 | 59.3±7.1 | 78.5±1.7 | 84.9±1.5 | 86.8±1.5 | 89.2±0.9 |
| MEA5 | 80.9±2.2 | 86.6±1.4 | 89.4±0.4 | 89.8±0.8 | 90.6±0.4 |
| AdvMEA | 59.0±9.8 | 66.2±9.0 | 66.4±8.2 | 69.1±3.4 | 65.6±4.6 |
| CEGA | **84.7±2.2** | **91.8±1.3** | **93.2±0.4** | **93.0±1.2** | **94.8±0.5** |
| Realistic | 79.7±2.2 | 82.7±2.3 | 89.0±0.7 | 91.0±1.3 | 92.4±1.6 |
| DFEA_I | 61.5±11.6 | 72.1±3.5 | 57.9±14.8 | 40.5±1.6 | 37.9±0.3 |
| DFEA_II | 52.2±9.5 | 50.6±9.3 | 52.2±10.2 | 47.5±6.3 | 37.7±0.5 |
| DFEA_III | 57.4±10.2 | 58.9±10.7 | 48.2±14.3 | 37.7±0.5 | 37.7±0.5 |

(d) Regime=data_free

| Attack | 0.05 | 0.10 | 0.25 | 0.50 | 1.00 |
|---|---|---|---|---|---|
| MEA0 | 30.7±9.8 | 45.5±0.5 | 40.0±3.6 | 40.0±3.6 | 40.0±3.6 |
| MEA1 | 45.5±0.5 | 45.5±0.5 | 45.5±0.5 | 45.5±0.5 | 45.5±0.5 |
| MEA2 | 21.4±6.5 | 21.4±6.5 | 21.4±6.5 | 21.4±6.5 | 21.4±6.5 |
| MEA3 | 37.6±6.1 | 32.8±11.7 | 36.0±13.6 | 40.0±3.6 | 40.0±3.6 |
| MEA4 | 37.7±0.5 | 31.0±10.0 | 33.7±12.4 | 33.4±12.1 | 43.2±3.6 |
| MEA5 | 16.9±0.1 | 40.0±3.6 | 24.4±9.8 | 23.6±9.5 | 43.4±3.7 |
| AdvMEA | 63.9±8.6 | 70.4±2.9 | 69.9±6.7 | 64.5±8.3 | 64.1±6.5 |
| CEGA | 87.2±2.8 | 90.9±1.0 | **93.4±0.6** | **93.8±0.3** | **94.4±0.2** |
| Realistic | **92.2±1.3** | **92.3±0.2** | 92.1±1.0 | 91.4±0.9 | 92.1±1.3 |
| DFEA_I | 37.8±0.4 | 39.4±2.0 | 37.7±0.5 | 37.7±0.5 | 40.4±3.3 |
| DFEA_II | 37.7±0.5 | 45.3±10.8 | 37.7±0.5 | 38.9±1.9 | 37.8±0.5 |
| DFEA_III | 37.7±0.5 | 37.7±0.5 | 37.7±0.5 | 37.7±0.5 | 37.7±0.5 |

| Dataset | backdoorwm | randomwm | survivewm | imperceptiblewm | integrity |
|---|---|---|---|---|---|
| Cora | 79.37 ± 0.45 | 76.06 ± 0.81 | 79.07 ± 0.33 | 71.94 ± 0.27 | 75.39 ± 0.36 |
| CiteSeer | 64.74 ± 0.35 | 64.99 ± 0.69 | 67.47 ± 0.27 | 58.07 ± 0.79 | 57.61 ± 0.38 |
| PubMed | 77.00 ± 0.59 | 73.77 ± 0.26 | 25.32 ± 1.19 | 76.00 ± 0.17 | 75.68 ± 0.37 |
| Computers | 55.39 ± 2.89 | 58.49 ± 2.93 | 37.62 ± 24.96 | 67.04 ± 1.66 | 14.08 ± 6.63 |
| Photo | 57.47 ± 2.27 | 52.15 ± 5.30 | 58.47 ± 3.84 | 61.90 ± 0.43 | 23.47 ± 21.57 |
| CoauthorCS | 69.13 ± 1.21 | 66.09 ± 6.69 | 72.75 ± 0.46 | 72.52 ± 0.73 | 73.43 ± 0.32 |
| CoauthorPhysics | 76.23 ± 1.03 | 57.32 ± 12.15 | 81.11 ± 0.89 | 80.50 ± 0.75 | 80.91 ± 0.81 |

Table 28: Defense results: F1 (mean ± std, in %).

| Dataset | backdoorwm | randomwm | survivewm | imperceptiblewm | integrity |
|---|---|---|---|---|---|
| Cora | 80.07 ± 0.40 | 76.13 ± 1.17 | 79.93 ± 0.26 | 71.97 ± 0.31 | 76.03 ± 0.37 |
| CiteSeer | 67.47 ± 0.40 | 67.90 ± 0.57 | 70.87 ± 0.12 | 60.30 ± 1.00 | 60.10 ± 0.29 |
| PubMed | 77.60 ± 0.79 | 74.13 ± 0.26 | 39.17 ± 1.01 | 76.37 ± 0.12 | 76.33 ± 0.38 |
| Computers | 68.27 ± 2.74 | 71.47 ± 2.49 | 45.97 ± 31.21 | 78.90 ± 0.73 | 11.97 ± 5.56 |
| Photo | 92.73 ± 1.50 | 85.33 ± 6.07 | 93.20 ± 3.89 | 96.70 ± 0.08 | 61.80 ± 20.58 |
| CoauthorCS | 88.43 ± 0.77 | 89.27 ± 2.46 | 91.70 ± 0.36 | 89.63 ± 0.76 | 90.97 ± 0.12 |
| CoauthorPhysics | 89.33 ± 0.37 | 68.60 ± 10.96 | 90.97 ± 0.33 | 90.20 ± 0.40 | 90.60 ± 0.65 |

Table 29: Defense results: Fidelity (mean ± std, in %).

| Dataset | backdoorwm | randomwm | survivewm | imperceptiblewm | integrity |
|---|---|---|---|---|---|
| Cora | 100.00 ± 0.00 | 75.33 ± 9.98 | 54.07 ± 5.79 | 100.00 ± 0.00 | 100.00 ± 0.00 |
| CiteSeer | 100.00 ± 0.00 | 72.00 ± 4.32 | 55.72 ± 4.51 | 100.00 ± 0.00 | 100.00 ± 0.00 |
| PubMed | 100.00 ± 0.00 | 64.00 ± 11.31 | 34.97 ± 0.71 | 100.00 ± 0.00 | 100.00 ± 0.00 |
| Computers | 93.33 ± 9.43 | 94.67 ± 1.89 | 11.08 ± 0.81 | 100.00 ± 0.00 | 100.00 ± 100.00 |
| Photo | 100.00 ± 0.00 | 98.67 ± 0.94 | 13.25 ± 1.10 | 100.00 ± 0.00 | 66.67 ± 47.14 |
| CoauthorCS | 100.00 ± 0.00 | 45.33 ± 3.77 | 8.51 ± 0.34 | 100.00 ± 0.00 | 33.33 ± 47.14 |
| CoauthorPhysics | 100.00 ± 0.00 | 56.67 ± 13.70 | 21.76 ± 0.49 | 100.00 ± 0.00 | 66.67 ± 47.14 |

Table 30: Defense results: Owner. verif. (WM Acc, %) (mean ± std, in %).

