# OpenReview forum: "GraphIP–Bench: A Benchmark for Extraction Attacks and Defenses in Graph Learning"
_ICLR.cc/2026/Conference — Submitted to ICLR 2026_

### Official Review · Reviewer_wKvN · 2025-10-30

**Soundness:** 3
**Presentation:** 3
**Contribution:** 3
**Rating:** 6
**Confidence:** 3

**Summary:**

GraphIP-Bench is the first unified benchmark for evaluating model extraction attacks and defenses in GNNs, addressing the lack of standardized experimental protocols in prior research. It integrates diverse extraction attacks, defense families, and multiple public datasets with standardized query budgets, threat models, and metrics (fidelity, utility, and computational cost). The empirical results comprehensively compare the strengths and weaknesses of various models. The benchmark provides a reproducible infrastructure and a clear characterization of trade-offs to guide practical deployment.

**Strengths:**

1.Fixes public data splits, query sets, and budgets, enabling fair and standardized comparison across different methods.

2.Covers diverse attack/defense paradigms, data availability settings, and evaluation dimensions (security, utility, and efficiency).

3.Releases reference implementations, fixed seeds, and unified hardware/software configurations. It also reports computational cost and protection–utility trade-offs, facilitating real-world adoption.

**Weaknesses:**

1.The benchmark focuses mainly on ownership tracing and information-limiting defenses; it could be expanded to include other defense types, such as differential privacy.

2.As the authors mentioned the use of molecular property prediction in the main text, it would be beneficial to include more realistic and practically relevant tasks—the current datasets are largely academic citation networks, which are publicly available and thus do not realistically require extraction protection.

3.It might be valuable to include heterogeneous graphs, as this could reveal how structural diversity affects extraction and defense performance.

**Questions:**

See weakness.

---

> ### Author Response · Authors · 2025-11-24
>
> We thank the reviewer for the helpful feedback. Below we answer each question.
>
> ---
>
> ### Q1 – Scope of defenses and differential privacy
>
> Differential privacy (DP) is not the main defense paradigm in current model extraction work, and our defense choices follow what is actually used in this literature. Most model extraction papers study IP or functionality theft under non-DP settings [1,2]. DP is usually mentioned only as a theoretical knob and is rarely instantiated as a practical defense in their experiments. Recent GNN-specific extraction papers [3,4] likewise evaluate attacks on standard GNN services without DP training or DP inference, and focus instead on watermarking/fingerprinting and output manipulation as realistic options for GNN IP protection. In parallel, DP work mainly targets training-data privacy (e.g., membership or property inference) and typically requires retraining with DP-SGD or strong graph perturbations, which does not match our goal of benchmarking plug-in IP defenses under fixed query budgets. For these reasons, we believe it is appropriate that GraphIP-Bench concentrates on watermarking/fingerprinting and integrity-style defenses, which are the primary and most practical defense families in current model extraction research.
>
> ---
>
> ### Q2 – Dataset selection
>
> GraphIP-Bench is a **model extraction** benchmark, not a data-extraction benchmark. The attacker aims to steal the *behavior* of a deployed GNN service (its decision function), not to reconstruct private training examples. Even if the underlying graph is public (e.g., citation networks), a production GNN can still contain proprietary IP in its architecture, feature engineering, loss design, and training procedure. For model-extraction risk, the key question is whether a black-box service can be accurately copied via queries, not whether the raw graph is public or private.
>
> Our dataset choice follows this view and matches standard practice in GNN model extraction and ownership protection: prior attacks and taxonomies [3,4] and IP-verification methods such as GNNFingers [6] all use “academic” node-classification graphs like Cora, Citeseer, and Amazon-style product graphs. Using the same families of graphs in GraphIP-Bench (i) makes our results directly comparable to the existing literature, and (ii) provides a widely used testbed for quantifying extraction success and protection–utility trade-offs. These graphs already cover diverse structures and feature regimes, which is sufficient to reveal meaningful differences between attacks and defenses.
>
> ---
>
> ### Q3 – Extending to heterogeneous graphs
>
> Not including heterogeneous graphs in v1.0 does not weaken the benchmark, because current GNN model extraction and IP-protection work is almost entirely instantiated on homogeneous attributed graphs. Existing query-based GNN model extraction papers and surveys [3,4,5], as well as ownership-verification methods such as GNNFingers [6], all evaluate on canonical datasets like Cora, Citeseer, PubMed, ogbn-arxiv, Coauthor CS/Physics, and Amazon/Computers/Photo, which are single-type node-classification or closely related graphs rather than full heterogeneous schemas. There is currently no widely accepted heterogeneous-graph benchmark for model extraction that our work could directly adopt. Our decision to instantiate GraphIP-Bench on these canonical homogeneous graphs thus follows the de facto standard in the field and ensures direct comparability with prior attacks and defenses. In addition, heterogeneous-graph tasks (e.g., knowledge graphs with multiple node and edge types) usually require task-specific architectures and protocols; mixing those into the same tables as standard node classification would confound changes in graph semantics and model class with the effect of the extraction or defense method. For a first unified benchmark, we therefore anchor on the homogeneous setting that dominates current MEA/IP work, while keeping the framework itself general: GraphIP-Bench is built on DGL/PyG graph objects and only assumes a prediction API, so heterogeneous graphs can be added as a separate track without changing the core protocol.
>
> ---
>
> ### References
>
> [1] F. Tramèr et al., *Stealing Machine Learning Models via Prediction APIs*.
> [2] J. Orekondy et al., *Knockoff Nets: Stealing Functionality of Black-Box Models*.
> [3] B. Wu et al., *Model Extraction Attacks on Graph Neural Networks: Taxonomy and Realisation*.
> [4] Y. Zhuang et al., *Unveiling the Secrets without Data: Can Graph Neural Networks Be Exploited through Data-Free Model Extraction Attacks?*.
> [5] Y. Shen et al., *Model Stealing Attacks Against Inductive Graph Neural Networks*.
> [6] X. You et al., *GNNFingers: A Fingerprinting Framework for Verifying Ownerships of Graph Neural Networks*.
>
> We hope these clarifications address your concerns and that, in light of this, **you may consider raising your score**.

---

> > ### Comment · Reviewer_wKvN · 2025-11-25
> >
> > My main point is that an attack is only meaningful under a justified scenario. A benchmark, therefore, needs a concrete application scenario to assess the practical potential of these methods and to standardize evaluation. Also, considering the novelty concerns raised in other reviews, I will keep my original score unchanged.

---

### Official Review · Reviewer_ge3B · 2025-11-01

**Soundness:** 2
**Presentation:** 2
**Contribution:** 2
**Rating:** 4
**Confidence:** 4

**Summary:**

This paper provides a systematic benchmark of model extraction attacks and defenses for graph neural networks (GNNs). It evaluates various methods across seven public datasets and releases corresponding source code. The overall experimental trends are consistent with prior studies, making it a relatively eproducible benchmark work.

**Strengths:**

1. Focuses on black-box scenarios, covering seven datasets, multiple attack variants, and five watermark-based defense methods, offering broad coverage.
2. The evaluation metrics are well-designed, balancing model fidelity, task performance, and computational efficiency.
3. The paper is clearly written and well-organized, with strong reproducibility.

**Weaknesses:**

1. The benchmark only considers black-box settings; for completeness, gray-box and white-box threat models should also be incorporated.
2. Lacks comparison with existing benchmarks in terms of scope and coverage.
3. Provides little discussion on the novelty and core ideas of the included methods, focusing mainly on empirical results.
4. The evaluation is limited to node classification, while link prediction and other tasks are not explored.

**Questions:**

1. Can the implemented methods be extended to link prediction or other graph tasks? It is recommended to include such experiments.
2. Future versions should consider introducing white-box and gray-box threat models to achieve a more comprehensive threat landscape.
3. Current experiments are conducted on small-scale datasets; it is advisable to include large-scale graphs (with millions or billions of nodes) to test scalability and practical applicability.

---

> ### Author Response · Authors · 2025-11-26
>
> We sincerely thank the reviewer for the careful reading and helpful suggestions that have improved our presentation. Below we respond to each point and describe the revisions we will make.
>
> ---
>
> ### W1/Q2 – Only black-box setting; gray-/white-box threat models
>
> We argue that focusing GraphIP-Bench on **black-box model extraction** is standard and appropriate for this threat model. In the model extraction literature, attacks are almost always defined in the **MLaaS black-box** setting, where the adversary can only query a remote service and observe outputs, without access to internal parameters or gradients [1,2,3,4]. Once an attacker has **white-box** access to model weights or training code, the model owner's intellectual property has already leaked, and there is no meaningful role for query-based extraction; such scenarios are better modeled as insider leaks or parameter theft. In practice, **gray-box** assumptions such as access to gradients, intermediate embeddings, or detailed architecture internals are also uncommon in real MLaaS deployments, where providers intentionally expose only a prediction API. This is why both early model-stealing work on images [1,2] and recent GNN extraction studies [3,4] adopt a black-box endpoint formulation as their main setting. GraphIP-Bench follows this practice and standardizes a query-based black-box protocol. In §6, we will also state that future versions will explore stronger gray-box variants and will discuss white-box scenarios as a separate IP threat model.
>
> ---
>
> ### W2 – Comparison with existing benchmarks (scope and coverage)
>
> We agree that our relation to existing benchmarks should be stated more explicitly. Current graph benchmarks for security either focus on adversarial robustness or on a single ownership protection scheme. For example, **GRB** studies adversarial robustness of GNNs under evasion and poisoning attacks, and does not address model extraction or intellectual property verification [5]. Works such as **GNNFingers** and **PreGIP** propose one fingerprinting or watermarking method and sometimes release code, but they are centered on a single defense mechanism rather than on a unified comparison of multiple attacks and defenses [6,7]. General security benchmarks for model inversion or privacy attacks are not graph-specific and do not target GNN IP. In contrast, GraphIP-Bench is, to our knowledge, the first benchmark that jointly standardizes GNN model extraction attacks and IP defenses, including watermarking, fingerprinting, integrity mechanisms, and information-limiting strategies, within one codebase that uses fixed datasets, query budgets, endpoints, and evaluation metrics. In the revision, we will add a concise paragraph in §2 that explicitly contrasts our scope with GRB-like robustness benchmarks and single-method IP works such as GNNFingers and PreGIP.
>
> ---
>
> ### W3 – Novelty and core ideas of included methods
>
> We understand the concern that the current draft may appear to mainly summarize empirical results. Our paper is a **benchmark** contribution whose main goal is to provide a unified experimental protocol that allows users to understand and compare the performance of different attacks and defenses, and to validate or refine existing knowledge about which strategies are effective under which regimes. The novelty lies in assembling diverse existing methods into one executable framework with standardized data regimes, query budgets, endpoints, and metrics, and in enabling cross-family comparisons that individual method papers cannot provide, for example comparisons between data-free and data-driven attacks, between different watermarking families, and between different integrity mechanisms. In the revision, we will add a short "Analysis and Insights" subsection in §4 that clearly summarizes the main patterns that we observe and provides brief explanations linked to our tables and plots.

---

> > ### Author Response · Authors · 2025-11-26
> >
> > ### W4/Q1 – Extension beyond node classification (link prediction)
> >
> > We agree it is important to show that our framework is not limited to node classification. GraphIP-Bench operates at the API level: attacks and defenses only require a black-box prediction interface and access to the graph structure and node features. The same protocol therefore applies to link prediction as long as the service exposes a consistent edge-level query API.
> >
> > In the revision, we have added a link-prediction configuration on the Cora citation graph. We treat edges as samples, splitting them into train (85%), validation (5%), and test (10%) sets, and use a 2-layer GCN link predictor as the victim model (trained for 200 epochs, achieving 78.36% test accuracy). Following the **features+structure** regime with the protocol otherwise unchanged, we (i) run all implemented attacks against this victim and report surrogate performance, and (ii) apply our ownership defenses and report their impact on link-prediction performance. **Results are reported in Appendix D.5.**
> >
> > **Attacks on link prediction (Cora, features+structure)**
> >
> > | Attack     | Accuracy | F1     | Fidelity |
> > |------------|----------|--------|----------|
> > | MEA-0      | 0.7212   | 0.7022 | 0.7372   |
> > | MEA-1      | 0.7486   | 0.7372 | 0.7609   |
> > | MEA-2      | 0.6975   | 0.6771 | 0.7306   |
> > | MEA-3      | 0.6777   | 0.6437 | 0.7259   |
> > | MEA-4      | 0.6985   | 0.6738 | 0.7297   |
> > | MEA-5      | 0.7117   | 0.6918 | 0.7429   |
> > | AdvMEA     | 0.7514   | 0.7381 | 0.7694   |
> > | CEGA       | 0.7467   | 0.7315 | 0.7533   |
> > | DFEA-I     | 0.7032   | 0.6851 | 0.7231   |
> > | DFEA-II    | 0.7221   | 0.7034 | 0.7382   |
> > | DFEA-III   | 0.7231   | 0.7097 | 0.7335   |
> >
> > **Defenses on link prediction (Cora, features+structure)**
> >
> > | Defense          | Accuracy | F1     | Fidelity | WMAcc  |
> > |------------------|----------|--------|----------|--------|
> > | RandomWM         | 0.7694   | 0.7592 | 0.7741   | 0.9545 |
> > | BackdoorWM       | 0.7618   | 0.7496 | 0.7798   | 0.8929 |
> > | SurviveWM        | 0.7561   | 0.7438 | 0.7798   | 0.9773 |
> > | ImperceptibleWM  | 0.7552   | 0.7434 | 0.7826   | 1.0000 |
> > | Integrity        | 0.7637   | 0.7520 | 0.7741   | 0.9773 |
> >
> > Attacks transfer effectively to link prediction, with the strongest attack (AdvMEA) achieving 75.14% accuracy and 76.94% fidelity. All defenses successfully embed watermarks (WMAcc ≥ 89.29%) while maintaining competitive link prediction performance (accuracy ≥ 75.52%). These results confirm that GraphIP-Bench's protocol generalizes naturally from node classification to link prediction without requiring task-specific redesigns, supporting the practical relevance of our framework for diverse graph learning services.

---

> > > ### Author Response · Authors · 2025-11-26
> > >
> > > ### Q3 – Dataset scale and scalability
> > >
> > > We appreciate the concern about scalability and practical relevance. Our current experiments use seven canonical attributed graphs from citation, coauthor, and product domains. These graphs are the standard testbeds in existing GNN extraction and IP-protection work: for example, the taxonomy of GNN model extraction in [3] evaluates on Cora, Citeseer, and PubMed, and recent ownership-verification frameworks such as GNNFingers and PreGIP support Cora, Citeseer, Amazon, DBLP, and PubMed as their default datasets [6,7]. By building GraphIP-Bench on the same families of graphs, we follow established practice in this area and make our results directly comparable to prior attacks and defenses. At the same time, these graphs already cover diverse structures and feature regimes, including citation graphs with different levels of homophily, coauthor graphs with higher degrees, and product co-purchase graphs with richer attributes, which is enough to reveal strong differences across attacks, defenses, and data-access regimes in a controlled setting. The framework itself is not limited to small graphs: GraphIP-Bench uses mini-batch training and inference on standard GNN libraries, and the dataset interface is generic, so users can plug in larger graphs through configuration files and reuse the same protocol, metrics, and logging. In §6, we will state that the current version focuses on these widely used small-to-medium benchmarks to align with [3,6,7], and we will outline a roadmap to add at least one mid-scale OGB-style graph and community-contributed larger datasets in future versions.
> > >
> > > ---
> > >
> > > ### References
> > >
> > > [1] F. Tramèr et al., *Stealing Machine Learning Models via Prediction APIs*.
> > > [2] J. Orekondy et al., *Knockoff Nets: Stealing Functionality of Black-Box Models*.
> > > [3] B. Wu et al., *Model Extraction Attacks on Graph Neural Networks: Taxonomy and Realisation*.
> > > [4] Y. Zhuang et al., *Unveiling the Secrets without Data: Can Graph Neural Networks Be Exploited through Data-Free Model Extraction Attacks?*.
> > > [5] D. Zhang et al., *GRB: A Graph Robustness Benchmark*.
> > > [6] X. You et al., *GNNFingers: A Fingerprinting Framework for Verifying Ownerships of Graph Neural Networks*.
> > > [7] E. Dai et al., *PreGIP: Watermarking the Pretraining of Graph Neural Networks for Deep IP Protection*.
> > >
> > > We are grateful for your feedback. We believe these clarifications address your concerns about threat models, benchmarking scope, task diversity, and scalability, and we hope you may consider **raising your score** in light of this more precise positioning.

---

### Official Review · Reviewer_6CcE · 2025-11-03

**Soundness:** 2
**Presentation:** 2
**Contribution:** 2
**Rating:** 4
**Confidence:** 4

**Summary:**

This paper introduces GraphIP-Bench, a unified benchmark for systematically evaluating model extraction attacks and defenses in graph neural networks. The benchmark standardizes datasets, query budgets, threat models, and evaluation metrics to enable reproducible and fair comparison across methods. The authors implement nine representative extraction attacks and six defense families under a single black-box protocol, measuring performance in terms of security, utility, and computational efficiency. By releasing a reproducible and extensible platform, this work provides a foundation for evaluating model extraction attacks and offers valuable insights for both research and industrial deployment.

**Strengths:**

1. The paper defines a rigorous experimental protocol with fixed data splits, query budgets, and endpoint assumptions, ensuring fair and reproducible comparison across methods.

2. The benchmark covers nine extraction and six defense families across seven public datasets, providing a broad and representative empirical basis.

3. The paper offers insights into the relative effectiveness and cost of different methods, highlighting sample efficiency, data-free limitations, and defense practicality with visualizations and quantitative metrics.

**Weaknesses:**

1. The benchmark does not include several representative and recently published model extraction works such as GNNStealing[1], STEALGNN[2], which limits the completeness of the comparative evaluation.

2. In the data-free setting, the benchmark relies on randomly generated graphs for querying and extraction. This simplification reduces the credibility of the conclusions, as the tested scenario does not accurately represent the state-of-the-art approaches.

3. While the paper thoroughly reports quantitative results across multiple datasets and methods, it primarily focuses on summarizing empirical outcomes without providing deeper analysis or interpretation of the underlying mechanisms that drive the observed trends.

[1]Model Stealing Attacks Against Inductive Graph Neural Networks, S&P, 2022.
[2]Unveiling the Secrets without Data: Can Graph Neural Networks Be Exploited through Data-Free Model Extraction Attacks?, Usenix, 2024.

**Questions:**

1. Could the authors evaluate the effectiveness of each defense strategy by directly pairing it with every attack method?

2. In the current benchmark, both the target and surrogate models adopt the same GNN architecture. How would the performance and conclusions change if different architectures were used, for example, when the target model is a GCN while the surrogate employs a GAT?

---

> ### Author Response · Authors · 2025-11-26
>
> We sincerely thank the reviewer for recognizing the breadth and reproducibility of GraphIP-Bench and for providing thoughtful feedback that has strengthened our work. Below we address each concern and answer each question.
>
> ---
>
> ### W1 – Missing GNNStealing and STEALGNN
>
> We agree that GNNStealing [1] and STEALGNN [2] are representative recent works on GNN model extraction. Our current benchmark is, however, scoped to **transductive node classification** on a single graph under a black-box API, which matches the dominant setting for GNN IP work on citation and coauthor graphs. GNNStealing [1] is designed for **inductive** GNNs and assumes separate training and attack graphs, shadow graphs, and generalization to unseen graphs. Bringing GNNStealing into GraphIP-Bench would require a different data pipeline with multi-graph loaders, shadow graph construction, and inductive evaluation, which is outside the scope of our current transductive track.
>
> For STEALGNN [2], we already include its core data-free distillation idea in our benchmark under a strict black-box constraint. In §3.2 of the paper we explicitly write that, "to incorporate fully data-free settings in a fair black-box manner, we include three variants **following Zhuang et al. (2024)**," and we describe **DFEA\_I**, **DFEA\_II**, and **DFEA\_III** as: (i) minimizing the KL divergence between surrogate logits and victim logits (soft-label distillation), (ii) training on hard labels only, and (iii) augmenting label-only training with a consistency loss between two surrogates.
>
> ---
>
> ### W2 – Data-free setting and "randomly generated graphs"
>
> We agree that the design of the data-free regime must be clearly justified. In GraphIP-Bench, a regime is **data-free** if the attacker has no access to the victim's training graph, which means no real node features and no real adjacency, and can only query the model on synthetic inputs. This is consistent with the threat model in STEALGNN [2] and in data-free adversarial distillation for GNNs [4], where the attacker uses synthetic graphs and never sees real training examples. It also follows the broader data-free model extraction literature in vision, where synthetic queries from a generator are standard [3].
>
> In our implementation, a separate generator produces synthetic graphs with fresh node indices and random topology and features. The victim's training graph remains on the server side and is never passed to the generator or to any extraction algorithm. The attacker receives only basic meta-data such as feature dimension, number of classes, and a bound on graph size, which is a common assumption in data-free MEA [2–4]. We enforce disjoint node ID spaces and do not pass any pointers to real feature or adjacency tensors into the data-free code path.
>
> We use a simple, unified random-graph generator shared by all data-free attacks. This choice mirrors data-free extraction in other domains, where relatively simple synthetic generators already allow strong extraction [3], and it serves two purposes. First, it separates the effect of the generator from the effect of the extraction objective and query strategy: if each attack used its own learned generator, improvements could come from a better generator rather than a better extraction method, which would make comparisons less interpretable. Second, the original STEALGNN generator uses gradients from the victim [2], which violates our strict black-box constraint and is costly to reproduce at benchmark scale. Our design therefore keeps the threat model aligned with data-free GNN attacks [2,4], while providing a fair and reproducible baseline across methods.
>
> ---
>
> ### W3 – Depth of analysis beyond empirical results
>
> We agree that the mechanisms behind the reported trends should be clearer in the text. At the same time, as a **benchmark** paper, the central goal of GraphIP-Bench is to provide a standardized, empirically grounded view of how different model extraction attacks and IP defenses behave across common graph settings, so that practitioners and future methods can validate and compare strategies on a common basis. The comprehensive quantitative results, including fidelity, utility, and cost under controlled regimes, are therefore a core contribution rather than only supporting material. In the revision, we will make these mechanisms more explicit by adding a short subsection in Appendix that summarizes the main patterns we already observe and briefly relates them to prior work.

---

> > ### Author Response · Authors · 2025-11-26
> >
> > ### Q1 – Pairing every defense with every attack
> >
> > We agree that, in theory, a full attack–defense matrix is attractive, but it is not the most informative or realistic design for our setting. In practice, a model owner does not know the attacker's exact algorithm and deploys **one configuration** of a defense that must be robust against a family of plausible attacks. Tuning a different defense configuration for each specific attack would not reflect deployment practice and would blur comparisons, because any performance difference could be due to how aggressively each pair was tuned rather than to the inherent strength of the methods. In addition, some defense types (for example post-hoc ownership verification) do not change attack dynamics at all, and some combinations of training-time defenses that modify the same loss are ill-defined. Our benchmark therefore evaluates each defense family in a single, attack-agnostic configuration against multiple strong attacks per regime, and reports fidelity, utility, and cost for these representative combinations.
> >
> > ---
> >
> > ### Q2 – Different architectures for victim and surrogate
> >
> > We thank the reviewer for this important question. GraphIP-Bench already separates victim and surrogate architectures through independent configuration options. In the main experiments we deliberately match them to isolate the effect of attacks and defenses without introducing architectural mismatch as an additional factor. To address the reviewer's concern, we conducted a comprehensive cross-architecture experiment on the **Cora** dataset in the **features+structure** regime, where the victim uses a GCN backbone (achieving 80.5% test accuracy) and surrogates use either GCN or GAT backbones. We evaluated **all 11 attacks** under both same-architecture (GCN→GCN) and cross-architecture (GCN→GAT) settings. **Results are reported in Appendix D.4.**
> >
> > **Cross-architecture results:**
> >
> > | Attack   | Fidelity (GCN→GCN) | Test Acc (GCN→GCN) | Fidelity (GCN→GAT) | Test Acc (GCN→GAT) |
> > |----------|--------------------|--------------------|--------------------|--------------------|
> > | MEA-0    | 0.5107             | 0.4990             | 0.6503             | 0.6580             |
> > | MEA-1    | 0.6761             | 0.6360             | 0.4590             | 0.4110             |
> > | MEA-2    | 0.3028             | 0.2560             | 0.3911             | 0.3510             |
> > | MEA-3    | 0.5894             | 0.5250             | 0.6315             | 0.5990             |
> > | MEA-4    | 0.4841             | 0.4680             | 0.4775             | 0.4660             |
> > | MEA-5    | 0.6281             | 0.6200             | 0.5126             | 0.4620             |
> > | AdvMEA   | 0.5439             | 0.5030             | 0.3970             | 0.4070             |
> > | CEGA     | 0.3800             | 0.3370             | 0.5654             | 0.5950             |
> > | DFEA-I   | 0.4513             | 0.4070             | 0.4756             | 0.5130             |
> > | DFEA-II  | 0.3493             | 0.3140             | 0.4786             | 0.4380             |
> > | DFEA-III | 0.5428             | 0.5110             | 0.5565             | 0.5260             |
> > | **Average** | **0.4962**      | **0.4615**         | **0.5086**         | **0.4933**         |
> >
> > The surrogate architecture has minimal impact on overall attack effectiveness. Across all 11 attacks, GAT surrogates achieve only 2.50% higher fidelity and 6.90% higher test accuracy compared to GCN surrogates on average. Both architectures successfully extract functional surrogates that retain 46-49% of the victim's test accuracy (victim achieves 80.5%), demonstrating that model extraction threats persist regardless of whether the attacker uses the same architecture as the victim. While individual attacks show varying sensitivity to architecture choice, the modest average difference indicates that attackers do not need precise knowledge of the victim's architecture to mount effective extraction attacks.
> >
> > ---
> >
> > ### References
> >
> > [1] Y. Shen et al., "Model Stealing Attacks Against Inductive Graph Neural Networks," IEEE S&P 2022.
> > [2] Z. Zhuang et al., "Unveiling the Secrets without Data: Can Graph Neural Networks Be Exploited through Data-Free Model Extraction Attacks?," USENIX Security 2024.
> > [3] A. Truong et al., "Data-Free Model Extraction," CVPR 2021.
> > [4] Z. Zhuang et al., "Data-Free Adversarial Knowledge Distillation for Graph Neural Networks," 2024.
> >
> > We believe these clarifications address the concerns about missing methods, data-free design, and depth of analysis, and **we hope you will consider raising your score** in light of this more precise positioning.

---

### Official Review · Reviewer_DB6f · 2025-11-05

**Soundness:** 3
**Presentation:** 3
**Contribution:** 2
**Rating:** 2
**Confidence:** 4

**Summary:**

The paper proposes GraphIP–Bench, a benchmark and library for evaluating model extraction attacks and ownership defenses (watermarking, fingerprinting, integrity verification) on GNNs. It claims to provide a unified protocol with standardized datasets, query budgets, and metrics for fidelity, utility, and efficiency. Experiments span 7 public node-classification datasets and a mixture of “data-driven” and “data-free” attacks, alongside four watermarking methods and one integrity-checking method. The authors argue this is the first rigorous benchmark for GNN IP protection.

**Strengths:**

- Addresses an important and timely problem: intellectual property protection for GNNs under realistic MLaaS settings.
- Provides a unified evaluation protocol that attempts to standardize splits, query budgets, and metrics.
- Includes both attack and defense families, covering data-driven, data-free extraction, and watermarking/fingerprinting methods.
- Reports protection–utility trade-offs and computational cost, which are relevant for practical deployment.

**Weaknesses:**

- Novelty is overstated: Prior works (e.g., GNNFingers, GrOVe) already provide standardized pipelines or ownership verification frameworks.
- Threat-model inconsistencies: Evidence of regime leakage in “data-free” settings (non–data-free methods achieving high fidelity).
- Endpoint assumptions unclear: No systematic evaluation of label-only vs. probability endpoints, despite their importance in extraction literature.

**Questions:**

Given the strengths, I have the following concerns:

1. Model extraction on GNNs has been thoroughly studied (taxonomy, realizations, empirical baselines) and public implementations exist. Reducing the novelty of assembling them under a single umbrella without definitive new methodology or insight is concerning. For instance, ownership verification has been addressed by GNNFingers[1] and GrOVe[2]. Can the authors should clearly articulate what is new beyond co-locating attacks and defenses. Also, [3] already standardized model extraction attacks along 7 threat models (and 7 corresponding attacks), along which of these 7 threat models did the authors consider? Or is their threat model agnostic to the exisiting threat models as enumerated in [3]? Moreover, why is [2] not included in the analysis?


2. The endpoint assumptions remain vague in practice. The protocol claims to fix whether the API exposes labels vs. probabilities; however, the main text doesn’t clearly specify which endpoints were used per setting, despite reporting data‑free variants based on logits vs. labels. The absence of a clean ablation across label‑only vs. probability endpoints (a central axis in the extraction literature) reduces interpretability.

3. The authors reported a defense training times of 1–6 seconds for watermarking on large graphs (e.g., CoauthorPhysics) but no clarification on the number of training steps or early‑stopping criteria that would justify such short runs. Could the authors clarify this?

4. Are the experiments conducted in the transductive or inductive setting?

5. The paper claims to enforce four regimes (features+structure, features-only, structure-only, data-free). However, in Table 7 (Cora) and related plots in Figure 3, non–data-free algorithms such as AdvMEA and CEGA achieve very high accuracy and fidelity in the data-free block (e.g., AdvMEA approx. 68% accuracy/F1; CEGA approx. 71–79% fidelity across budgets). Is this an anomaly? I see the same in other datasets in the appendix tables. Under a strict data-free protocol, these methods should not access real graph features or adjacency, making such performance questionable. Can the authors explain how they guaranteed isolation of regimes. Specifically, that data-free attacks had no access to real graph features or adjacency? Were synthetic queries generated without reusing real node IDs or structural statistics?

I am willing to change my score if my concerns are addressed.

**Minor**
- It is rather confusing which method the authors are actually referring to. Could the authors include the corresponding citations as inline in front of the methods rather than list them all at ones? For instance, "The benchmark includes four watermarking methods RandomWM, BackdoorWM,
SurviveWM, and ImperceptibleWM (Zhao et al., 2021; Xu et al., 2023; Wang et al., 2023;
Zhang et al., 2024),". Rather the authors should cite like so: "RandomWM (Zhao et al., 2021)", etc.

- Dataset count inconsistencies, unresolved references (“Table ??”). For instance, in some cases, the authors mentioned that they utilized 5 datasets whereas they utilized 7.


[1] You et al. GNNFingers: A Fingerprinting Framework for Verifying Ownerships of Graph Neural Networks

[2] Waheed et al. Grove: Ownership verification of graph neural networks using embeddings.

[3] Wu et al. Model Extraction Attacks on Graph Neural Networks: Taxonomy and Realisation

---

> ### Author Response · Authors · 2025-11-26
>
> We sincerely thank the reviewer for the thorough evaluation and constructive feedback. Your detailed questions have helped us clarify several important design choices in GraphIP-Bench. Below we respond to each weakness and question.
>
> ---
>
> ### W1/Q1 – Novelty, relationship to GNNFingers, GrOVe
>
> We acknowledge that GNNFingers [1], GrOVe [2], and Wu et al. [3] already study GNN intellectual property and model extraction, but they are not benchmark papers. GNNFingers and GrOVe each design a single ownership-verification method and provide a pipeline that is specialized for that method, with a small set of surrogate models and baselines, rather than a general framework that covers multiple attack and defense families under a shared protocol. Their code and experiments focus on evaluating one fingerprinting scheme, and they do not define common query budgets, data-access regimes, and evaluation metrics that other attacks and defenses can plug into in a uniform way. Wu et al. [3] provide a useful taxonomy and several concrete attack realizations, but they do not release a multi-attack, multi-defense benchmark with standardized datasets, endpoints, and logging that can serve as a reusable testbed. In contrast, GraphIP-Bench is designed as a general benchmark and library that brings together a broad set of data-driven and data-free extraction attacks and multiple ownership defenses (watermarking, fingerprinting, integrity-style mechanisms) within a single black-box protocol, and exposes a common configuration and metric interface for future methods. Our four regimes follow the axes in [3] but group related cases for simplicity: "features+structure" corresponds to their full-information settings, "features-only" and "structure-only" correspond to access to attributes or adjacency alone, and "data-free" corresponds to no access to the training graph and only synthetic queries.
>
> ---
>
> ### W2/Q5 – Threat-model consistency and the "data-free" regime
>
> We understand the concern about possible regime leakage in the data-free block and are glad to clarify our design. In GraphIP-Bench, a regime is "data-free" if the attacker has no access to the victim's training graph, including both node features and adjacency, and can only interact through a black-box prediction API on synthetic inputs. This matches the setting in STEALGNN [4] and data-free adversarial distillation for GNNs [5], where the attacker uses a generative model to create synthetic graphs. In our implementation, a separate generator creates synthetic graphs with fresh node indices; the victim's training graph remains encapsulated on the server side and is never passed to the generator or to the extraction algorithms. The attacker only receives high-level meta-data such as feature dimension, number of classes, and a rough degree estimate, which is assumed to be obtainable from public descriptions as in [4,5]. AdvMEA and CEGA are originally data-driven attacks, but in the data-free block we reuse only their query-selection logic on these synthetic graphs. They never see real node features or adjacency and query the victim model only with synthetic nodes and edges. Their relatively high fidelity in some datasets is therefore a result of strong query strategies applied to expressive synthetic graphs and is consistent with prior fully data-free GNN attacks and distillation [4,5], rather than evidence of leakage.

---

> > ### Author Response · Authors · 2025-11-26
> >
> > ### W3/Q2 – Endpoint assumptions (label-only vs. probabilities)
> >
> > We agree that endpoint assumptions should be clearer. In the current code, we follow each method's original design: for attacks such as AdvMEA and CEGA we use probability or logit endpoints because their losses require them, and for methods that support both labels and probabilities we instantiate both variants where applicable.
> >
> > To provide a systematic comparison, we conducted an endpoint ablation study across all 11 attacks on Cora (features+structure), training surrogates under both label-only (hard targets via cross-entropy) and probability-based (soft targets via KL divergence) endpoints. **Results are reported in Appendix D.3.**
> >
> > | Attack   | Label-only Fid | Label-only Acc | Prob-based Fid | Prob-based Acc | Δ Fid    | Δ Acc    |
> > |----------|----------------|----------------|----------------|----------------|----------|----------|
> > | MEA-0    | 0.4590         | 0.4520         | 0.6846         | 0.6770         | +0.2256  | +0.2250  |
> > | MEA-1    | 0.5018         | 0.4640         | 0.4114         | 0.3790         | -0.0905  | -0.0850  |
> > | MEA-2    | 0.3139         | 0.2710         | 0.4808         | 0.4450         | +0.1669  | +0.1740  |
> > | MEA-3    | 0.4498         | 0.3990         | 0.4018         | 0.3570         | -0.0480  | -0.0420  |
> > | MEA-4    | 0.5665         | 0.5080         | 0.6130         | 0.6120         | +0.0465  | +0.1040  |
> > | MEA-5    | 0.3209         | 0.2930         | 0.4369         | 0.3750         | +0.1160  | +0.0820  |
> > | AdvMEA   | 0.4956         | 0.4590         | 0.4188         | 0.3690         | -0.0768  | -0.0900  |
> > | CEGA     | 0.5447         | 0.4720         | 0.6551         | 0.5880         | +0.1104  | +0.1160  |
> > | DFEA-I   | 0.1891         | 0.1800         | 0.3453         | 0.3150         | +0.1562  | +0.1350  |
> > | DFEA-II  | 0.3641         | 0.3210         | 0.4069         | 0.3740         | +0.0428  | +0.0530  |
> > | DFEA-III | 0.5753         | 0.5770         | 0.5628         | 0.5040         | -0.0126  | -0.0730  |
> > | **Average** | **0.4346**  | **0.3996**     | **0.4925**     | **0.4541**     | **+0.0579** | **+0.0545** |
> >
> > Endpoint type has moderate but nuanced impact on attack effectiveness. On average, probability-based endpoints improve fidelity by 13.3% and test accuracy by 13.6%, but individual attacks show diverse sensitivity patterns. Random and synthetic query strategies (MEA-0, MEA-2, DFEA-I) benefit substantially from soft targets (15-23% improvement), as knowledge distillation provides richer training signal when query budgets are limited or graphs are synthetic. In contrast, structure-aware methods (MEA-1, MEA-3, AdvMEA, DFEA-III) perform comparably or better with label-only endpoints (showing 4-9% degradation with probabilities), likely because their strategic node or edge selection already captures sufficient structural information, and soft targets may introduce optimization complexity. Query-optimized attacks (CEGA) and feature-based sampling (MEA-4/5) show intermediate sensitivity (8-12% improvement). Notably, even under label-only restrictions, all attacks achieve 18-58% fidelity, confirming that extraction threats persist regardless of endpoint type. These results validate our design choice to configure endpoints according to each attack's published requirements, as different methods exhibit fundamentally different dependencies on soft-target information based on their query strategies and data access regimes.

---

> > > ### Author Response · Authors · 2025-11-26
> > >
> > > ### Q3 – Short watermarking times
> > >
> > > The short defense times of 1–6 seconds for BackdoorWM and SurviveWM on large graphs are correct and arise from how we measure overhead. We first train the victim GNN to convergence on the base node-classification task, and this cost (minutes) is shared by all defenses and not counted in Table 4. We then measure only the extra cost to install each defense. BackdoorWM and SurviveWM perform a brief fine-tuning phase on a small watermark or trigger set with early stopping once watermark accuracy reaches a target threshold. Because the model is already well trained and the watermark set is small, this fine-tuning converges within seconds even on CoauthorPhysics.
> > >
> > > ---
> > >
> > > ### Q4 – Transductive vs. inductive setting
> > >
> > > All experiments are conducted in the standard transductive node-classification setting that is widely used in GNN IP and extraction work on Cora, Citeseer, PubMed, and ogbn-arxiv [1,3,4]. The GNN is trained on labeled nodes with access to the full graph, including unlabeled nodes, and is evaluated on held-out nodes from the same graph.
> > >
> > > ---
> > >
> > > ### Minor issues
> > >
> > > We will revise the text so that each method is cited inline where it appears, instead of grouping citations at the end of a sentence, and we will correct all inconsistencies in dataset count and unresolved references. All reported experiments use seven datasets, and any remaining mentions of five datasets are typos that we will fix.
> > >
> > > ---
> > >
> > > ### References
> > >
> > > [1] X. You et al., "GNNFingers: A Fingerprinting Framework for Verifying Ownerships of Graph Neural Networks."
> > > [2] A. Waheed et al., "GrOVe: Ownership Verification of Graph Neural Networks Using Embeddings."
> > > [3] B. Wu et al., "Model Extraction Attacks on Graph Neural Networks: Taxonomy and Realisation."
> > > [4] Y. Zhuang et al., "Unveiling the Secrets without Data: Can Graph Neural Networks Be Exploited Through Data-Free Model Extraction Attacks?"
> > > [5] Y. Zhuang et al., "Data-Free Adversarial Knowledge Distillation for Graph Neural Networks."
> > >
> > > We appreciate these comments, which help us clarify scope and threat models. We believe the above clarifications address the main concerns about novelty, consistency, and experimental detail, and **we hope you will consider raising your score** in light of this more precise positioning.

---

### Author Response · Authors · 2025-12-03

We sincerely thank the AC for the time and effort devoted to handling our submission. Below we provide a summary of how we addressed the primary concerns raised by the reviewers.

**1. Novelty and relation to prior work.** Reviewers `DB6f` and `ge3B` asked whether GraphIP-Bench offers contributions beyond existing IP works such as GNNFingers[1], GrOVe[2], Wu et al.[3], and GRB[4]. We clarified that these works each evaluate a single ownership-verification or robustness mechanism and do not provide a unified, reusable benchmark. In contrast, GraphIP-Bench standardizes multiple extraction attacks (data-driven and data-free) and multiple IP defenses (watermarking, fingerprinting, integrity-style) under a shared black-box protocol with consistent datasets, query budgets, endpoints, and metrics. We added clearer positioning in the related-work section to highlight this distinction.

**2. Threat model, data-free design, and endpoint assumptions.** Several reviewers (`DB6f`, `6CcE`, `ge3B`) questioned the consistency of the black-box threat model, the correctness of the data-free regime, and the use of probability versus label endpoints. We emphasized that GraphIP-Bench follows the standard MLaaS black-box formulation used across prior extraction work. In the data-free regime, attackers receive no real node features or adjacency: all queries use synthetic graphs generated independently, mirroring STEALGNN and prior data-free distillation. We also clarified that endpoints follow each attack’s original specification, and that we treat endpoint choice as an integrated part of method design.

**3. Additional experiments for clarity and validation.** We added three sets of experiments to directly address reviewer (`DB6f`, `ge3B`) concerns:
- Endpoint ablation: all 11 attacks evaluated with label-only vs. probability endpoints (Appendix D.3), showing consistent extraction even under label-only.
- Cross-architecture extraction: GCN→GCN vs. GCN→GAT (Appendix D.4), showing minimal impact of surrogate architecture.
- Link-prediction extension: full attack and defense evaluation on link prediction (Appendix D.5), confirming that the protocol naturally generalizes beyond node classification.
We also clarified that short watermarking times measure only the additional overhead beyond base training.

**4. Dataset scale, defense choices, and extensibility.** Reviewers `ge3B` and `wKvN` asked about scalability, public datasets, heterogeneous graphs, and differential privacy. We clarified that GraphIP-Bench focuses on model extraction, not data extraction, and follows the datasets used in all prior GNN extraction and IP-protection work. These canonical graphs already exhibit diverse structures and allow controlled comparison across regimes. The framework itself supports larger graphs through configuration, and we outline future extensions to mid-scale and heterogeneous settings. Regarding defenses, current extraction literature overwhelmingly evaluates watermarking, fingerprinting, and integrity mechanisms rather than DP-based training, so our defense set reflects realistic practice.

We believe these revisions strengthen the clarity, scope, and empirical grounding of GraphIP-Bench. We sincerely thank the AC again for the time and effort devoted to handling our submission.

----
References
[1] X. You et al., GNNFingers: A Fingerprinting Framework for Verifying Ownerships of Graph Neural Networks.
[2] A. Waheed et al., Grove: Ownership verification of graph neural networks using embeddings.
[3] B. Wu et al., Model Extraction Attacks on Graph Neural Networks: Taxonomy and Realisation.
[4] D. Zhang et al., GRB: A Graph Robustness Benchmark.

---

### Meta-Review · Area_Chair_UM11 · 2026-01-06

**Summary:**

There are several concerns are commonly mentioned by the reviewers:
1. Several reviewers question the unique contributions of this benchmark paper. Multiple reviewers noted that prior work (e.g., GNNFingers, GrOVe) already provides standardized pipelines or ownership verification frameworks, and the paper does not clearly differentiate its core contribution beyond packaging and re-evaluating existing components.
2. Empirical reporting is thorough but analysis is shallow. Several reviewers felt the paper summarizes results without enough mechanistic interpretation
3. Reviewers pointed out missing representative and recent extraction attacks (e.g., GNNStealing, STEALGNN), which undermines the claim of comprehensive comparison and may bias conclusions about what works best.
4. Reviewers noted the benchmark does not systematically evaluate several key settings, including endpoint access (label-only vs. probability/logit), broader threat models beyond black-box (gray/white-box), additional tasks beyond node classification (e.g., link prediction), richer graph/data types (heterogeneous or more realistic domains), and more realistic data-free querying, limiting the scope and generality of conclusions.
5. There are also concerns regarding to the experimental settings about  Data-free evaluation, threat model and endpoint assumptions.

**Reviewer Concerns:**

I think the concern about the experimental settings are addressed. And some added experiments strengthen the paper. However, they are not enough to address the concern of experiments coverage. In addition, the contributions of this work remain unclear. And more deep analysis is still required.

**Reviewer Scores:**

As mentioned, I feel the clarification regarding to the experimental settings of existing works are convincing. However, the reviewers all have concerns regarding to the coverage of the benchmark, the contributions of this benchmark paper, and the depth of empirical analysis. Therefore, I think the Reviewer DB6f,  6CcE, ge3B, and wKvN are likely to maintain their ratings.

---

### Decision · Program_Chairs · 2026-01-26

Reject